# A self-consistent theory of Gaussian Processes captures feature learning effects in finite CNNs

**Gadi Naveh**
Racah Institute of Physics
Hebrew University
Jerusalem, Israel 91904
gad.mintz@mail.huji.ac.il

**Zohar Ringel**
Racah Institute of Physics
Hebrew University
Jerusalem, Israel 91904
zohar.ringel@mail.huji.ac.il

## Abstract

Deep neural networks (DNNs) in the infinite width/channel limit have received much attention recently, as they provide a clear analytical window to deep learning via mappings to Gaussian Processes (GPs). Despite its theoretical appeal, this viewpoint lacks a crucial ingredient of deep learning in finite DNNs, laying at the heart of their success — *feature learning*. Here we consider DNNs trained with noisy gradient descent on a large training set and derive a self-consistent Gaussian Process theory accounting for *strong* finite-DNN and feature learning effects. Applying this to a toy model of a two-layer linear convolutional neural network (CNN) shows good agreement with experiments. We further identify, both analytically and numerically, a sharp transition between a feature learning regime and a lazy learning regime in this model. Strong finite-DNN effects are also derived for a non-linear two-layer fully connected network. We have numerical evidence demonstrating that the assumptions required for our theory hold true in more realistic settings (Myrtle5 CNN trained on CIFAR-10). Our self-consistent theory provides a rich and versatile analytical framework for studying strong finite-DNN effects, most notably - feature learning.

## 1 Introduction

The correspondence between Gaussian Processes (GPs) and deep neural networks (DNNs) has been instrumental in advancing our understanding of these complex algorithms. Early results related randomly initialized strongly over-parameterized DNNs with GP priors [33, 24, 30]. More recent results considered training using gradient flow (or noisy gradients), where DNNs, potentially following some ensembling, map to Bayesian inference on GPs governed by the neural tangent kernel [21, 25, 18] (or the NNGP kernel [32]). These correspondences carry over to a wide variety of architectures, going beyond fully connected networks (FCNs) to convolutional neural networks (CNNs) [4, 36], recurrent neural networks (RNNs) [2] and even attention networks [20]. They provide us with closed analytical expressions for the outputs of strongly over-parameterized trained DNNs, which have been used to make accurate predictions for DNN learning curves [11, 9, 8].

Despite their theoretical appeal, GPs are unable to capture *feature learning* [47, 46], which is a well-observed key property of trained DNNs. Indeed, it was noticed [21] that as the width tends to infinity, the neural tangent kernel (NTK) tends to a constant kernel that does not evolve during training and the weights in hidden layers change infinitesimally from their initialization values. This regime of training was thus dubbed *lazy training* [10]. Other studies showed that for CNNs trained on image classification tasks, the feature learning regime generally tends to outperform the lazy regime [15, 14, 23]. Clearly, working in the feature learning regime is also crucial for performing *transfer learning* [45, 46].

35th Conference on Neural Information Processing Systems (NeurIPS 2021), .

It is therefore desirable to have a theoretical approach to deep learning which enjoys the generality and analytical power of GPs while capturing feature learning effects in finite DNNs. Here we make several contributions towards this goal:

1. We show that the mean predictor of a finite DNN trained on a large data set with noisy gradients, weight decay and MSE loss, can be obtained from GP regression on a shifted target (§3). Central to our approach is a non-linear self-consistent equation involving the higher cumulants of the finite DNN (at initialization) which predicts this target shift.

2. Using this machinery on a toy model of a two-layer linear CNN in a teacher-student setting, we derive explicit analytical predictions which are in very good agreement with experiments even well away from the GP/lazy-learning regime (large number of channels, $C$) thus accounting for *strong* finite-DNN corrections (§4.1). The match to empirical values of our theory in this toy model is shown to be clearly superior to two alternative theoretical predictions, one of them being a purely perturbative theory. Similarly strong corrections to GPs, yielding qualitative improvements in performance, are demonstrated for the quadratic two-layer fully connected model of Ref. [28].

3. We show how our framework can be used to study statistical properties of weights in hidden layers. In particular, in the CNN toy model, we identify, both analytically and numerically, a sharp transition between a feature learning phase and a lazy learning phase (§4.1.4). We define the feature learning phase as the regime where the features of the teacher network leave a clear signature in the spectrum of the student's hidden weights posterior covariance matrix. In essence, this phase transition is analogous to the transition associated with the recovery of a low-rank signal matrix from a noisy matrix taken from the Wishart ensemble, when varying the strength of the low-rank component [6].

## 1.1 Additional related work

Several previous papers derived leading order finite-DNN corrections to the GP results [32, 44, 13, 39]. While these results are in principle extendable to any order in perturbation theory, such high order expansions have not been studied much, perhaps due to their complexity. These previous perturbative approaches are expected to be satisfactory only when the GP limit already gives a reasonable approximation to the DNN behavior. In contrast, we develop an analytically tractable *non-perturbative* approach which we find crucial for obtaining non-negligible feature learning and associated performance enhancement effects.

Previous works [15, 14, 43] studied how the behavior of infinite DNNs depends on the scaling of the top layer weights with its width. In [45] it is shown that the standard and NTK parameterizations of a neural network do not admit an infinite-width limit that can learn features, and instead suggest an alternative parameterization which can learn features in this limit. While unifying various viewpoints on infinite DNNs, this approach does not immediately lend itself to analytical analysis of the kind proposed here. Also, in our work, feature learning as a purely finite-width effect.

Several works [38, 27, 16, 3] show that finite width models can generalize either better or worse than their infinite width counterparts, and provide examples where the relative performance depends on the optimization details, the DNN architecture and the statistics of the data. Another study [1] considered cases where the enhanced flexibility of a finite DNN relative to its infinite width counterpart provides the former with superior performance. Here we demonstrate analytically that finite DNNs outperform their GP counterparts when the latter have a prior that lacks some constraint found in the data (e.g. positive-definiteness [28] or translation invariance / equivariance [36]).

Deep linear networks (FCNs and CNNs) similar to our CNN toy example have been studied in the literature [5, 40, 22, 17]. These studies use different approaches and assumptions and do not discuss the target shift mechanism which applies also for non-linear CNNs. In addition, their analytical results hinge strongly on linearity whereas our approach could be useful whenever several leading cumulants of the DNN output are known or can be approximated, and we give an example of this in a non-linear setting in §4.2. While there are some similarities between the phase transition we discuss in §4.1.4 and the one appearing in Ref. [22], there are several important differences, to mention one of them: that study considers the statistics of a noisy teacher, whereas we consider the statistics of the first layer of the trained student network.

Two concurrent works [48, 34] derived exact expressions for the output priors of finite FCNs induced by Gaussian priors over their weights. However, these results only apply to the limited case of a *prior* over a *single* training point and only for a FCN. In contrast, our approach applies to the setting of a large training set, it is not restricted to FCNs and yields results for the posterior predictions, not the prior. Focusing on deep linear fully connected DNNs, recent work [26] derived analytical finite-width renormalization results for the GP kernel, by sequentially integrating out the weights of the DNN, starting from the output layer and working backwards towards the input. Our analytical approach, its scope, and the models studied here differ substantially from that work.

## 2  Preliminaries

We consider a fixed set of $n$ training inputs $\{\mathbf{x}_\mu\}_{\mu=1}^n \subset \mathbb{R}^d$ and a single test point $\mathbf{x}_*$ over which we wish to model the distribution of the outputs of a DNN. We consider a generic DNN architecture where for simplicity we assume a scalar output $f(\mathbf{x}) \in \mathbb{R}$. The learnable parameters of the DNN that determine its output, are collected into a single vector $\theta$. We pack the outputs evaluated on the training set and on the test point into a vector $\vec{f} \equiv (f(\mathbf{x}_1), \ldots, f(\mathbf{x}_n), f(\mathbf{x}_{n+1})) \in \mathbb{R}^{n+1}$, where we denoted the test point as $\mathbf{x}_* = \mathbf{x}_{n+1}$. We train the DNNs using *full-batch* gradient decent with weight decay and external white Gaussian noise. The discrete dynamics of the parameters are thus

$$\theta_{t+1} - \theta_t = -\left(\gamma\theta_t + \nabla_\theta \mathcal{L}(f_\theta)\right)\eta + 2\sigma\sqrt{\eta}\xi_t \tag{1}$$

where $\theta_t$ is the vector of all network parameters at time step $t$, $\gamma$ is the strength of the weight decay, $\mathcal{L}(f_\theta)$ is the loss as a function of the DNN output $f_\theta$ (where we have emphasized the dependence on the parameters $\theta$), $\sigma$ is the magnitude of noise, $\eta$ is the learning rate and $\xi_t \sim \mathcal{N}(0, I)$. As $\eta \to 0$ this discrete-time dynamics converge to the continuous-time Langevin equation given by $\dot{\theta}(t) = -\nabla_\theta\left(\frac{\gamma}{2}\|\theta(t)\|^2 + \mathcal{L}(f_\theta)\right) + 2\sigma\xi(t)$ with $\langle\xi_i(t)\xi_j(t')\rangle = \delta_{ij}\delta(t - t')$, so that as $t \to \infty$ the DNN parameters $\theta$ will be sampled from the equilibrium Gibbs distribution $P(\theta)$.

As shown in [32], the parameter distribution $P(\theta)$ induces a posterior distribution over the trained DNN outputs $P(\vec{f})$ with the following partition function:

$$Z\left(\vec{J}\right) = \int d\vec{f} P_0\left(\vec{f}\right) \exp\left(-\frac{1}{2\sigma^2}\mathcal{L}\left(\{f_\mu\}_{\mu=1}^n, \{g_\mu\}_{\mu=1}^n\right) + \sum_{\mu=1}^{n+1} J_\mu f_\mu\right) \tag{2}$$

Here $P_0(\vec{f})$ is the prior generated by the finite-DNN with $\theta$ drawn from $\mathcal{N}(0, 2\sigma^2/\gamma)$ where the weight decay $\gamma$ may be layer-dependent, $\{g_\mu\}_{\mu=1}^n$ are the training targets and $\vec{J}$ are source terms used to calculate the statistics of $f$. We keep the loss function $\mathcal{L}$ arbitrary at this point, committing to a specific choice in the next section. As standard [19], to calculate the posterior mean at any of the training points or the test point $\mathbf{x}_{n+1}$ from this partition function one uses

$$\forall\mu \in \{1, \ldots, n+1\}: \qquad \langle f_\mu\rangle = \partial_{J_\mu} \log Z\left(\vec{J}\right)\Big|_{\vec{J}=\vec{0}} \tag{3}$$

## 3  A self-consistent theory for the posterior mean and covariance

In this section we show that for a large training set, the posterior mean predictor (Eq. 3) amounts to GP regression on a shifted target ($g_\mu \to g_\mu - \Delta g_\mu$). This shift to the target ($\Delta g_\mu$) is determined by solving certain self-consistent equations involving the cumulants of the prior $P_0(\vec{f})$. For concreteness, we focus here on the MSE loss $\mathcal{L} = \sum_{\mu=1}^n (f_\mu - g_\mu)^2$ and comment on extensions to other losses, e.g. the cross entropy, in App. C. To this end, consider first the prior of the output of a finite DNN. Using standard manipulations (see App. A), it can be expressed as follows

$$P_0\left(\vec{f}\right) \propto \int_{\mathbb{R}^{n+1}} d\vec{t}\exp\left(-\sum_{\mu=1}^{n+1} it_\mu f_\mu + \sum_{r=2}^\infty \frac{1}{r!}\sum_{\mu_1,\ldots,\mu_r=1}^{n+1} \kappa_{\mu_1,\ldots,\mu_r} it_{\mu_1}\cdots it_{\mu_r}\right) \tag{4}$$

where $\kappa_{\mu_1,\ldots,\mu_r}$ is the $r$'th multivariate cumulant of $P_0(\vec{f})$ [31]. The second term in the exponent is the *cumulant generating function (CGF)*, denoted by $\mathcal{C}(\vec{t})$, corresponding to $P_0$. As discussed in

App. B and Ref. [32], for standard initialization protocols the $r$'th cumulant will scale as $1/C^{(r/2-1)}$, where $C$ controls the over-parameterization, e.g. number of neurons / channels in each layer for FCNs / CNNs, respectively. The second ($r = 2$) cumulant which is $C$-independent, describes the NNGP kernel of the finite DNN and is denoted by $K(\mathbf{x}_{\mu_1}, \mathbf{x}_{\mu_2}) = \kappa_{\mu_1,\mu_2}$.

Consider first the case of $C \to \infty$ [24, 30, 33] where all $r > 2$ cumulants vanish. Here one can explicitly perform the integration in Eq. 4 to obtain the standard GP prior $P_0\left(\vec{f}\right) \propto \exp\left(-\frac{1}{2}\sum_{\mu_1,\mu_2=1}^{n+1}\kappa_{\mu_1,\mu_2}f_{\mu_1}f_{\mu_2}\right)$. Plugging this prior into Eq. 2 with MSE loss, one recovers standard GP regression formulas [37]. In particular, the predictive mean at $\mathbf{x}_*$ is: $\langle f(\mathbf{x}_*)\rangle = \sum_{\mu,\nu=1}^{n} K_\mu^* \tilde{K}_{\mu\nu}^{-1} g_\nu$ where $K_\mu^* = K(\mathbf{x}_*, \mathbf{x}_\mu)$ and $\tilde{K}_{\mu\nu} = K(\mathbf{x}_\mu, \mathbf{x}_\nu) + \sigma^2\delta_{\mu\nu}$. Another set of quantities we shall find useful are the *discrepancies in GP prediction*, which for the training set read

$$\forall \mu \in \{1, \ldots, n\}: \quad \langle\hat{\delta}g_\mu\rangle \equiv g_\mu - \langle f(\mathbf{x}_\mu)\rangle = g_\mu - \sum_{\nu,\nu'=1}^{n} K_{\mu\nu'}\tilde{K}_{\nu',\nu}^{-1}g_\nu \tag{5}$$

**Saddle-point approximation for the mean predictor.** For a DNN with finite $C$, the prior $P_0(\vec{f})$ will no longer be Gaussian and cumulants with $r > 2$ would contribute. This renders the partition function in Eq. 2 intractable and so some approximation is needed to make progress. To this end we note that $f$ can be integrated out (see App. A.1) to yield a partition function of the form

$$Z\left(\vec{J}\right) \propto \int_{\mathbb{R}^n} dt_1 \cdots dt_n e^{-\mathcal{S}(\vec{t},\vec{J})} \tag{6}$$

where $\mathcal{S}(\vec{t}, \vec{J})$ is the action whose exact form is given in Eq. A.14. Interestingly, the $it_\mu$ variables appearing above are closely related to the discrepancies $\hat{\delta}g_\mu$, in particular $\langle it_\mu\rangle = \langle\hat{\delta}g_\mu\rangle/\sigma^2$.

To proceed analytically we adopt the *saddle-point (SP) approximation* [12] which often relies on having a partition function of the form $Z = \int dt e^{-n\mathcal{S}(t)}$ where $n$ is a large number and $\mathcal{S}$ is $O(1)$. In our settings we cannot simply extract such a large factor from the action and make it $O(1)$. Nonetheless, we argue in App. A.4 that the saddle-point is still a good approximation for large $n$ (training set size). This relies on the fact that the non-linear terms in the action comprise of a sum of many $it_\mu$'s. Given that this sum is dominated by collective effects coming from all data points (as opposed to only a selected few), expanding $\mathcal{S}(\vec{t}, \vec{J})$ around the saddle-point yields terms with increasingly negative powers of $n$.

For the training points $\mu \in \{1, \ldots, n\}$, taking the saddle-point approximation amounts to setting $\partial_{it_\mu}\mathcal{S}\left(\vec{t}, \vec{J}\right)\big|_{\vec{J}=\vec{0}} = 0$. This yields a set of equations that has precisely the form of Eq. 5, but where the target is shifted as $g_\nu \to g_\nu - \Delta g_\nu$ and the target shift is determined self-consistently by

$$\Delta g_\nu = \sum_{r=3}^{\infty} \frac{1}{(r-1)!} \sum_{\mu_1,\ldots,\mu_{r-1}=1}^{n} \kappa_{\nu,\mu_1,\ldots,\mu_{r-1}} \left\langle\sigma^{-2}\hat{\delta}g_{\mu_1}\right\rangle \cdots \left\langle\sigma^{-2}\hat{\delta}g_{\mu_{r-1}}\right\rangle \tag{7}$$

Equation 7 is thus an implicit equation for $\Delta g_\nu$ involving all training points, and it holds for the training set and the test point $\nu \in \{1, \ldots, n+1\}$. Once solved, either analytically or numerically, one calculates the predictions on the test point via

$$\langle f_*\rangle = \Delta g_* + \sum_{\mu,\nu=1}^{n} K_\mu^* \tilde{K}_{\mu\nu}^{-1}(g_\nu - \Delta g_\nu) \tag{8}$$

Equation 5 with $g_\nu \to g_\nu - \Delta g_\nu$ along with Eqs. 7 and 8 are the first main result of this paper. Viewed as an algorithm, the procedure to predict the finite DNN's output on a test point $\mathbf{x}_*$ is as follows: we shift the target in Eq. 5 as $g \to g - \Delta g$ with $\Delta g$ as in Eq. 7, arriving at a closed equation for the average discrepancies $\langle\hat{\delta}g_\mu\rangle$ on the training set. For some models, the cumulants $\kappa_{\nu,\mu_2,\ldots,\mu_r}$ can be computed for any order $r$ and it can be possible to sum the entire series, while for other models several leading cumulants might already give a reasonable approximation due to their $1/C^{r/2-1}$

scaling. The resulting coupled non-linear equations can then be solved numerically, to obtain $\Delta g_\mu$ from which predictions on the test point are calculated using Eq. 8.

Notwithstanding, solving such equations analytically is challenging and one of our main goals here is to provide concrete analytical insights. Thus, in §4.1.2 we propose an additional approximation wherein to leading order we replace all summations over data-points with integrals over the measure from which the data-set is drawn. This approximation, taken in some cases beyond leading order as in Ref. [11], will yield analytically tractable equations which we solve for two simple toy models, one of a linear CNN and the other of a non-linear FCN.

**Saddle-point plus Gaussian fluctuations for the posterior covariance.** The SP approximation can be extended to compute the predictor variance by expanding the action $\mathcal{S}$ to quadratic order in the deviation from the SP value $\delta t_\mu \equiv t_\mu - t_\mu^{\mathrm{SP}}$ (see App. A.3). Due to the saddle-point being an extremum this leads to $\mathcal{S} \approx \mathcal{S}_{\mathrm{SP}} + \frac{1}{2}\sum_{\mu,\nu}\delta t_\mu A_{\mu\nu}^{-1}\delta t_\nu$ This leaves the previous SP approximation for the posterior mean on the training set unaffected (since the mean and maximizer of a Gaussian coincide), but is necessary to get sensible results for the posterior covariance. Using the standard Gaussian integration formula, one finds that $A_{\mu\nu}$ is the covariance matrix of $it_\mu$. Performing such an expansion one finds

$$A_{\mu\nu}^{-1} = -\left(\sigma^2\delta_{\mu\nu} + K_{\mu\nu} + \Delta K_{\mu\nu}\right) \tag{9}$$
$$\Delta K_{\mu\nu} = \partial_{it_\mu}\partial_{it_\nu}\tilde{\mathcal{C}}\left(it_1,\ldots,it_n\right)$$

where the $it$'s on the r.h.s. are those of the saddle-point and $\tilde{\mathcal{C}}$ is the CGF $\mathcal{C}$ without the second cumulant (see App. A.1). This gives an expression for the posterior covariance matrix on the training set:

$$\Sigma_{\mu\nu} \equiv \langle f_\mu f_\nu\rangle - \langle f_\mu\rangle\langle f_\nu\rangle = -\sigma^4\left[\sigma^2 I + K + \Delta K\right]_{\mu\nu}^{-1} + \sigma^2\delta_{\mu\nu} \tag{10}$$

where the r.h.s. coincides with the posterior covariance of a GP with a kernel equal to $K + \Delta K$ [37]. The variance on the test point is given by (repeating indices are summed over the training set)

$$\Sigma_{**} = K_{**} - K_\mu^* A_{\mu\nu}^{-1} K_\nu^* + \left\langle\partial_{it_*}^2\tilde{\mathcal{C}}|_{it_*=0}\right\rangle + 2\left(\langle\Delta g_* K_\mu^* it_\mu\rangle - \langle\Delta g_*\rangle\langle K_\mu^* it_\mu\rangle\right) + \mathrm{Var}\left(\Delta g_*\right) \tag{11}$$

where here $\Delta g_*$ is as in Eq. 7 but where the $\left\langle\sigma^{-2}\hat{\delta}g_\mu\right\rangle$'s are replaced the $it_\mu$'s that have Gaussian fluctuations. The first two terms in Eq. 11 yield the standard result for the GP posterior covariance matrix on a test point [37], for the case of $\Delta K = 0$ (see Eq. 9). The rest of the terms can be evaluated by the SP plus Gaussian fluctuations approximation, where the details would depend on the model at hand.

# 4 Experiments

## 4.1 The two layer linear CNN

### 4.1.1 Setting of the model and its properties

Here we define a teacher-student toy model showing several qualitative real-world aspects of feature learning and analyze it via our self-consistent shifted target approach. Concretely, we consider the simplest student CNN $f(\mathbf{x})$, having one hidden layer with linear activation, and a corresponding teacher CNN, $g(\mathbf{x})$

$$f\left(\mathbf{x}\right) = \sum_{i=1}^{N}\sum_{c=1}^{C}a_{i,c}\mathbf{w}_c\cdot\tilde{\mathbf{x}}_i \qquad g\left(\mathbf{x}\right) = \sum_{i=1}^{N}\sum_{c=1}^{C^*}a_{i,c}^*\mathbf{w}_c^*\cdot\tilde{\mathbf{x}}_i \tag{12}$$

This describes a CNN that performs 1-dimensional convolution where the convolutional weights for each channel are $\mathbf{w}_c \in \mathbb{R}^S$. These are dotted with a convolutional window of the input $\tilde{\mathbf{x}}_i = \left(x_{S(i-1)+1},\ldots,x_{S\cdot i}\right)^\mathsf{T} \in \mathbb{R}^S$ and there are no overlaps between them so that $\mathbf{x} = \left(x_1,\ldots,x_{N\cdot S}\right)^\mathsf{T} = \left(\tilde{\mathbf{x}}_1,\ldots,\tilde{\mathbf{x}}_N\right)^\mathsf{T} \in \mathbb{R}^{N\cdot S}$. Namely, the input dimension is $d = NS$, where $N$ is

the number of (non-overlapping) convolutional windows, $S$ is the stride of the conv-kernel and it is also the length of the conv-kernel, hence there is no overlap between the strides. The inputs $\mathbf{x}$ are sampled from $\mathcal{N}(0, I_d)$.

Despite its simplicity, this model distils several key differences between feature learning models and lazy learning or GP models. Due to the lack of pooling layers, the GP associated with the student fails to take advantage of the weight sharing property of the underlying CNN [36]. In fact, here it coincides with a GP of a fully-connected DNN (see Eq. 13) which is quite inappropriate for the task. We thus expect that the finite network will have good performance already for $n = C^*(N + S)$ whereas the GP will need $n$ of order of the dimension $(NS)$ to learn well [11]. Thus, for $N + S \ll NS$ there should be a broad regime in the value of $n$ where the finite network substantially outperforms the corresponding GP. We later show (§4.1.4) that this performance boost over GP is due to feature learning, as one may expect.

Conveniently, the cumulants of the student DNN of any order can be worked out exactly. Assuming $\gamma$ and $\sigma^2$ of the noisy GD training are chosen such that[1] $a_{i,c} \sim \mathcal{N}\left(0, \sigma_a^2/CN\right), \quad \mathbf{w}_c \sim \mathcal{N}\left(\mathbf{0}, \frac{\sigma_w^2}{S} I_S\right)$ (and similarly for the teacher DNN) the covariance function for the associated GP reads

$$K\left(\mathbf{x}, \mathbf{x}'\right) = \frac{\sigma_a^2 \sigma_w^2}{NS} \sum_{i=1}^{N} \tilde{\mathbf{x}}_i^\mathsf{T} \tilde{\mathbf{x}}_i' = \frac{\sigma_a^2 \sigma_w^2}{NS} \mathbf{x}^\mathsf{T} \mathbf{x}' \tag{13}$$

Denoting $\lambda := \frac{\sigma_a^2}{N} \frac{\sigma_w^2}{S}$, the even cumulant of arbitrary order $2m$ is (see App. F):

$$\kappa_{2m}\left(\mathbf{x}_1, \ldots, \mathbf{x}_{2m}\right) = \frac{\lambda^m}{C^{m-1}} \sum_{i_1, \ldots, i_m=1}^{N} \left(\bullet_{i_1}, \bullet_{i_2}\right) \cdots \left(\bullet_{i_{m-2}}, \bullet_{i_{m-1}}\right) \left(\bullet_{i_{m-1}}, \bullet_{i_m}\right) \cdots [(2m-1)!] \tag{14}$$

while all odd cumulants vanish due to the sign flip symmetry of the last layer. In this notation, we mean that the $\bullet$'s stand for integers in $\{1, \ldots, 2m\}$ and e.g. $\left(1_{i_1}, 2_{i_2}\right) \equiv \left(\tilde{\mathbf{x}}_{i_1}^1 \cdot \tilde{\mathbf{x}}_{i_2}^2\right)$ and the bracket notation $[(2m-1)!]$ stands for the number of ways to pair the integers $\{1, ..., 2m\}$ into the above form. This result can then be plugged in 7 to obtain the self-consistent (saddle-point) equations on the training set. See App. A.4 for a convergence criterion for the saddle-point, supporting its application here.

### 4.1.2 Self-consistent equation in the limit of a large training set

In §3 our description of the self-consistent equations was for a finite and fixed training set. Further analytical insight can be gained if we consider the limit of a large training set, known in the GP literature as the Equivalent Kernel (EK) limit [37, 42]. For a short review of this topic, see App. D. In essence, in the EK limit we replace the discrete sums over a specific draw of training set, as in Eqs. 5, 7, 8, with integrals over the entire input distribution $\mu(\mathbf{x})$. Given a kernel that admits a spectral decomposition in terms of its eigenvalues and eigenfunctions: $K\left(\mathbf{x}, \mathbf{x}'\right) = \sum_s \lambda_s \psi_s\left(\mathbf{x}\right) \psi_s\left(\mathbf{x}'\right)$, the standard result for the GP posterior mean at a test point is approximated by [37]

$$\langle f\left(\mathbf{x}_*\right)\rangle = \int d\mu\left(\mathbf{x}\right) h\left(\mathbf{x}_*, \mathbf{x}\right) g\left(\mathbf{x}\right); \qquad h\left(\mathbf{x}_*, \mathbf{x}\right) = \sum_s \frac{\lambda_s}{\lambda_s + \sigma^2/n} \psi_s\left(\mathbf{x}_*\right) \psi_s\left(\mathbf{x}\right) \tag{15}$$

This has several advantages, already at the level of GP analysis. From a theoretical point of view, the integral expressions retain the symmetries of the kernel $K(\mathbf{x}, \mathbf{x}')$ unlike the discrete sums that ruin these symmetries. Also, Eq. 15 does not involve computing the inverse matrix $\tilde{K}^{-1}$ which is costly for large matrices.

In the context of our theory, the EK limit allows for a derivation of a simple analytical form for the self-consistent equations. As shown in App. E.1, in our toy CNN we can write both $\Delta g$ and $\hat{\delta}g$ in terms of the target $g$ using corresponding proportionality factors. Thus the self-consistent equations can be reduced to a single equation governing the proportionality factor $\alpha$ between $\hat{\delta}g$ and $g$ ($\hat{\delta}g = \alpha g$). Notice that $\alpha$ itself is governed by an equation that is non-linear in $g$, which means that $\Delta g$ and $\hat{\delta}g$ do not scale linearly with $g$, only that we can trade the function-valued self-consistent

---

[1] Generically this requires $C$ dependent and layer dependent weight decay.

equation for a scalar-valued one. Thus, starting from the general self-consistent equations, 5, 7, 8, taking their EK limit, and plugging in the general cumulant for our toy model (14) we arrive at the following equation for $\alpha$

$$\alpha = \frac{\sigma^2/n}{\lambda + \sigma^2/n} + \frac{(1-q)\lambda}{\lambda + \sigma^2/n} + \left( q\frac{\lambda}{\lambda + \sigma^2/n} - 1 \right) \frac{\lambda^2}{C} \left( \frac{\alpha}{\sigma^2/n} \right)^3 \left[ 1 - \frac{\lambda}{C} \left( \frac{\alpha}{\sigma^2/n} \right)^2 \right]^{-1} \quad (16)$$

Setting for simplicity $\sigma_a^2 = 1 = \sigma_w^2$ we have $\lambda = 1/(NS)$ and we also introduced the constant $q \equiv \lambda^{-1}(1 - \hat{\alpha}_{\mathrm{GP}})(\lambda + \sigma^2/n)$ where $\hat{\alpha}_{\mathrm{GP}}$ is computed using the empirical GP predictions on either the training set or test set: $\hat{\alpha}_{\mathrm{GP}} \equiv 1 - \left( \sum_\mu f_\mu^{\mathrm{GP}} g_\mu \right) / \left( \sum_\mu g_\mu^2 \right)$, or analytically in the perturbation theory approach developed in [11]. The quantity $q$ has an interpretation as a $1/n$ correction to the EK approximation [11] but here can be considered as a fitting parameter. It is non-negative and is typically $O(1)$; for more details and analytical estimates of $q$ see App. E.2.

Equation 16 is the second main analytical result of this work. It simplifies the highly non-linear inference problem to a single equation that embodies strong non-linear finite-DNN effect and feature learning (see also §4.1.4). In practice, to compute $\alpha_{\mathrm{test}}$ we numerically solve 16 using $q_{\mathrm{train}}$ for the training set to get $\alpha_{\mathrm{train}}$, and then set $\alpha = \alpha_{\mathrm{train}}$ in the r.h.s. of 16 but use $q = q_{\mathrm{test}}$. Equation 16 can also be used to bound $\alpha$ analytically on both the training set and test point, given the reasonable assumption that $\alpha$ changes continuously with $C$. Indeed, at large $C$ the pole in this equation lays at $\alpha_{\mathrm{pole}} = (\sigma^2/n)(C/\lambda)^{1/2} \gg 1$ whereas $\alpha \approx \alpha_{\mathrm{GP}} < \alpha_{\mathrm{pole}}$. As $C$ diminishes, continuity implies that $\alpha$ must remain smaller than $\alpha_{\mathrm{pole}}$. The latter decays as $\sigma^2\sqrt{CNS}/n$ implying that the amount of data required for good performance scales as $\sqrt{CNS}$ rather than as $NS$ in the GP case.

### 4.1.3 Numerical verification

In this section we numerically verify the predictions of the self-consistent theory of Sec. §4.1.2, by training linear shallow student CNNs on a teacher with $C^* = 1$ as in Eq. 12, using noisy gradients as in Eq. 1, and averaging their outputs across noise realizations and across dynamics after reaching equilibrium.

For simplicity we used $N = S$ and $n \in \{62, 200, 650\}$, $S \in \{15, 30, 60\}$ so that $n \propto S^{1.7}$. The latter scaling places us in the poorly performing regime of the associated GP while allowing good performance of the CNN. Indeed, as aforementioned, the GP here requires $n$ on the scale of $\lambda^{-1} = NS = O(S^2)$ for good performance [11], while the CNN requires $n$ on the scale of the number of parameters ($C(N + S) = O(S)$).

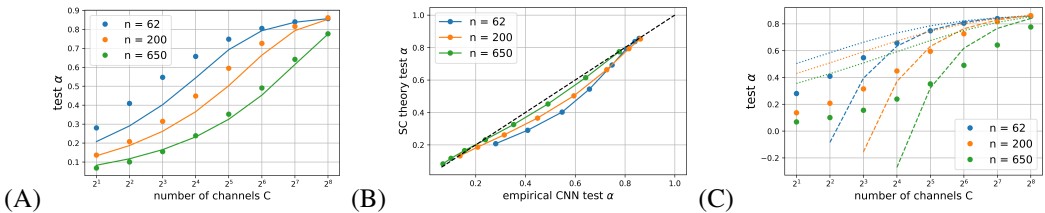

Figure 1: **(A)** The CNNs' cosine distance $\alpha$, defined by $\langle f \rangle = (1 - \alpha)g$ between the ensemble-averaged prediction $\langle f \rangle$ and ground truth $g$ plotted vs. number of channels $C$ for the test set (for the train set, see App. H.1). As $n$ increases, the solution of the self-consistent equation 16 (solid line) yields an increasingly accurate prediction of these empirical values (dots). **(B)** Same data as in (A), presented as empirical $\alpha$ vs. predicted $\alpha$. As $n$ grows, the two converge to the identity line (dashed black line). Solid lines connecting the dots here are merely for visualization purposes. **(C)** The theoretical predictions of the self-consistent theory but using only the fourth cumulant rather than all cumulants (dotted lines), and the predictions of perturbation theory to order $1/C$ (dashed lines, truncated to avoid large negative values in the figure).

The results are shown in Fig. 1 where we compare the theoretical predictions given by the solution of the self-consistent equation (16) to the empirical values of $\alpha$ obtained by training actual CNNs and averaging their outputs across the ensemble. We can see that as $n$ increases, the predictions of our theory match the empirical data more closely (panels A and B), as can be expected from our SP approximation which is valid for large $n$.

In panel C, we compare the empirical values to two alternative theoretical predictions: (i) our self-consistent theory but using only the fourth cumulant, equivalent to equation Eq. 16 while ignoring the $[\cdots]^{-1}$ term, rather than summing the geometric series of cumulants of all orders which gives rise to this term (dotted lines); (ii) A purely perturbative theory (see e.g. [32, 44, 39]) keeping only a leading $O(1/C)$ term on top of the GP predictions (dashed lines, truncated to avoid large negative values in the figure). We see that the predictions of our full self-consistent theory (panels A and B) are clearly better than those of either (i) or (ii) above, at least for sufficiently small $C$ or sufficiently large $n$: e.g. the predictions of (ii) for $n \in \{62, 200\}$ match the empirical data nicely for $C \geq 2^4$ but diverge rapidly for smaller values of $C$, as one would expect. The predictions of (i) don't diverge so rapidly from the empirical data and always gives $0 \leq \alpha \leq 1$, but are also quite poor for $C \lesssim 2^5$. This shows that in these settings knowing all the cumulants provides a much more accurate theory. For sufficiently large $C$, all predictions, including our own theory, coincide as they all converge to the GP limit.

### 4.1.4 Feature learning phase transition in the CNN model

At this point there is evidence that our self-consistent shifted target approach works well within the feature learning regime of the toy model. Indeed, GP is sub-optimal here, since it does not represent the CNN's weight sharing present in the teacher network. Weight sharing is intimately tied with feature learning in the first layer, since it aggregates the information coming from all convolutional windows to refine a single set of repeating convolution-filters. Empirically, we observed a large performance gap of finite-$C$ CNNs over the infinite-$C$ (GP) limit, which was also observed previously in more realistic settings [23, 15, 36]. Taken together with the existence of a clear feature in the teacher, a natural explanation for this performance gap is that feature learning, which is completely absent in GPs, plays a major role in the behavior of finite $C$ CNNs.

To analyze this we wish to track how the feature of the teacher $\mathbf{w}^*$ is reflected in the student network's first layer weights $\mathbf{w}_c$ across training time (after reaching equilibrium) and across training realizations. However, as our formalism deals with *ensembles of DNNs*, computing averages of $\mathbf{w}_c$ with respect to these ensembles would simply give zero. Indeed, the chance of a DNN with specific parameters $\theta = \{a_{i,c}, \mathbf{w}_c\}$ appearing is the same as that of $-\theta$. Consequently, to detect feature learning the first reasonable object to examine is the empirical covariance matrix $\Sigma_W \equiv \frac{S}{C} WW^\mathsf{T}$, where the matrix $W \in \mathbb{R}^{S \times C}$ has $\mathbf{w}_c$ as its $c$'th column. This $\Sigma_W$ is invariant under such a change of signs and provides important information on the statistics of $\mathbf{w}_c$.

As shown in App. G, using our field-theory or function-space formulation, we find that to leading order in $1/C$ the ensemble average of the empirical covariance matrix, for a teacher with a single feature $\mathbf{w}^*$, is

$$\langle [\Sigma_W]_{ss'} \rangle = \left(1 + \left(\frac{1}{\lambda} + \frac{n}{\sigma^2}\right)^{-1}\right)\delta_{ss'} + \frac{2}{C}\frac{\lambda}{(\lambda + \sigma^2/n)^2}w_s^* w_{s'}^* + O(1/C^2) \qquad (17)$$

A first conclusion that could be drawn here, is that given access to an ensemble of such trained CNNs, feature learning happens for any finite $C$ as a statistical property. We turn to discuss the more common setting where one wishes to use the features learned by a specific randomly chosen CNN from this ensemble.

To this end, we follow Ref. [29] and model $\Sigma_W$ as a Wishart matrix with a rank-one perturbation. The variance of the matrix and details of the rank one perturbation are then determined by the above equation. Consequently the eigenvalue distribution is expected to follow a spiked Marchenko-Pastur (MP) model, which was studied extensively in [7]. To test this modeling assumption, for each snapshot of training time (after reaching equilibrium) and noise realization we compute $\Sigma_W$'s eigenvalues and aggregate these across the ensemble. In Fig. 2 we plot the resulting empirical spectral distribution for varying values of $C$ while keeping $S$ fixed. Note that, differently from the usual spiked-MP model, varying $C$ here changes both the distribution of the MP bulk (which is determined by the ratio $S/C$) as well as the strength of the low-rank perturbation.

Our main finding is a phase transition (analogous to [7]) between two regimes which becomes sharp as one takes $n, S \to \infty$. In the regime of large $C$ the eigenvalue distribution of $\Sigma_W$ is indistinguishable from the MP distribution, whereas in the regime of small $C$ an outlier eigenvalue $\lambda_m$ departs from the support of the bulk MP distribution and the associated top eigenvector has a non-zero overlap

with $\mathbf{w}^*$, see Fig. 2. We refer to the latter as the feature-learning regime, since the feature $\mathbf{w}^*$ is manifested in the spectrum of the students weights, whereas the former is the non-feature learning regime. We use the quantity $\mathcal{Q} \equiv \mathbf{w}^{*\mathsf{T}} \Sigma_W \mathbf{w}^*$ as a surrogate for $\lambda_m$, as it is valid on both sides of the transition. Having established the correspondence to the MP plus low rank model, we can use the results of [7] to find the exact location of the phase transition, which occurs at the critical value $C_{\text{crit}}$ given by

$$C_{\text{crit}} = \frac{4}{S\left(S^{-1} + (\sigma^2/n) S\right)^4} \left(1 + \left(S^2 + \frac{n}{\sigma^2}\right)^{-1}\right) + O\left(1 + \left(\frac{1}{\lambda} + \frac{n}{\sigma^2}\right)^{-1}\right) \quad (18)$$

where we assumed for simplicity $N = S$ so that $\lambda = S^{-2}$.

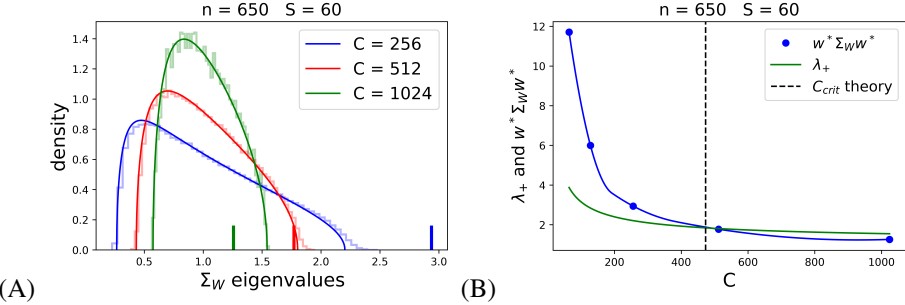

$\qquad$ (A) $\hspace{7cm}$ (B)

Figure 2: **(A)** Aggregated histograms of $\Sigma_W$ eigenvalues where $\Sigma_W = \frac{S}{C} WW^\mathsf{T}$ is the normalized empirical covariance matrix of the hidden layer weights during training. Different colors indicate varying number of channels, $C$. Solid smooth lines indicate the corresponding Marchenko-Pastur (MP) distributions with support on $[\lambda_-, \lambda_+]$ where: $\lambda_\pm = \left(1 \pm \sqrt{S/C}\right)^2$. The quantity $\mathcal{Q} \equiv \mathbf{w}^{*\mathsf{T}} \Sigma_W \mathbf{w}^*$, which correlates with the strength of rank-1 component of the feature $\mathbf{w}^*$, is represented by thick short bars. For large $C$, $\mathcal{Q}$ remains within the MP bulk whereas for small $C$ it pops out. **(B)** The theoretical $\lambda_+$ curve and interpolated curve of $\mathcal{Q}$ intersect very close to the theoretically predicted value given in Eq. 18, here given by $C_{\text{crit}} = 473$ (dashed vertical line).

### 4.2 Two-layer FCN with average pooling and quadratic activations

Another setting where GPs are expected to under-perform finite-DNNs is the case of quadratic fully connected teacher and student DNNs where the teacher is rank-1, also known as the phase retrieval problem [28]. Here we consider some positive target of the form $g(\mathbf{x}) = (\mathbf{w}_* \cdot \mathbf{x})^2 - \sigma_w^2 \|\mathbf{x}\|^2$ where $\mathbf{w}_*, \mathbf{x} \in \mathbb{R}^d$ and a student DNN given by $f(\mathbf{x}) = \sum_{m=1}^M (\mathbf{w}_m \cdot \mathbf{x})^2 - \sigma_w^2 \|\mathbf{x}\|^2$. We consider training this DNN on $n$ train points $\{\mathbf{x}_\mu\}_{\mu=1}^n$ using noisy GD training with weight decay $\gamma = 2M\sigma^2/\sigma_w^2$.

Similarly to the previous toy model, here too the GP associated with the student at large $M$ (and finite $\sigma^2$) overlooks a qualitative feature of the finite DNN — the fact that the first term in $f(\mathbf{x})$ is non-negative. Interestingly, this feature provides a strong performance boost [28] in the $\sigma^2 \to 0$ limit compared to the associated GP. Namely the DNN, even at large $M$, performs well for $n > 2d$ [28] whereas the associated GP is expected to work well only for $n = O(d^2)$ [11].

We wish to solve for the predictions of this model with our self-consistent GP based approach. As shown in App. I, the cumulants of this model can be obtained from the following cumulant generating function

$$\mathcal{C}(t_1, ..., t_{n+1}) = -\frac{M}{2} \operatorname{Tr}\left(\log\left[I - 2M^{-1}\sigma_w^2 \sum_\mu it_\mu \mathbf{x}_\mu \mathbf{x}_\mu^\mathsf{T}\right]\right) - \sum_{\mu=1}^{n+1} it_\mu \sigma_w^2 \|\mathbf{x}_\mu\|^2 \quad (19)$$

The associated GP kernel is given by $K(\mathbf{x}_\mu, \mathbf{x}_\nu) = 2M^{-1}\sigma_w^4 (\mathbf{x}_\mu \cdot \mathbf{x}_\nu)^2$. Following this, the target shift equation, at the saddle-point level, appears as

$$\Delta g_\nu = -\sum_\mu K(\mathbf{x}_\nu, \mathbf{x}_\mu) \frac{\hat{\delta} g_\mu}{\sigma^2} + \sigma_w^2 \mathbf{x}_\nu^\mathsf{T} \left[I - 2M^{-1}\sigma_w^2 \sum_\mu \frac{\hat{\delta} g_\mu}{\sigma^2} \mathbf{x}_\mu \mathbf{x}_\mu^\mathsf{T}\right]^{-1} \mathbf{x}_\nu - \sigma_w^2 \|\mathbf{x}_\nu\|^2 \quad (20)$$

In App. I, we solve these equations numerically for $\sigma^2 = 10^{-5}$ and show that our approach captures the correct $n = 2d$ threshold value. An analytic solution of these equations at low $\sigma^2$ using EK or other continuum approximations is left for future work (see Refs. [11, 8, 9] for potential approaches). As a first step towards this goal, in App. I we consider the simpler case of $\sigma^2 = 1$ and derive the asymptotics of the learning curves which deviate strongly from those of GP for $M \ll d$.

### 4.3 Validity of the saddle-point approximation in realistic settings

In this subsection we complement the above results on controlled toy models by some preliminary results in more realistic settings. The approximation underlying our analysis was that the fluctuations of the integrand in the partition function of Eq. 6 are near-Gaussian. A strong indication of how adequate this approximation is in real-world settings can be given by performing normality tests on an ensemble of the outputs of trained DNNs on the training set. Indeed, this is the ensemble described by our partition function. In App. H.3 we report some preliminary results we obtained by training the Myrtle-5 deep CNN [41] using our protocol on tiny subsets ($n \in \{16, 32, 64\}$) of CIFAR-10, and inspecting the normality of output fluctuations, as measured by the 4th cumulant of the outputs. In the experiments with small enough $C$, although the CNN predictions already deviated strongly from those at infinite width, we haven't found a measurable signal of non-normality, thus our Gaussian fluctuation assumption is consistent with these findings.

## 5 Discussion

In this work we presented a correspondence between ensembles of finite DNNs trained with noisy gradients and GPs trained on a shifted target. The shift in the target can be found by solving a set of self-consistent equations for which we give a general form. We found explicit expressions for these equations for the case of a 2-layer linear CNN and a non-linear FCN, and solved them analytically and numerically. For the former model, we performed numerical experiments on CNNs that agree well with our theory both in the GP regime *and well away from it*, i.e. for small number of channels $C$, thus accounting for strong finite $C$ effects. For the latter model, the numerical solution of these equations captures a remarkable and subtle effect in these DNNs which the GP approach completely overlooks — the $n = 2d$ threshold value.

Considering feature learning in the CNN model, we found that averaging over ensembles of such networks always leads to a form of feature learning. Namely, the teacher always leaves a signature on the statistics of the student's weights. However, feature learning is usually considered at the level of a single DNN instance rather than an ensemble of DNNs. Focusing on this case, we show numerically that the eigenvalues of $\Sigma_W$, the student hidden weights covariance matrix, follow a Marchenko–Pastur distribution plus a rank-1 perturbation. We then use our approach to derive the critical number of channels $C_{\text{crit}}$ below which the student is in a feature learning regime.

There are many directions for future research. Our toy models were chosen to be as simple as possible in order to demonstrate the essence of our theory on problems where lazy learning grossly under-performs finite-DNNs. Even within this setting, various extensions are interesting to consider such as adding more features to the teacher CNN (e.g. biases or a subset of linear functions which are more favorable), studying linear CNNs with overlapping convolutional windows, or deeper linear CNNs. As for non-linear CNNs, we believe it is possible to find the exact cumulants of any order for a variety of toy CNNs involving, for example, quadratic activation functions. For other cases it may be useful to develop methods for characterizing and approximating the cumulants.

More generally, we advocated here a physics-style methodology using approximations, self-consistency checks, and experimental tests. As DNNs are very complex experimental systems, we believe this mode of research is both appropriate and necessary. Nonetheless we hope the insights gained by our approach would help generate a richer and more relevant set of toy models on which mathematical proofs could be made.

## Acknowledgements

We would like to thank Haim Sompolinsky for useful discussions and Jonathan Kadmon, Oded Ben-David, Dar Gilboa and Inbar Seroussi for comments on the manuscript. GN was partially

supported by the Gatsby Charitable Foundation, the Swartz Foundation, the National Institutes of Health (Grant No. 1U19NS104653) and the MAFAT Center for Deep Learning.

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
