# A    Derivation of the target shift equations

## A.1    The partition function in terms of dual variables

Consider the general setting of Bayesian inference with Gaussian measurement noise (or equivalently a DNN trained with MSE loss, weight decay, and white noise added to the gradients). Let $\{\mathbf{x}_\mu, g_\mu\}_{\mu=1}^n$ denote the inputs and targets on the training set and let $\mathbf{x}_* \equiv \mathbf{x}_{n+1}$ be the test point. Denote the prior (or equivalently the equilibrium distribution of a DNN trained with no data) by $P_0(\vec{f})$ where $f_\mu = f(\mathbf{x}_\mu) \in \mathbb{R}$ is the output of the model, and $\vec{f} \equiv (f_1, \ldots, f_n, f_{n+1}) \in \mathbb{R}^{n+1}$. The model's predictions (or equivalently the ensemble averaged DNN output) on the point $\mathbf{x}_\mu$ can be obtained by

$$\forall \mu \in \{1, \ldots, n+1\}: \qquad \langle f_\mu \rangle = \partial_{J_\mu} \log Z\left(\vec{J}\right)\Big|_{\vec{J}=\vec{0}} \tag{A.1}$$

with the following partition function

$$Z\left(\vec{J}\right) = \int_{\mathbb{R}^{n+1}} d\vec{f} P_0\left(\vec{f}\right) \exp\left(-\frac{1}{2\sigma^2}\sum_{\mu=1}^n (f_\mu - g_\mu)^2 + \sum_\mu J_\mu f_\mu\right) \tag{A.2}$$

where unless explicitly written otherwise, summations over $\mu$ run from 1 to $n+1$ (i.e. include the test point). Here we commit to the MSE loss which facilitates the derivation, and in App. C we give an alternative derivation that may also be applied to other losses such as cross-entropy. Our goal in this appendix is to establish that the target shift equations are in fact saddle-point equations of the partition function A.2 following some transformations on the variables of integration. To this end, consider the cumulant generating function of the prior $P_0\left(\vec{f}\right)$, given by the natural logarithm of the characteristic function[2]

$$\mathcal{C}\left(\vec{t}\right) = \log\left(\int_{\mathbb{R}^{n+1}} d\vec{f} e^{i\sum_\mu t_\mu f_\mu} P_0\left(\vec{f}\right)\right) \tag{A.3}$$

or expressed via the cumulant tensors:

$$\mathcal{C}\left(\vec{t}\right) = \sum_{r=2}^\infty \frac{1}{r!} \sum_{\mu_1,\ldots,\mu_r=1}^{n+1} \kappa_{\mu_1,\ldots,\mu_r} it_{\mu_1} \cdots it_{\mu_r} \tag{A.4}$$

where the sum over the cumulant tensors $\kappa_{\mu_1,\ldots,\mu_r}$ does not include $r=1$ since our DNN priors are assumed to have zero mean. Notably one can re-express $P_0\left(\vec{f}\right)$ as the inverse Fourier transform of $e^{\mathcal{C}(\vec{t})}$:

$$P_0\left(\vec{f}\right) \propto \int_{\mathbb{R}^{n+1}} d\vec{t} \exp\left(-i\sum_\mu t_\mu f_\mu + \mathcal{C}\left(\vec{t}\right)\right) \tag{A.5}$$

Plugging this in Eq. A.2 we obtain

$$Z\left(\vec{J}\right) \propto \int_{\mathbb{R}^{n+1}} \int_{\mathbb{R}^{n+1}} d\vec{f} d\vec{t} \exp\left(-\frac{1}{2\sigma^2}\sum_{\mu=1}^n (f_\mu - g_\mu)^2 + \sum_\mu (J_\mu - it_\mu) f_\mu + \mathcal{C}\left(\vec{t}\right)\right) \tag{A.6}$$

where for clarity we do not keep track of multiplicative $\pi$ factors that have no effect on moments of $f_\mu$. As the term in the exponent (the action) is quadratic in $\{f_\mu\}_{\mu=1}^n$ and linear in $f_{n+1}$ these can be integrated out to yield an equivalent partition function phrased solely in terms of $t_1, \ldots, t_{n+1}$:

$$Z\left(\vec{J}\right) \propto \int_{\mathbb{R}^n} dt_1 \cdots dt_n e^{-\mathcal{S}(\vec{t}, \vec{J})} \tag{A.7}$$

where the action is now

$$\mathcal{S} = -\mathcal{C}(t_1, \ldots, t_n, -iJ_{n+1}) + \sum_{\mu=1}^n \left[\frac{\sigma^2}{2} t_\mu^2 + it_\mu g_\mu + J_\mu\left(i\sigma^2 t_\mu - g_\mu\right) - \frac{\sigma^2}{2} J_\mu^2\right] \tag{A.8}$$

---

[2]We use the characteristic function rather than the moment-generating function since the Fourier transform and its inverse are more analytically tractable than the Laplace transform and its inverse.

The identification $t_{n+1} = -iJ_{n+1}$ arises from the delta function:

$$\frac{1}{2\pi} \int_{-\infty}^{\infty} df_{n+1} e^{-if_{n+1}(iJ_{n+1}+t_{n+1})} = \delta\left(iJ_{n+1} + t_{n+1}\right) \tag{A.9}$$

Recall that $\langle f_\mu \rangle = \partial_{J_\mu} \log Z\left(\vec{J}\right)\Big|_{\vec{J}=\vec{0}}$ and notice that the first term in Eq. A.8 (the cumulant generating function, $\mathcal{C}$) depends on $J_{n+1}$ and not on $\{J_\mu\}_{\mu=1}^{n}$ whereas the rest of the action depends on $\{J_\mu\}_{\mu=1}^{n}$ and not on $J_{n+1}$. Thus, for training points $\langle f_\mu \rangle$ amounts to the average of $g_\mu - i\sigma^2 t_\mu$, and so we identify

$$\forall \mu \in \{1, \ldots, n\}: \qquad \langle it_\mu \rangle = \frac{g_\mu - \langle f_\mu \rangle}{\sigma^2} \equiv \frac{\langle \hat{\delta} g_\mu \rangle}{\sigma^2} \tag{A.10}$$

where $\langle \cdots \rangle$ denotes an expectation value using $Z(\vec{J} = \vec{0})$. We comment that the above relation holds also for any (non-mixed) cumulants of $it_\mu$ and $\hat{\delta} g_\mu / \sigma^2$ except the covariance, where a constant difference appears due to the $O(J^2)$ term in the action, namely

$$\langle it_\mu it_\nu \rangle - \langle it_\mu \rangle \langle it_\nu \rangle = \frac{1}{\sigma^4} \left( \left\langle \hat{\delta} g_\mu \hat{\delta} g_\nu \right\rangle - \left\langle \hat{\delta} g_\mu \right\rangle \left\langle \hat{\delta} g_\nu \right\rangle \right) - \frac{1}{\sigma^2} \delta_{\mu\nu} \tag{A.11}$$

In the GP case the r.h.s. of Eq. A.11 would equal simply $-\tilde{K}_{\mu\nu}^{-1}$, since on the training set the posterior covariance of $\hat{\delta} g$ is the same as that of $f$ and for a GP takes the form

$$\begin{aligned} \Sigma &= K - K\tilde{K}^{-1}K \\ &= K - \left(K + \sigma^2 I - \sigma^2 I\right)\tilde{K}^{-1}K \\ &= \sigma^2 \tilde{K}^{-1}K = \sigma^2 \tilde{K}^{-1}\left(K + \sigma^2 I - \sigma^2 I\right) \\ &= \sigma^2 I - \sigma^4 \tilde{K}^{-1} \end{aligned} \tag{A.12}$$

namely $\Sigma_{\mu\nu} = \sigma^2 \delta_{\mu\nu} - \sigma^4 \tilde{K}_{\mu\nu}^{-1}$ and thus $\frac{1}{\sigma^4}\Sigma_{\mu\nu} - \frac{1}{\sigma^2}\delta_{\mu\nu} = -\tilde{K}_{\mu\nu}^{-1}$. The reader should not be alarmed by having a negative definite covariance matrix for $it_\mu$, since $it_\mu$ cannot be understood as a standard real random variable as its partition function contains imaginary terms.

To make contact with GPs it is beneficial to expand $\mathcal{C}(t_1, \ldots, t_n, -iJ_{n+1})$ in terms of its cumulants, and split the second cumulant, describing the DNNs' NNGP kernel, from the rest. Namely, using Einstein summation

$$\mathcal{C}\left(t_1, \ldots, t_n, -iJ_{n+1}\right) = \frac{1}{2!}\kappa_{\mu_1,\mu_2} it_{\mu_1} it_{\mu_2} + \frac{1}{3!}\kappa_{\mu_1,\mu_2,\mu_3} it_{\mu_1} it_{\mu_2} it_{\mu_3} + O((it)^4) \tag{A.13}$$

$$\kappa_{\mu_1,\mu_2} \equiv K(\mathbf{x}_{\mu_1}, \mathbf{x}_{\mu_2})$$

$$\tilde{\mathcal{C}}\left(t_1, \ldots, t_n, -iJ_{n+1}\right) \equiv \mathcal{C}\left(t_1, \ldots, t_n, -iJ_{n+1}\right) + \frac{1}{2}\kappa_{\mu_1,\mu_2} t_{\mu_1} t_{\mu_2}$$

Writing Eq. A.8 in this fashion gives the action:

$$S = -\tilde{\mathcal{C}}\left(\vec{t}\right) - \frac{1}{2}\sum_{\mu_1,\mu_2}\kappa_{\mu_1,\mu_2} it_{\mu_1} it_{\mu_2} + \sum_{\mu=1}^{n}\left[-\frac{\sigma^2}{2}(it_\mu)^2 + it_\mu g_\mu + J_\mu\left(i\sigma^2 t_\mu - g_\mu\right) - \frac{\sigma^2}{2}J_\mu^2\right] \tag{A.14}$$

## A.2 Saddle-point equation for the mean predictor

Having arrived at the action A.14, we can readily derive the saddle-point equations for the training points by setting:

$$\forall \nu \in \{1, \ldots, n\}: \qquad \partial_{it_\nu} S\left(\vec{t}, \vec{J}\right)\Big|_{\vec{J}=\vec{0}} = 0 \tag{A.15}$$

This corresponds to treating the variables $\{it_\mu\}_{\mu=1}^{n}$ as non-fluctuating quantities, i.e. replacing them with their mean value: $it_\mu \to \langle it_\mu \rangle$. Performing this for the training set $\nu \in \{1, \ldots, n\}$ yields

$$\sum_{\mu=1}^{n}\left(\kappa_{\mu,\nu} + \sigma^2 \delta_{\mu\nu}\right)\langle it_\mu \rangle = g_\nu - \Delta g_\nu \tag{A.16}$$

where

$$\Delta g_\nu = \sum_{r=3}^{\infty} \frac{1}{(r-1)!} \sum_{\mu_1,\ldots,\mu_{r-1}=1}^{n} \kappa_{\nu,\mu_1,\ldots,\mu_{r-1}} \langle it_{\mu_1} \rangle \cdots \langle it_{\mu_{r-1}} \rangle \tag{A.17}$$

where this target shift is related to $\mathcal{C}$ of Eq. A.13 by

$$\Delta g_\nu = \partial_{it_\nu} \tilde{\mathcal{C}}(t_1,\ldots,t_n,t_{n+1}) \tag{A.18}$$

Finally, we get the expression for the mean predictor at the test point by setting $\langle f_* \rangle = \partial_{J_*} \log Z\left(\vec{J}\right)\Big|_{\vec{J}=\vec{0}}$ and plugging in the SP values for $it_\mu$ on the training set from Eq. A.16 and the target shift $\Delta g$ from Eq. A.17. This gives

$$\langle f_* \rangle = \Delta g_* + \sum_{\mu,\nu}^{n} K_\mu^* \tilde{K}_{\mu\nu}^{-1} (g_\nu - \Delta g_\nu) \tag{A.19}$$

## A.3 Posterior covariance

### A.3.1 Posterior covariance on the test point

The posterior covariance on the test point is important for determining the average MSE loss on the test-set, as the latter involves the MSE of the mean-predictor plus the posterior covariance. Concretely, we wish to calculate $\partial_{J_*}^2 \log(Z)$, express it as an expectation value w.r.t $Z$ and calculate this using $Z$ with the self-consistent target shift. To this end, note that generally if $Z(J) = \int e^{-\mathcal{S}(J)}$ then

$$\partial_J^2 \log(Z(J)) = \langle \mathcal{S}'(0)^2 \rangle_{Z(J=0)} - \langle \mathcal{S}'(0) \rangle_{Z(J=0)}^2 - \langle \mathcal{S}''(0) \rangle_{Z(J=0)} \tag{A.20}$$

For the action in Eq. A.14 we have

$$\mathcal{S}'(0) = -\kappa_{*\nu} it_\nu - \underbrace{\partial_{it_*} \tilde{\mathcal{C}}}_{\Delta g_*} \qquad \mathcal{S}''(0) = -\kappa_{**} - \partial_{it_*}^2 \tilde{\mathcal{C}} \tag{A.21}$$

where an Einstein summation over $\nu = 1,\ldots,n$ is implicit. Further recalling that on the training set we have

$$\langle it_\mu \rangle = \sum_\nu \tilde{K}_{\mu\nu}^{-1}(g_\nu - \Delta g_\nu) \tag{A.22}$$

$$\langle it_\mu it_\nu \rangle - \langle it_\mu \rangle \langle it_\nu \rangle = -\tilde{K}_{\mu,\nu}^{-1} \tag{A.23}$$

where here $\Delta g$ is the full quantity without any SP approximations

$$\Delta g_\nu = \sum_{r=3}^{\infty} \frac{1}{(r-1)!} \sum_{\mu_1,\ldots,\mu_{r-1}=1}^{n} \kappa_{\nu,\mu_1,\ldots,\mu_{r-1}} it_{\mu_1} \cdots it_{\mu_{r-1}} \tag{A.24}$$

We obtain

$$\Sigma_{**} = K_{**} + \left\langle \partial_{it_*}^2 \tilde{\mathcal{C}}(it_1,\ldots,it_n,it_*)\,|_{t_*=0} \right\rangle + \left\langle (\kappa_{*\nu} it_\nu + \Delta g_*)^2 \right\rangle - \langle \kappa_{*\nu} it_\nu + \Delta g_* \rangle^2 \tag{A.25}$$

where we can unpack the last two terms as

$$\left\langle (\kappa_{*\nu} it_\nu + \Delta g_*)^2 \right\rangle - \langle \kappa_{*\nu} it_\nu + \Delta g_* \rangle^2 \tag{A.26}$$

$$= \kappa_{*\mu} \kappa_{*\nu} \left( \langle it_\mu it_\nu \rangle - \langle it_\mu \rangle \langle it_\nu \rangle \right) + 2 \left( \langle \Delta g_* \kappa_{*\mu} it_\mu \rangle - \langle \Delta g_* \rangle \langle \kappa_{*\mu} it_\mu \rangle \right) + \left( \langle \Delta g_*^2 \rangle - \langle \Delta g_* \rangle^2 \right)$$

One can verify that for the GP case where $\tilde{\mathcal{C}} = 0$, Eq. A.25 simplifies to

$$\Sigma_{**} = K_{**} - K_\mu^* \tilde{K}_{\mu,\nu}^{-1} K_\nu^* \tag{A.27}$$

which is the standard posterior covariance of a GP [37].

The expressions in Eqs. A.25, A.26 are exact, but to evaluate them in a more compact form we approximate the $t_\mu$ distribution as a Gaussian centered around the SP value.

### A.3.2 Posterior covariance on the training set

Our target shift approach at the saddle-point level allows a computation of the fluctuations of $it_\mu$ using the standard procedure of expanding the action at the saddle-point to quadratic order in the deviation from the SP: $\delta t_\mu \equiv t_\mu - t_\mu^{\mathrm{SP}}$. Due to the saddle-point being an extremum this leads to $\mathcal{S} \approx \mathcal{S}_{\mathrm{saddle}} + \frac{1}{2} \sum_{\mu,\nu} \delta t_\mu A_{\mu\nu}^{-1} \delta t_\nu$ and thus using the standard Gaussian integration formula, one finds that $A_{\mu\nu}$ is the covariance matrix of $it_\mu$. Performing such an expansion on the action of Eq. A.14 one finds

$$A_{\mu\nu}^{-1} = -\left(\sigma^2 \delta_{\mu\nu} + K_{\mu\nu} + \Delta K_{\mu\nu}\right) \tag{A.28}$$
$$\Delta K_{\mu\nu} = \partial_{it_\mu} \underbrace{\partial_{it_\nu} \tilde{\mathcal{C}}\left(it_1, \ldots, it_n\right)}_{\Delta g_\nu}$$

where the $it_\mu$ on the r.h.s. are those of the saddle-point. Recalling Eq. A.11 we have

$$\Sigma_{\mu\nu} = \langle f_\mu f_\nu \rangle - \langle f_\mu \rangle \langle f_\nu \rangle = -\sigma^4 \left[\sigma^2 I + K + \Delta K\right]_{\mu\nu}^{-1} + \sigma^2 \delta_{\mu\nu} \tag{A.29}$$

and the r.h.s. coincides with the posterior covariance of a GP with a kernel equal to $K + \Delta K$.

### A.4 A criterion for the saddle-point regime

Saddle-point approximations are commonly used in statistics [12] and physics and often rely on having partition functions of the form $Z = \int dt e^{-n\mathcal{S}(t)}$ where $n$ is a large number and $\mathcal{S}$ is $O(1)$. In our settings we cannot simply extract such a large factor from the action and make it $O(1)$. Nonetheless, we argue that expanding the action to quadratic order around the saddle-point is still a good approximation at large $n$, with $n$ being the training set size. Concretely we give the following two consistency criteria based on comparing the saddle-point results with their leading order beyond-saddle-point corrections. The first is given by the latter correction to the mean predictor over the scale of the saddle-point prediction

$$\frac{1}{2} \left[\partial_{it_\nu} \partial_{it_\eta} \Delta g_\mu\right] \tilde{K}_{\mu_0\mu}^{-1} \tilde{K}_{\nu\eta}^{-1} \ll O(g) \tag{A.30}$$

where an Einstein summation over the training-set is implicit and the derivatives are evaluated at the saddle-point value. This criterion can be calculated for any specific model to verify the appropriateness of the saddle-point approach. We further provide a simpler criterion

$$n \left(\frac{\hat{\delta} g}{\sigma^2}\right)^2 \gg 1 \tag{A.31}$$

which however relies on heuristic assumptions. The main purpose of this heuristic criterion is to provide a qualitative explanation for why we expect the first criterion to be small in many interesting large $n$ settings.

To this end we first obtain the leading (beyond quadratic) correction to the mean. Consider the partition function in terms of $it$ and its expansion around the saddle-point. As $P_0[f]$ is effectively bounded (by the Gaussian tails of the finite set of weights), the corresponding characteristic function $(e^{\mathcal{C}(t_1,\ldots,t_n)})$ is well defined over the entire complex plane. Given this, one can deform the integration contour, along each dimension $(\int_{-\infty}^{\infty} dt_\mu)$, which originally laid on the real axis, to $\int_{-\infty+t_{\mathrm{SP},\mu}}^{\infty+t_{\mathrm{SP},\mu}} dt_\mu$ where $t_{\mathrm{SP}}$ is purely imaginary and equals $-i\hat{\delta}g/\sigma^2$ so it crosses the saddle-point (see also Ref. [12]). Next we expand the action in the deviation from the SP value: $\delta t_\mu = t_\mu - t_{\mathrm{SP},\mu}$ to obtain

$$Z = \int_{-\infty}^{\infty} \delta t_1 \cdots \delta t_n \exp\left(-\mathcal{S}_0 - \frac{1}{2!}\delta t_\mu \mathcal{S}_{\mu\nu} \delta t_\nu - \frac{1}{3!}\mathcal{S}_{\mu\nu\eta}\delta t_\mu \delta t_\nu \delta t_\eta + O\left(\delta t^4\right)\right) \tag{A.32}$$

where an Einstein summation over the training set is implicit and where we denoted for $m \geq 3$

$$\mathcal{S}_{\mu_1\ldots\mu_m} \equiv \partial_{t_{\mu_1}} \cdots \partial_{t_{\mu_m}} \mathcal{S}\Big|_{\vec{t}=\vec{t}_{\mathrm{SP}}} = -\partial_{t_{\mu_1}} \cdots \partial_{t_{\mu_m}} \tilde{\mathcal{C}}\Big|_{\vec{t}=\vec{t}_{\mathrm{SP}}} = i\,\partial_{t_{\mu_1}} \cdots \partial_{t_{\mu_{m-1}}} \Delta g_{\mu_m}\Big|_{\vec{t}=\vec{t}_{\mathrm{SP}}} \tag{A.33}$$

Next we consider first order perturbation theory in the cubic term and calculate the correction to the mean of $it_\mu$ or equivalently $\hat{\delta}g_\mu/\sigma^2$.

$$\langle it_{\mu_0}\rangle \approx \langle it_{\mu_0}\rangle_{\text{SP}} - \frac{1}{6}\mathcal{S}_{\mu\nu\eta}\langle i\delta t_{\mu_0}\delta t_\mu \delta t_\nu \delta t_\eta\rangle_{\text{SP,connected}} \tag{A.34}$$

$$= \frac{\hat{\delta}g_{\mu_0}}{\sigma^2} - \frac{i}{2}\mathcal{S}_{\mu\nu\eta}\tilde{K}^{-1}_{\mu_0\mu}\tilde{K}^{-1}_{\nu\eta}$$

$$= \frac{\hat{\delta}g_{\mu_0}}{\sigma^2} + \frac{1}{2}\partial_{it_\nu}\partial_{it_\eta}\Delta g_\mu|_{\vec{t}=\vec{t}_{\text{SP}}}\tilde{K}^{-1}_{\mu_0\mu}\tilde{K}^{-1}_{\nu\eta}$$

where here the kernel is shifted: $\tilde{K} = \sigma^2 I + K + \Delta K$, as in Eq. A.28 and $\langle ...\rangle_{\text{SP,connected}}$ means keeping terms in Wick's theorem which connect the operator being averaged ($i\delta t_{\mu_0}$) with the perturbation, as standard in perturbation theory. Comparing the last term on the right hand side (i.e. the correction) with the predictions which are $O(g)$ gives the first criterion, Eq. A.30. Depending on context, it may be more appropriate to compare this term with the discrepancy rather than the prediction.

Next we turn to study the scaling of this correction with $n$. To this end we first consider a single derivative of $\Delta g$ ($\partial_{it_\nu}\Delta g_\mu$). Note that $\Delta g_\mu$, by its definition, includes contributions from at least $n^3$ different $it$'s. In many cases, one expects that the value of this sum will be dominated by some finite fraction of the training set rather than by a vanishing fraction. This assumption is in fact implicit in our EK treatment where we replaced all $\sum_\mu$ with $n\int$. Given so, the derivative $\partial_{it_\nu}\Delta g_\mu$, which can be viewed as the sensitivity to changing $it_\nu$, is expected to go as one over the size of that fraction of the training set, namely as $1/n$. Under this collectivity assumption we expect the scaling

$$\partial_{it_\nu}\Delta g_\mu = O(it^{-1}\Delta g n^{-1}) \tag{A.35}$$

Making a similar collectivity assumption on higher derivatives yields

$$\partial_{it_\eta}\partial_{it_\nu}\Delta g_\mu = O(it^{-2}\Delta g n^{-2}) \tag{A.36}$$

Following this we count powers of $n$ in Eq. A.34 and find a $n^{-2}$ contribution from the second derivative of $\Delta g$ and a contribution from the summation over $\sum_{\eta\nu}$. Despite containing two summations, we argue that the latter is in fact order $n$. To this end consider $n^2\partial_\nu\partial_\eta\Delta g_\mu$ for fixed $\nu,\eta$, as an effective target function ($G_\mu(\nu,\eta)$) where we multiplied by the scaling of the second derivative to make $G$ order 1. The above summation appears then as $\sum_{\eta,\nu}\tilde{K}^{-1}_{\eta\nu}G_\mu(\nu,\eta)$. Next we recall that $it_{\mu_0} = \tilde{K}^{-1}_{\mu_0\mu}g_\mu = \hat{\delta}g_{\mu_0}/\sigma^2$, and so multiplication of a vector with $\tilde{K}^{-1}$ can be interpreted as the discrepancy w.r.t. the $G_\mu$ target. Accordingly the above summation over $\mu$ can be viewed as performing GP Regression on $G_\mu(\nu,\eta)$ leading to train discrepancy ($it_\mu[G(\nu,\eta)]$) which is order $G_\mu(\nu,\eta)$ and hence order 1. The remaining summation has now a summand of the order 1 and hence is $O(n)$ or smaller. We thus find that the correction to the saddle-point scales as

$$\langle it_{\mu_0}\rangle - \langle it_{\mu_0}\rangle_{\text{SP}} = O\left(\frac{\Delta g}{n(\hat{\delta}g/\sigma^2)^2}\right) \tag{A.37}$$

Generally we expect $\Delta g \approx 0$ at strong over-parameterization (as non-linear effects are suppressed by $C^{-1}$) and $\Delta g \sim O(g)$ at good performances (as this implies good performance on the training set). Thus we generally expect $\Delta g = O(g) = O(1)$ and hence large $n(\hat{\delta}g/\sigma^2)^2$ controls the magnitude of the corrections. Considering the $\sigma^2 \to 0$ limit, we note in passing that $\hat{\delta}g/\sigma^2$ typically remains finite. For instance it is simply $K^{-1}g$ for a Gaussian Process.

Considering the linear CNN model of the main text, we estimate the above heuristic criterion for $n = 650$ and $C = 8$ where $\Delta g = O(g)$ and $\hat{\delta}g \approx 0.1g$. This then gives $(6.5O(g)^2)^{-1}$ as the small factor dominating the correction. As we choose $O(g) = 3$ in that experiment, we find that the correction is roughly $1/60$. As the discrepancy is $0.1g = O(0.3)$ we expect roughly a 5% relative error in predicting the discrepancy.

## B  Review of the Edgeworth expansion

In this section we give a review of the Edgeworth expansion, starting from the simplest case of a scalar valued RV and then moving on vector valued RVs so we can write down the expansion for the output of a generic neural network on a fixed set of inputs.

## B.1 Edgeworth expansion for a scalar random variable

Consider scalar valued continuous iid RVs $\{Z_i\}$ and assume WLOG $\langle Z_i \rangle = 0$, $\langle Z_i^2 \rangle = 1$, with higher cumulants $\kappa_r^Z$ for $r \geq 3$. Now consider their normalized sum $Y_N = \frac{1}{\sqrt{N}} \sum_{i=1}^N Z_i$. Recall that cumulants are *additive*, i.e. if $Z_1, Z_2$ are independent RVs then $\kappa_r(Z_1 + Z_2) = \kappa_r(Z_1) + \kappa_r(Z_2)$ and that the $r$-th cumulant is *homogeneous* of degree $r$, i.e. if $c$ is any constant, then $\kappa_r(cZ) = c^r \kappa_r(Z)$. Combining additivity and homogeneity of cumulants we have a relation between the cumulants of $Z$ and $Y$

$$\kappa_{r \geq 2} := \kappa_{r \geq 2}^Y = \frac{N \kappa_r^Z}{(\sqrt{N})^r} = \frac{\kappa_r^Z}{N^{r/2-1}} \tag{B.1}$$

Now, let $\varphi(y) := (2\pi)^{-1/2} e^{-y^2/2}$ be the PDF of the standard normal distribution. The *characteristic function* of $Y$ is given by the Fourier transform of its PDF $P(y)$ and is expressed via its cumulants

$$\hat{P}(t) := \mathcal{F}[P(y)] = \exp\left(\sum_{r=1}^\infty \kappa_r \frac{(it)^r}{r!}\right) = \exp\left(\sum_{r=3}^\infty \kappa_r \frac{(it)^r}{r!}\right) \hat{\varphi}(t) \tag{B.2}$$

where the last equality holds since $\kappa_1 = 0$, $\kappa_2 = 1$ and $\hat{\varphi}(t) = e^{-\frac{t^2}{2}}$. From the CLT, we know that $P(y) \to \varphi(y)$ as $N \to \infty$. Taking the inverse Fourier transform $\mathcal{F}^{-1}$ has the effect of mapping $it \mapsto -\partial_y$ thus

$$P(y) = \exp\left(\sum_{r=3}^\infty \kappa_r \frac{(-\partial_y)^r}{r!}\right) \varphi(y) = \varphi(y)\left(1 + \sum_{r=3}^\infty \frac{\kappa_r}{r!} H_r(y)\right) \tag{B.3}$$

where $H_r(y)$ is the $r$th *probabilist's Hermite polynomial*, defined by

$$H_r(y) = (-)^r e^{y^2/2} \frac{d^r}{dy^r} e^{-y^2/2} \tag{B.4}$$

e.g. $H_4(y) = y^4 - 6y^2 + 3$.

## B.2 Edgeworth expansion for a vector valued random variable

Consider now the analogous procedure for vector-valued RVs in $\mathbb{R}^n$ (see [31]). We perform an Edgeworth expansion around a centered multivariate Gaussian distribution with covariance matrix $\kappa^{i,j}$

$$\varphi(\vec{y}) = \frac{1}{(2\pi)^{d/2} \det(\kappa^{i,j})} \exp\left(-\frac{1}{2}\kappa_{i,j} y^i y^j\right) \tag{B.5}$$

where $\kappa_{i,j}$ is the matrix inverse of $\kappa^{i,j}$ and Einstein summation is used. The $r$'th order cumulant becomes a tensor with $r$ indices, e.g. the analogue of $\kappa_4$ is $\kappa^{i,j,k,l}$. The Hermite polynomials are now multi-variate polynomials, so that the first one is $H_i = \kappa_{i,j} y^j$ and the fourth one is

$$\begin{aligned}
H_{ijkl}(\vec{y}) &= e^{\frac{1}{2}\kappa_{i',j'} y^{i'} y^{j'}} \partial_i \partial_j \partial_k \partial_l e^{-\frac{1}{2}\kappa_{i',j'} y^{i'} y^{j'}} \\
&= H_i H_j H_k H_l - H_i H_j \kappa_{k,l}[6] + \kappa_{i,j}\kappa_{k,l}[3]
\end{aligned} \tag{B.6}$$

where the postscript bracket notation is simply a convenience to avoid listing explicitly all possible partitions of the indices, e.g. $\kappa_{i,j}\kappa_{k,l}[3] = \kappa_{i,j}\kappa_{k,l} + \kappa_{i,k}\kappa_{j,l} + \kappa_{i,l}\kappa_{j,k}$

In our context we are interested in even distributions where all odd cumulants vanish, so the Edgeworth expansion reads

$$P(\vec{y}) = \exp\left(\frac{\kappa^{i,j,k,l}}{4!}\partial_i \partial_j \partial_k \partial_l + \dots\right) \varphi(\vec{y}) = \varphi(\vec{y})\left(1 + \frac{\kappa^{i,j,k,l}}{4!} H_{ijkl} + \dots\right) \tag{B.7}$$

## B.3 Edgeworth expansion for the posterior of Bayesian neural network

Consider an on-data formulation, i.e. a distribution over a vector space - the NN output evaluated on the training set and on a single test point, rather than a distribution over the whole function space:

$$f(\mathbf{x}) \to \vec{f} \equiv (f(\mathbf{x}_1), \dots, f(\mathbf{x}_n), f(\mathbf{x}_{n+1})) \in \mathbb{R}^{n+1} \qquad \mathbf{x}_{n+1} = \mathbf{x}_* \tag{B.8}$$

where $\mathbf{x}_*$ is the test point. Let $\kappa_r$ denote the $r$th cumulant of the prior $P_0\left(\vec{f}\right)$ of the network over this space:

$$[\kappa_r]_{\mu_1,\ldots,\mu_r} = \langle f\left(\mathbf{x}_{\mu_1}\right),\ldots,f\left(\mathbf{x}_{\mu_r}\right)\rangle - \text{"disconnected averages"} \qquad \mu \in \{1,\ldots,n+1\} \quad \text{(B.9)}$$

Take the baseline distribution to be Gaussian $P_G\left(\vec{f}\right) \propto \exp\left(-\frac{1}{2}\vec{f}^{\mathsf{T}}K^{-1}\vec{f}\right)$, around which we perform the Edgeworth expansion, thus the characteristic function of the prior reads

$$\hat{P}_0\left(\vec{t}\right) = \exp\left(\sum_{r=4}^{\infty}\frac{\kappa_r\left(i\vec{t}\right)^r}{r!}\right)\hat{P}_G\left(\vec{t}\right) \tag{B.10}$$

and thus

$$P_0\left(\vec{f}\right) = \exp\left(\sum_{r=4}^{\infty}\frac{(-)^r\kappa_r\vec{\partial}^r}{r!}\right)P_G\left(\vec{f}\right) \tag{B.11}$$

where we used the shorthand notation:

$$\kappa_r\vec{\partial}^r \equiv \sum_{\mu_1,\ldots,\mu_r}[\kappa_r]_{\mu_1,\ldots,\mu_r}\,\partial_{f_{\mu_1}}\cdots\partial_{f_{\mu_r}} \tag{B.12}$$

and the indices range over both the train set and the test point $\mu \in \left\{1,\ldots,\underbrace{n+1}_{*}\right\}$. In our case, all odd cumulants vanish, thus

$$\exp\left(\sum_{r=4}^{\infty}\frac{(-)^r\kappa_r\vec{\partial}^r}{r!}\right) = \exp\left(\sum_{r=4}^{\infty}\frac{\kappa_r\vec{\partial}^r}{r!}\right) \tag{B.13}$$

Introducing the data term and a source term, the partition function reads (denote $f\left(\mathbf{x}_\mu\right) \equiv f_\mu$, $f\left(\mathbf{x}_*\right) \equiv f_*$)

$$Z\left(J\right) = \int d\vec{f}\left(\exp\left(\sum_{r=4}^{\infty}\frac{\kappa_r\vec{\partial}^r}{r!}\right)P_G\left(\vec{f}\right)\right)\exp\left(-\frac{1}{2\sigma^2}\sum_{\mu=1}^{n}(g_\mu - f_\mu)^2 + \sum_{\mu=1}^{n+1}J_\mu f_\mu\right) \tag{B.14}$$

## C   Target shift equations - alternative derivation

Here we derive our self-consistent target shift equations from a different approach which does not require the introduction of the $it_\mu$ integration variables by transforming to Fourier space. While this approach requires an additional assumption (see below) it also has the benefit of being extendable to any smooth loss function comprised of a sum over training points. In particular, below we derive it for both MSE loss and cross entropy loss.

To this end, we examine the Edgeworth expansion for the partition function given by Eq. B.14. By using a series of integration by parts and noting the boundary terms vanish, one can shift the action of the higher cumulants from the prior to the data dependent term

$$Z\left(\vec{J}\right) = \int d\vec{f}P_G\left(\vec{f}\right)\left[\exp\left(\sum_{r=3}^{\infty}\frac{1}{r!}\sum_{\mu_1,\ldots,\mu_r=1}^{n+1}\kappa_{\mu_1,\ldots,\mu_r}\partial_{f_{\mu_1}}\cdots\partial_{f_{\mu_r}}\right)\exp\left(-\frac{1}{2\sigma^2}\sum_{\mu=1}^{n}(g_\mu - f_\mu)^2 + \sum_{\mu=1}^{n+1}J_\mu f_\mu\right)\right]$$
$$\text{(C.1)}$$

Doing so yields an equivalent viewpoint on the problem, wherein the Gaussian data term and the non-Gaussian prior appearing in Eq. B.14 are replaced in Eq. C.1 by a Gaussian prior and a non-Gaussian data term.

Next we argue that in the large $n$ limit, the non-Gaussian data-term can be expressed as a Gaussian-data term but on a shifted target. To this end we note that when $n$ is large, most combinations of derivatives in the exponents act on different data points. In such cases derivatives could simply be replaced as $\partial_{\mu_i} \to \sigma^{-2}\hat{\delta}g_{\mu_i}$, where $\hat{\delta}g_{\mu_i} \equiv g_{\mu_i} - f_{\mu_i}$ denotes the discrepancy on the training point $\mu_i$.

Consider next how $f_\nu$ on a particular training point ($\nu$) is affect by these derivative terms. Following the above observation, most terms in the exponent will not act on $f_\nu$ and a $1/n$ portion will contain a single derivative. The remaining rarer cases, where two derivatives act on the same $\nu$, are neglected. For each $f_\nu$ we thus replace $r-1$ derivatives in the order $r$ term in C.1 by discrepancies, leaving a single derivative operator that is multiplied by the following quantity

$$\Delta g_\nu \equiv \sum_{r=3}^{\infty} \frac{1}{(r-1)!} \sum_{\mu_1,\ldots,\mu_{r-1}}^{n} \kappa_{\nu,\mu_1\ldots\mu_{r-1}} (\sigma^{-2}\hat{\delta}g_{\mu_1})\cdots(\sigma^{-2}\hat{\delta}g_{\mu_{r-1}}) \qquad \text{(C.2)}$$

Note that the summation indices span only the training set, not the test point: $\mu_1, ..., \mu_{r-1} \in \{1, \ldots, n\}$, whereas the free index spans also the test point $\nu \in \{1, \ldots, n+1\}$.

Recall that an exponentiated derivative operator acts as a shifting operator, e.g. for some constant $a \in \mathbb{R}$, any smooth scalar function $\varphi$ obeys $e^{a\partial_x}\varphi(x) = \varphi(x+a)$. If this $\Delta g$ was a constant, the differential operator could now readily act on the data term. Next we make again our collectivity assumption: as $\Delta g$ involves a sum over many data-points, it will be a weakly fluctuating quantity in the large $n$ limit provided the contribution to $\Delta g$ comes from a collective effect rather than by a few data points. We thus perform our second approximation, of the mean-field type, and replace $\Delta g$ by its average $\overline{\Delta g}$, leading to

$$Z\left(\vec{J};\overline{\Delta g}\right) = \int d\vec{f}P_G\left(\vec{f}\right)\exp\left(-\frac{1}{2\sigma^2}\sum_{\mu=1}^{n}\left(g_\mu - \overline{\Delta g}_\mu - f_\mu\right)^2 + \sum_{\mu=1}^{n+1} J_\mu\left(f_\mu + \overline{\Delta g}_\mu\right)\right) \quad \text{(C.3)}$$

Given a fixed $\overline{\Delta g}$, C.3 is the partition function corresponding to a GP with the train targets shifted by $\overline{\Delta g}_\mu$ and the test target shifted by $\overline{\Delta g}_*$. Following this we find that $\overline{\Delta g}$ depends on the discrepancy of the GP prediction which in turn depends on $\overline{\Delta g}$. In other words we obtain our self-consistent equation: $\overline{\Delta g} = \langle \Delta g_\mu \rangle_{Z(\vec{J};\overline{\Delta g})}$.

The partition function C.3 reflects the correspondence between finite DNNs and a GP with its target shifted by $\overline{\Delta g}$. To facilitate the analytic solution of this self-consistent equation, we focus on the case $\left\langle \hat{\delta}g_\mu \hat{\delta}g_\nu \right\rangle \ll \left\langle \hat{\delta}g_\mu \right\rangle \left\langle \hat{\delta}g_\nu \right\rangle$ at least for $\mu \neq \nu$. We note that this was the case for the two toy models we studied.

Given this, the expectation value over $\Delta g$ using the GP defined by $Z\left(\vec{J};\overline{\Delta g}\right)$, which consists of products of expectation values of individual discrepancies and correlations between two discrepancies, can then be expressed using only the former. Omitting correlations within the GP expectation value, one obtains a simplified self-consistent equation involving only the average discrepancies:

$$\forall\mu \in \{1,\ldots,n\}: \qquad \langle \hat{\delta}g_\mu \rangle_{Z(\vec{J};\overline{\Delta g})} \equiv g_\mu - \overline{\Delta g}_\mu - \langle f_\mu \rangle_{Z(\vec{J};\overline{\Delta g})} = g_\mu - \overline{\Delta g}_\mu - \sum_{\nu,\rho=1}^{n} K_{\mu\nu}\tilde{K}_{\nu\rho}^{-1}\left(g_\rho - \overline{\Delta g}_\rho\right) \tag{C.4}$$

with $\hat{\delta}g_\mu$ now understood as a number, also within C.2. Lastly, we plug the solution to these equations to find the prediction on the test point: $\langle f(\mathbf{x}_*)\rangle_{Z(\vec{J};\overline{\Delta g})}$. These coincide with the self-consistent equations derived via the saddle-point approximation in the main text.

Notably the above derivation did not hinge on having MSE loss. For any loss given as a sum over training points, $\mathcal{L} = \sum_{\mu}^{n} L_\mu(f_\mu)$, the above derivation should hold with $\sigma^{-2}\hat{\delta}g_\mu$ in $\Delta g_\nu$ replaced by $\partial_{f_\mu}L_\mu$. In particular for the cross entropy loss where $f_{\nu,i}$ is the pre-softmax output of the DNN for class $i$ we will have

$$\partial_{f_{\nu,i}}L_\nu = -\delta_{i_\nu,i} + \frac{e^{f_{\nu,i}}}{\sum_j e^{f_{\nu,j}}} \tag{C.5}$$

where $i$ and $j$ run over all classes, $i_\nu$ is the class of $\mathbf{x}_\nu$. Neatly, the above r.h.s. is again a form of discrepancy but this time in probability space. Namely it is $p_{\text{model}}(i|\mathbf{x}_\nu) - p_{\text{data}}(i|\mathbf{x}_\nu)$, where $p_{\text{model}}$ is the distribution generated by the softmax layer, and $p_{\text{data}}$ is the empirical distribution. Following this one can readily derive self-consistent equations for cross entropy loss and solve them numerically. Further analytical progress hinges on developing analogous of the EK approximation for cross entropy loss.

# D Review of the Equivalent Kernel (EK)

In this appendix we generally follow [37], see also [42] for more details. The posterior mean for GP regression

$$\bar{f}_{\text{GP}}(\mathbf{x}_*) = \sum_{\mu,\nu} K_\mu^* \tilde{K}_{\mu\nu}^{-1} y_\nu \tag{D.1}$$

can be obtained as the function which minimizes the functional

$$J[f] = \frac{1}{2\sigma^2} \sum_{\alpha=1}^{n} (y_\alpha - f(\mathbf{x}_\alpha))^2 + \frac{1}{2} ||f||_{\mathcal{H}}^2 \tag{D.2}$$

where $||f||_{\mathcal{H}}$ is the RKHS norm corresponding to kernel $K$. Our goal is now to understand the behaviour of the minimizer of $J[f]$ as $n \to \infty$. Let the data pairs $(\mathbf{x}_\alpha, y_\alpha)$ be drawn from the probability measure $\mu(\mathbf{x}, y)$. The expectation value of the MSE is

$$\mathbb{E}\left[ \sum_{\alpha=1}^{n} (y_\alpha - f(\mathbf{x}_\alpha))^2 \right] = n \int (y - f(\mathbf{x}))^2 \, d\mu(\mathbf{x}, y) \tag{D.3}$$

Let $g(\mathbf{x}) \equiv \mathbb{E}[y|\mathbf{x}]$ be the ground truth regression function to be learned. The variance around $g(\mathbf{x})$ is denoted $\sigma^2(\mathbf{x}) = \int (y - g(\mathbf{x}))^2 \, d\mu(y|\mathbf{x})$. Then writing $y - f = (y - g) + (g - f)$ we find that the MSE on the data target $y$ can be broken up into the MSE on the ground truth target $g$ plus variance due to the noise

$$\int (y - f(\mathbf{x}))^2 \, d\mu(\mathbf{x}, y) = \int (g(\mathbf{x}) - f(\mathbf{x}))^2 \, d\mu(\mathbf{x}) + \int \sigma^2(\mathbf{x}) \, d\mu(\mathbf{x}) \tag{D.4}$$

Since the right term on the RHS of D.4 does not depend on $f$ we can ignore it when looking for the minimizer of the functional which is now replaced by

$$J_\mu[f] = \frac{n}{2\sigma^2} \int (g(\mathbf{x}) - f(\mathbf{x}))^2 \, d\mu(\mathbf{x}) + \frac{1}{2} ||f||_{\mathcal{H}}^2 \tag{D.5}$$

To proceed we project $g$ and $f$ on the eigenfunctions of the kernel with respect to $\mu(\mathbf{x})$ which obey $\int \mu(\mathbf{x}') K(\mathbf{x}, \mathbf{x}') \psi_s(\mathbf{x}') = \lambda_s \psi_s(\mathbf{x})$. Assuming that the kernel is non-degenerate so that the $\psi$'s form a complete orthonormal basis, for a sufficiently well behaved target we may write $g(\mathbf{x}) = \sum_s g_s \psi_s(\mathbf{x})$ where $g_s = \int g(\mathbf{x}) \psi_s(\mathbf{x}) \, d\mu(\mathbf{x})$, and similarly for $f$. Thus the functional becomes

$$J_\mu[f] = \frac{n}{2\sigma^2} \sum_s (g_s - f_s)^2 + \frac{1}{2} \sum_s \frac{f_s^2}{\lambda_s} \tag{D.6}$$

This is easily minimized by taking the derivative w.r.t. each $f_s$ to yield

$$f_s = \frac{\lambda_s}{\lambda_s + \sigma^2/n} g_s \tag{D.7}$$

In the limit $n \to \infty$ we have $\sigma^2/n \to 0$ thus we expect that $f$ would converge to $g$. The rate of this convergence will depend on the smoothness of $g$, the kernel $K$ and the measure $\mu(\mathbf{x}, y)$. From D.7 we see that if $n\lambda_s \ll \sigma^2$ then $f_s$ is effectively zero. This means that we cannot obtain information about the coefficients of eigenfunctions with small eigenvalues until we get a sufficient amount of data. Plugging the result D.7 into $f(\mathbf{x}) = \sum_s f_s \psi_s(\mathbf{x})$ and recalling $g_s = \int g(\mathbf{x}') \psi_s(\mathbf{x}') \, d\mu(\mathbf{x}')$ we find

$$\langle f(\mathbf{x}) \rangle_{\text{EK}} = \sum_s \frac{\lambda_s g_s}{\lambda_s + \sigma^2/n} \psi_s(\mathbf{x}) = \int \underbrace{\sum_s \frac{\lambda_s \psi_s(\mathbf{x}) \psi_s(\mathbf{x}')}{\lambda_s + \sigma^2/n}}_{h(\mathbf{x}, \mathbf{x}')} g(\mathbf{x}') \, d\mu(\mathbf{x}') \tag{D.8}$$

The term $h(\mathbf{x}, \mathbf{x}')$ it the *equivalent kernel*. Notice the similarity to the vector-valued equivalent kernel weight function $\mathbf{h}(\mathbf{x}_*) = (\mathbf{K} + \sigma^2 I)^{-1} \mathbf{k}(\mathbf{x}_*)$ where $\mathbf{K}$ denotes the $n \times n$ matrix of covariances between the training points with entries $K(\mathbf{x}_\mu, \mathbf{x}_\nu)$ and $\mathbf{k}(\mathbf{x}_*)$ is the vector of covariances with elements $K(\mathbf{x}_\mu, \mathbf{x}_*)$. The difference is that in the usual discrete formulation the prediction was obtained as a linear combination of a finite number of observations $y_i$ with weights given by $h_i(\mathbf{x})$ while here we have instead a continuous integral.

# E  Additional technical details for solving the self-consistent equations

## E.1  EK limit for the CNN toy model

In this subsection we show how to arrive at Eq. 16 in the main text, which is a self-consistent equation for the proportionality constant, $\alpha$, defined by $\hat{\delta}g = \alpha g$. We first show that both the shift and the discrepancy can be written as a proportionality factor times the target, and then derive the equation.

### E.1.1  The shift and the discrepancy in terms of the target

Recall that we assume a linear target with a single channel:

$$g\left(\mathbf{x}\right) = \sum_{k=1}^{N} a_k^* \left(\mathbf{w}^* \cdot \tilde{\mathbf{x}}_k\right) \tag{E.1}$$

A useful relation in our context is

$$\int d\mu\left(\mathbf{x}^2\right)\left(\tilde{\mathbf{x}}_i^1 \cdot \tilde{\mathbf{x}}_j^2\right) g\left(\mathbf{x}^2\right) = \int d\mu\left(\mathbf{x}^2\right)\left(\tilde{\mathbf{x}}_i^1 \cdot \tilde{\mathbf{x}}_j^2\right) \sum_{k=1}^{N} a_k^*\left(\mathbf{w}^* \cdot \tilde{\mathbf{x}}_k^2\right) \tag{E.2}$$

$$= \left(\tilde{\mathbf{x}}_i^1\right)^{\mathsf{T}} \left(\sum_{k=1}^{N} a_k^* \underbrace{\int d\mu\left(\mathbf{x}^2\right) \tilde{\mathbf{x}}_j^2 \left(\tilde{\mathbf{x}}_k^2\right)^{\mathsf{T}}}_{\delta_{jk} I_S}\right) \mathbf{w}^*$$

$$= a_j^*\left(\tilde{\mathbf{x}}_i^1 \cdot \mathbf{w}^*\right)$$

This integral is useful in our context since the general cumulant of order $2m$ in Eq. 14 involves such inner products of the form $\left(\tilde{\mathbf{x}}_i^1 \cdot \tilde{\mathbf{x}}_j^2\right)$ and in the EK limit each such inner product is integrated against a discrepancy $\hat{\delta}g(\mathbf{x})$. Thus, if it were the case that $\hat{\delta}g(\mathbf{x})$ could be written in terms of $g$ in a simple way, the self-consistent equation could be simplified dramatically. Indeed, we argue here that the discrepancy is related to the target by the simple relation:

$$\hat{\delta}g \equiv g - \langle f \rangle = \alpha g \tag{E.3}$$

Note that $\alpha$ itself is the solution to an equation that is non-linear in $g$, hence it is also non-linear in $g$ so Eq. E.3 does not imply that $\hat{\delta}g$ scales linearly with $g$. However, as mentioned in the main text, this ansatz simplifies the self-consistent equation in the EK limit from a functional equation to a scalar equation for the factor $\alpha$.

One direct way to argue for this ansatz is to simply plug it in the self-consistent equation, solve it and compare with a numerical simulation. This is indeed what we have done, arriving at Eq. 16, and the very good match with experiments shown in Fig. 1 provides strong evidence for the validity of this ansatz. We can also provide a more systematic argument to justify this ansatz. In the large data set limit, the posterior over functions, constrained in this model to linear functions, becomes symmetric to rotations of the inputs which preserve the target $g$. Following this, the average of $f$ must be of the form of Eq. E.3, as any function orthogonal to this would violate this symmetry.

Now we argue that the target shift also admits a similar relation to the target, with some other factor: $\Delta g = \alpha_\Delta g$. As we show in the next sub-subsection, this factor of $\alpha_\Delta$ can be written in terms of $\alpha$ thereby arriving at a single equation for $\alpha$. Notice that the target shift has a form of a geometric series. In the linear CNN toy model we are able to sum this entire series, whose first term is related to

(using the notation introduced in §F):

$$\int d\mu\left(\mathbf{x}^2\right) d\mu\left(\mathbf{x}^3\right) d\mu\left(\mathbf{x}^4\right) \kappa_4\left(\mathbf{x}^1, \mathbf{x}^2, \mathbf{x}^3, \mathbf{x}^4\right) g\left(\mathbf{x}^2\right) g\left(\mathbf{x}^3\right) g\left(\mathbf{x}^4\right) \tag{E.4}$$

$$= \frac{\lambda^2}{C} \int d\mu_{2:4} \sum_{i,j=1}^{N} \left\{(1_i 3_j)\left[(2_i 4_j) + (4_i 2_j)\right] + (1_i 4_j)\left[(2_i 3_j) + (3_i 2_j)\right] + (1_i 2_j)\left[(3_i 4_j) + (4_i 3_j)\right]\right\} g\left(\mathbf{x}^2\right) g\left(\mathbf{x}^3\right) g\left(\mathbf{x}^4\right)$$

$$= \frac{\lambda^2}{C} \sum_{i,j=1}^{N} \left\{2 a_i^* \left(a_j^*\right)^2 \|\mathbf{w}^*\|^2 \left(\tilde{\mathbf{x}}_i^1 \cdot \mathbf{w}^*\right) + 2 a_i^* \left(a_j^*\right)^2 \|\mathbf{w}^*\|^2 \left(\tilde{\mathbf{x}}_i^1 \cdot \mathbf{w}^*\right) + 2 a_i^* \left(a_j^*\right)^2 \|\mathbf{w}^*\|^2 \left(\tilde{\mathbf{x}}_i^1 \cdot \mathbf{w}^*\right)\right\}$$

$$= \frac{6\lambda^2}{C} \|\mathbf{w}^*\|^2 \underbrace{\left(\sum_{j=1}^{N} \left(a_j^*\right)^2\right)}_{\approx \sigma_a^2} \underbrace{\sum_{i=1}^{N} a_i^* \left(\tilde{\mathbf{x}}_i^1 \cdot \mathbf{w}^*\right)}_{g(\mathbf{x}^1)}$$

$$\approx \frac{6\lambda^2}{C} \sigma_a^2 \|\mathbf{w}^*\|^2 g\left(\mathbf{x}^1\right)$$

For simplicity we can assume $\|\mathbf{w}^*\|^2 = 1$ and $\sigma_a^2 = 1$, thus getting a simple proportionality constant of $\frac{6\lambda^2}{C}$. If we were to trade $g$ for $\hat{\delta}g$, as we have in $\Delta g$, we would get a similar result, with an extra factor of $\left(\frac{\alpha}{\sigma^2/n}\right)^3$. The factor of 6 will cancel out with the factor of $1/(4-1)!$ appearing in the definition of $\Delta g$. Repeating this calculation for the sixth cumulant, one would arrive to the same result multiplied by a factor of $\frac{\lambda}{C}\left(\frac{\alpha}{\sigma^2/n}\right)^2$ due to the general form of the even cumulants (Eq. F.20) and the fact that there an extra two $(\sigma^2/n)^{-1}\hat{\delta}g$'s.

### E.1.2 Self-consistent equation in the EK limit

Starting from the proportionality relations $\hat{\delta}g = \alpha g$ and $\Delta g = \alpha_\Delta g$, we can now write the self-consistent equation for the discrepancy as

$$\hat{\delta}g = (g - \Delta g) - q\frac{\lambda}{\lambda + \sigma_n^2}\left(g - \Delta g\right) \tag{E.5}$$

Dividing both sides by $g$ we get a scalar equation

$$\alpha = (1 - \alpha_\Delta) - q\frac{\lambda}{\lambda + \sigma_n^2}\left(1 - \alpha_\Delta\right) \tag{E.6}$$

$$= \frac{\lambda + \sigma_n^2}{\lambda + \sigma_n^2} - q\frac{\lambda}{\lambda + \sigma_n^2} + \left(q\frac{\lambda}{\lambda + \sigma_n^2} - 1\right)\alpha_\Delta$$

$$= \frac{\sigma_n^2}{\lambda + \sigma_n^2} + (1 - q)\frac{\lambda}{\lambda + \sigma_n^2} + \left(q\frac{\lambda}{\lambda + \sigma_n^2} - 1\right)\alpha_\Delta$$

The factor $\alpha_\Delta$ can be calculated by noticing that $\Delta g$ has the form of a geometric series. To better understand what follows next, the reader should first go over §F. The first term in this series is related to contracting the fourth cumulant $\kappa_4$ with three $\hat{\delta}g$'s thus yielding a factor of $\frac{\lambda^2}{C}\left(\frac{\alpha}{\sigma^2/n}\right)^3$ (recall that in the EK approximation we trade $\sigma^2 \rightarrow \sigma^2/n$). The ratio of two consecutive terms in this series is given by $\frac{\lambda}{C}\left(\frac{\alpha}{\sigma^2/n}\right)^2$. Using the formula for the sum of a geometric series we have

$$\alpha = \frac{\sigma^2/n}{\lambda + \sigma^2/n} + \frac{(1-q)\lambda}{\lambda + \sigma^2/n} + \left(q\frac{\lambda}{\lambda + \sigma^2/n} - 1\right)\frac{\lambda^2}{C}\left(\frac{\alpha}{\sigma^2/n}\right)^3 \left[1 - \frac{\lambda}{C}\left(\frac{\alpha}{\sigma^2/n}\right)^2\right]^{-1} \tag{E.7}$$

### E.2 Corrections to EK and estimation of the $q_{\text{train}}$ factor in the main text

The EK approximation can be improved systematically using the field-theory approach of Ref. [11] where the EK result is interpreted as the leading order contribution, in the large $n$ limit, to the

average of the GP predictor over many data-set draws from the dataset measure. However, that work focused on the test performance whereas for $q_{\text{train}}$ we require the performance on the training set. We briefly describe the main augmentations needed here and give the sub-leading and sub-sub-leading corrections to the EK result on the training set, enabling us to estimate $q_{\text{train}}$ analytically within a $16.3\%$ relative error compared with the empirical value. Further systematic improvements are possible but are left for future work.

We thus consider the quantity $\sum_\mu \varphi(\mathbf{x}_\mu) \bar{f}(\mathbf{x}_\mu)$ where $\mathbf{x}_\mu$ is drawn from the training set, $\bar{f}(\mathbf{x}_\mu)$ is the predictive mean of the GP on that specific training set, and $\varphi(\mathbf{x}_\mu)$ is some function which we will later take to be the target function ($\varphi(\mathbf{x}) = g(\mathbf{x})$). We wish to calculate the average of this quantity over all training set draws of size $n$. We begin by adding a source term of the form $J \sum_\mu \varphi(\mathbf{x}_\mu) f(\mathbf{x}_\mu)$ to the action and notice a similar term appearing in the GP action ($-\sum_\mu (f(\mathbf{x}_\mu) - g(\mathbf{x}_\mu))^2$) due to the MSE loss. Examining this extra term one notices that it can be absorbed as a $J$ dependent shift to the target on training set ($g(\mathbf{x}_\mu) \to g(\mathbf{x}_\mu) + \frac{J\sigma^2}{2} \varphi(\mathbf{x}_\mu)$) following which the analysis of Ref. [11] carries through straightforwardly. Doing so, the general result for the leading EK term and sub-leading correction are

$$n \int d\mu(\mathbf{x}) \varphi(\mathbf{x}) \langle f(\mathbf{x}) \rangle_{\text{EK}} - \frac{n}{\sigma^2} \int d\mu(\mathbf{x}) \varphi(\mathbf{x}) \left[ \text{Cov}(\mathbf{x}, \mathbf{x})(\langle f(\mathbf{x}) \rangle_{\text{EK}} - g(\mathbf{x})) \right] \qquad \text{(E.8)}$$

where $\text{Cov}(\mathbf{x}, \mathbf{x}) = \langle f(\mathbf{x}) f(\mathbf{x}) \rangle_{\text{EK}} - \langle f(\mathbf{x}) \rangle_{\text{EK}} \langle f(\mathbf{x}) \rangle_{\text{EK}}$, $\langle ... \rangle_{\text{EK}}$ means averaging with $Z_{\text{EK}}$ of Ref. [11], and $\langle f(\mathbf{x}) \rangle_{\text{EK}}$ is the EK prediction of the previous section, Eq. D.8.

Turning to the specific linear CNN toy model and carrying the above expansion up to an additional term leads to

$$\alpha_{\text{train}} \approx \alpha_{\text{EK}} \left( 1 - \frac{\alpha_{\text{EK}}}{\sigma^2} + \frac{3}{4} \frac{\alpha_{\text{EK}}^2}{\sigma^4} \right) \qquad \text{(E.9)}$$

$$\alpha_{\text{EK}} = \frac{\sigma^2/n}{\sigma^2/n + \lambda} = \frac{\sigma^2/n}{\sigma^2/n + (NS)^{-1}}$$

Considering for instance $n = 200, \sigma^2 = 1.0, N = 30$ and $S = 30$, we find $\alpha_{\text{EK}} = 0.818$ and so

$$\alpha_{\text{train}} \approx 0.559 \qquad \text{(E.10)}$$

recalling that $q_{\text{train}} = \frac{\lambda + \sigma^2/n}{\lambda} (1 - \alpha_{\text{train}})$ we have

$$q_{\text{train}} \approx 2.4255 \qquad \text{(E.11)}$$

whereas the empirical value here is $2.8995$.

# F  Cumulants for a two-layer linear CNN

In this section we explicitly derive the leading (fourth and sixth) cumulants of the toy model of §4.1, and arrive at the general formula for the even cumulant of arbitrary order.

## F.1  Fourth cumulant

### F.1.1  Fourth cumulant for a CNN with general activation function (averaging over the readout layer)

For a general activation, we have in our setting for a 2-layer CNN

$$f(\mathbf{x}^\mu) = \sum_{i=1}^{N} \sum_{c=1}^{C} a_{i,c} \phi(\mathbf{w}_c \cdot \tilde{\mathbf{x}}_i^\mu) =: \sum_{i=1}^{N} \sum_{c=1}^{C} a_{i,c} \phi_{i,c}^\mu \qquad \text{(F.1)}$$

The kernel is

$$K\left(\mathbf{x}^1, \mathbf{x}^2\right) = \left\langle f\left(\mathbf{x}^1\right) f\left(\mathbf{x}^2\right)\right\rangle \tag{F.2}$$

$$= \left\langle \sum_{i,i'=1}^{N} \sum_{c,c'=1}^{C} a_{i,c}\phi_{i,c}^1 a_{i',c'}\phi_{i',c'}^2 \right\rangle$$

$$= \sum_{i,i'=1}^{N} \sum_{c,c'=1}^{C} \underbrace{\left\langle a_{i,c}a_{i',c'}\right\rangle_a}_{\delta_{ii'}\delta_{cc'}\sigma_a^2/CN} \left\langle \phi_{i,c}^1 \phi_{i',c'}^2 \right\rangle_{\mathbf{w}}$$

$$= \frac{\sigma_a^2}{CN} \sum_{i=1}^{N} \sum_{c=1}^{C} \left\langle \phi_{i,c}^1 \phi_{i,c}^2 \right\rangle_{\mathbf{w}} = \frac{\sigma_a^2}{N} \sum_{i=1}^{N} \left\langle \phi_{i,c}^1 \phi_{i,c}^2 \right\rangle_{\mathbf{w}}$$

The fourth moment is

$$\left\langle f\left(\mathbf{x}^1\right) f\left(\mathbf{x}^2\right) f\left(\mathbf{x}^3\right) f\left(\mathbf{x}^4\right)\right\rangle_{\mathbf{a},\mathbf{w}} = \sum_{i_{1:4}} \sum_{c_{1:4}} \left\langle a_{i_1,c_1} a_{i_2,c_2} a_{i_3,c_3} a_{i_4,c_4}\right\rangle_{\mathbf{a}} \left\langle \phi_{i_1,c_1}^1 \phi_{i_2,c_2}^2 \phi_{i_3,c_3}^3 \phi_{i_4,c_4}^4 \right\rangle_{\mathbf{w}}$$

$$\tag{F.3}$$

Averaging over the last layer weights gives

$$\left\langle a_{i_1,c_1} a_{i_2,c_2} a_{i_3,c_3} a_{i_4,c_4}\right\rangle_{\mathbf{a}} = \left(\frac{\sigma_a^2}{CN}\right)^2 \left(\delta_{i_1 i_2}\delta_{c_1 c_2}\delta_{i_3 i_4}\delta_{c_3 c_4} + \{(13)\,(24) + (14)\,(23)\}\right) \tag{F.4}$$

So this will always make two pairs out of four $\phi$'s, each with the same $i, c$ indices. Notice that, regardless of the input indices, for different channels $c \neq c'$ we have

$$\left\langle \phi_{i,c}^\mu \phi_{i,c}^\nu \phi_{j,c'}^{\mu'} \phi_{j,c'}^{\nu'} \right\rangle_{\mathbf{w}} = \left\langle \phi_{i,c}^\mu \phi_{i,c}^\nu \right\rangle_{\mathbf{w}} \left\langle \phi_{j,c'}^{\mu'} \phi_{j,c'}^{\nu'} \right\rangle_{\mathbf{w}} \tag{F.5}$$

so, e.g. the first term out of three is

$$\sum_{i_{1:4}} \sum_{c_{1:4}} \delta_{i_1 i_2}\delta_{c_1 c_2}\delta_{i_3 i_4}\delta_{c_3 c_4} \left\langle \phi_{i_1,c_1}^1 \phi_{i_2,c_2}^2 \phi_{i_3,c_3}^3 \phi_{i_4,c_4}^4 \right\rangle_{\mathbf{w}} \tag{F.6}$$

$$= \sum_{i_1,i_3} \sum_{c_1,c_3} \left\langle \phi_{i_1,c_1}^1 \phi_{i_1,c_1}^2 \phi_{i_3,c_3}^3 \phi_{i_3,c_3}^4 \right\rangle_{\mathbf{w}}$$

$$= \sum_{i_1,i_3} \left\{ \sum_c \left\langle \phi_{i_1,c}^1 \phi_{i_1,c}^2 \phi_{i_3,c}^3 \phi_{i_3,c}^4 \right\rangle_{\mathbf{w}} + \sum_{\substack{c_1, c_3 \\ c_1 \neq c_3}} \left\langle \phi_{i_1,c_1}^1 \phi_{i_1,c_1}^2 \right\rangle_{\mathbf{w}} \left\langle \phi_{i_3,c_3}^3 \phi_{i_3,c_3}^4 \right\rangle_{\mathbf{w}} \right\}$$

where in the last line we separated the diagonal and off-diagonal terms in the channel indices. So

$$\left(\frac{\sigma_a^2}{CN}\right)^{-2} \left\langle f\left(\mathbf{x}^1\right) f\left(\mathbf{x}^2\right) f\left(\mathbf{x}^3\right) f\left(\mathbf{x}^4\right)\right\rangle_{\mathbf{a},\mathbf{w}} \tag{F.7}$$

$$= \sum_{i_1,i_2} \sum_c \left\{ \left\langle \phi_{i_1,c}^1 \phi_{i_1,c}^2 \phi_{i_2,c}^3 \phi_{i_2,c}^4 \right\rangle_{\mathbf{w}} + \left\langle \phi_{i_1,c}^1 \phi_{i_1,c}^3 \phi_{i_2,c}^2 \phi_{i_2,c}^4 \right\rangle_{\mathbf{w}} + \left\langle \phi_{i_1,c}^1 \phi_{i_1,c}^4 \phi_{i_2,c}^2 \phi_{i_2,c}^3 \right\rangle_{\mathbf{w}} \right\}$$

$$+ \sum_{i_1,i_2} \sum_{\substack{c_1, c_2 \\ c_1 \neq c_2}} \left\{ \left\langle \phi_{i_1,c_1}^1 \phi_{i_1,c_1}^2 \right\rangle_{\mathbf{w}} \left\langle \phi_{i_2,c_2}^3 \phi_{i_2,c_2}^4 \right\rangle_{\mathbf{w}} + \left\langle \phi_{i_1,c_1}^1 \phi_{i_1,c_1}^3 \right\rangle_{\mathbf{w}} \left\langle \phi_{i_2,c_2}^2 \phi_{i_2,c_2}^4 \right\rangle_{\mathbf{w}} + \left\langle \phi_{i_1,c_1}^1 \phi_{i_1,c_1}^4 \right\rangle_{\mathbf{w}} \left\langle \phi_{i_2,c_2}^2 \phi_{i_2,c_2}^3 \right\rangle_{\mathbf{w}} \right\}$$

On the other hand

$$\langle f^1 f^2 \rangle \langle f^3 f^4 \rangle \tag{F.8}$$

$$= \left( \frac{\sigma_a^2}{CN} \sum_{i=1}^{N} \sum_{c=1}^{C} \langle \phi_{i,c}^1 \phi_{i,c}^2 \rangle_{\mathbf{w}} \right) \left( \frac{\sigma_a^2}{CN} \sum_{i'=1}^{N} \sum_{c'=1}^{C} \langle \phi_{i',c'}^3 \phi_{i',c'}^4 \rangle_{\mathbf{w}} \right)$$

$$= \left( \frac{\sigma_a^2}{CN} \right)^2 \sum_{i,i'=1}^{N} \sum_{c,c'=1}^{C} \langle \phi_{i,c}^1 \phi_{i,c}^2 \rangle_{\mathbf{w}} \langle \phi_{i',c'}^3 \phi_{i',c'}^4 \rangle_{\mathbf{w}}$$

$$= \left( \frac{\sigma_a^2}{CN} \right)^2 \sum_{i,i'=1}^{N} \left\{ \sum_{\substack{c,\, c' = 1 \\ c \neq c'}}^{C} \langle \phi_{i,c}^1 \phi_{i,c}^2 \rangle_{\mathbf{w}} \langle \phi_{i',c'}^3 \phi_{i',c'}^4 \rangle_{\mathbf{w}} + \sum_{c=1}^{C} \langle \phi_{i,c}^1 \phi_{i,c}^2 \rangle_{\mathbf{w}} \langle \phi_{i',c}^3 \phi_{i',c}^4 \rangle_{\mathbf{w}} \right\}$$

Putting it all together, the off-diagonal terms in the channel indices cancel and we are left with

$$\left( \frac{\sigma_a^2}{CN} \right)^{-2} \kappa_4 \left( \mathbf{x}_1, \mathbf{x}_2, \mathbf{x}_3, \mathbf{x}_4 \right) \tag{F.9}$$

$$= \left( \frac{\sigma_a^2}{CN} \right)^{-2} \left( \langle f^1 f^2 f^3 f^4 \rangle - \left( \langle f^1 f^2 \rangle \langle f^3 f^4 \rangle + \langle f^1 f^3 \rangle \langle f^2 f^4 \rangle + \langle f^1 f^4 \rangle \langle f^2 f^3 \rangle \right) \right)$$

$$= \sum_{i_1,i_2} \sum_{c} \left\{ \langle \phi_{i_1,c}^1 \phi_{i_1,c}^2 \phi_{i_2,c}^3 \phi_{i_2,c}^4 \rangle_{\mathbf{w}} + \langle \phi_{i_1,c}^1 \phi_{i_1,c}^3 \phi_{i_2,c}^2 \phi_{i_2,c}^4 \rangle_{\mathbf{w}} + \langle \phi_{i_1,c}^1 \phi_{i_1,c}^4 \phi_{i_2,c}^2 \phi_{i_2,c}^3 \rangle_{\mathbf{w}} \right\}$$

$$- \sum_{i_1,i_2} \sum_{c} \left\{ \langle \phi_{i_1,c}^1 \phi_{i_1,c}^2 \rangle_{\mathbf{w}} \langle \phi_{i_2,c}^3 \phi_{i_2,c}^4 \rangle_{\mathbf{w}} + \langle \phi_{i_1,c}^1 \phi_{i_1,c}^3 \rangle_{\mathbf{w}} \langle \phi_{i_2,c}^2 \phi_{i_2,c}^4 \rangle_{\mathbf{w}} + \langle \phi_{i_1,c}^1 \phi_{i_1,c}^4 \rangle_{\mathbf{w}} \langle \phi_{i_2,c}^2 \phi_{i_2,c}^3 \rangle_{\mathbf{w}} \right\}$$

$$:= \sum_{i,j=1}^{N} \sum_{c=1}^{C} \left\{ \langle \phi_{i,c}^1 \phi_{i,c}^2 \phi_{j,c}^3 \phi_{j,c}^4 \rangle_{\mathbf{w}} - \langle \phi_{i,c}^1 \phi_{i,c}^2 \rangle_{\mathbf{w}} \langle \phi_{j,c}^3 \phi_{j,c}^4 \rangle_{\mathbf{w}} \right\} + [(1_i 3_i)(2_j 4_j) + (1_i 4_i)(2_j 3_j)]$$

where in the last line we introduced a short-hand notation to compactly keep track of the combinations of the indices.

### F.1.2 Fourth cumulant for linear CNN

Here, $\phi_{i,c}^\mu := \mathbf{w}_c \cdot \tilde{\mathbf{x}}_i^\mu = \sum_{s=1}^{S} w_s^{(c)} \tilde{x}_s^{(\mu,i)}$. The fourth moment is

$$\langle \phi_{i,c}^1 \phi_{i,c}^2 \phi_{j,c}^3 \phi_{j,c}^4 \rangle_{\mathbf{w}} \tag{F.10}$$

$$= \sum_{s_{1:4}=1}^{S} \left\langle \left( w_{s_1}^{(c)} \tilde{x}_{s_1}^{(1,i)} \right) \left( w_{s_2}^{(c)} \tilde{x}_{s_2}^{(2,i)} \right) \left( w_{s_3}^{(c)} \tilde{x}_{s_3}^{(3,j)} \right) \left( w_{s_4}^{(c)} \tilde{x}_{s_4}^{(4,j)} \right) \right\rangle_{\mathbf{w}}$$

$$= \sum_{s_{1:4}=1}^{S} \underbrace{\left\langle w_{s_1}^{(c)} w_{s_2}^{(c)} w_{s_3}^{(c)} w_{s_4}^{(c)} \right\rangle_{\mathbf{w}}}_{(\sigma_w^2/S)^2 \cdot \delta_{s_1 s_2} \delta_{s_3 s_4}[3]} \tilde{x}_{s_1}^{(1,i)} \tilde{x}_{s_2}^{(2,i)} \tilde{x}_{s_3}^{(3,j)} \tilde{x}_{s_4}^{(4,j)}$$

$$= \left( \frac{\sigma_w^2}{S} \right)^2 \sum_{s_{1:4}=1}^{S} \left( \delta_{s_1 s_2} \delta_{s_3 s_4} + \delta_{s_1 s_3} \delta_{s_2 s_4} + \delta_{s_1 s_4} \delta_{s_2 s_3} \right) \tilde{x}_{s_1}^{(1,i)} \tilde{x}_{s_2}^{(2,i)} \tilde{x}_{s_3}^{(3,j)} \tilde{x}_{s_4}^{(4,j)}$$

$$= \left( \frac{\sigma_w^2}{S} \right)^2 \left\{ \left( \tilde{\mathbf{x}}_i^1 \cdot \tilde{\mathbf{x}}_i^2 \right) \left( \tilde{\mathbf{x}}_j^3 \cdot \tilde{\mathbf{x}}_j^4 \right) + \left( \tilde{\mathbf{x}}_i^1 \cdot \tilde{\mathbf{x}}_j^3 \right) \left( \tilde{\mathbf{x}}_i^2 \cdot \tilde{\mathbf{x}}_j^4 \right) + \left( \tilde{\mathbf{x}}_i^1 \cdot \tilde{\mathbf{x}}_j^4 \right) \left( \tilde{\mathbf{x}}_i^2 \cdot \tilde{\mathbf{x}}_j^3 \right) \right\}$$

$$:= \left( \frac{\sigma_w^2}{S} \right)^2 \left\{ (1_i 2_i)(3_j 4_j) + (1_i 3_j)(2_i 4_j) + (1_i 4_j)(2_i 3_j) \right\}$$

Similarly

$$\left(\frac{\sigma_w^2}{S}\right)^{-2}\left\langle\phi_{i,c}^1\phi_{i,c}^3\phi_{j,c}^2\phi_{j,c}^4\right\rangle_{\mathbf{w}} = (1_i3_i)\,(2_j4_j) + (1_i2_j)\,(3_i4_j) + (1_i4_j)\,(3_i2_j) \tag{F.11}$$

$$\left(\frac{\sigma_w^2}{S}\right)^{-2}\left\langle\phi_{i,c}^1\phi_{i,c}^4\phi_{j,c}^2\phi_{j,c}^3\right\rangle_{\mathbf{w}} = (1_i4_i)\,(3_j2_j) + (1_i3_j)\,(4_i2_j) + (1_i2_j)\,(4_i3_j)$$

Notice that the 2nd and 3rd terms have $(ij)\,(ij)$ while the first term has $(ii)\,(jj)$. The latter will cancel out with the $\left\langle\phi_{i,c}^\mu\phi_{i,c}^\nu\right\rangle_{\mathbf{w}}\left\langle\phi_{j,c}^{\mu'}\phi_{j,c}^{\nu'}\right\rangle_{\mathbf{w}}$ terms. Thus

$$[\cancel{(1_i2_i)\,(3_j4_j)} + (1_i3_j)\,(2_i4_j) + (1_i4_j)\,(2_i3_j)] \tag{F.12}$$
$$+ [\cancel{(1_i3_i)\,(2_j4_j)} + (1_i2_j)\,(3_i4_j) + (1_i4_j)\,(3_i2_j)]$$
$$+ [\cancel{(1_i4_i)\,(3_j2_j)} + (1_i3_j)\,(4_i2_j) + (1_i2_j)\,(4_i3_j)]$$
$$- [\cancel{(1_i2_i)\,(3_j4_j)} + \cancel{(1_i3_i)\,(2_j4_j)} + \cancel{(1_i4_i)\,(3_j2_j)}]$$
$$= (1_i3_j)\,(2_i4_j) + (1_i4_j)\,(2_i3_j) + (1_i2_j)\,(3_i4_j) + (1_i4_j)\,(3_i2_j) + (1_i3_j)\,(4_i2_j) + (1_i2_j)\,(4_i3_j)$$
$$= (1_i3_j)\,[(2_i4_j) + (4_i2_j)] + (1_i4_j)\,[(2_i3_j) + (3_i2_j)] + (1_i2_j)\,[(3_i4_j) + (4_i3_j)]$$

Denote $\lambda := \frac{\sigma_a^2}{N}\frac{\sigma_w^2}{S}$ The fourth cumulant is

$$\kappa_4\,(\mathbf{x}_1,\mathbf{x}_2,\mathbf{x}_3,\mathbf{x}_4) \tag{F.13}$$
$$= \frac{\lambda^2}{C}\sum_{i,j=1}^N\{(1_i2_j)\,[(3_i4_j) + (4_i3_j)] + (1_i3_j)\,[(2_i4_j) + (4_i2_j)] + (1_i4_j)\,[(2_i3_j) + (3_i2_j)]\}$$

Notice that all terms involve inner products between $\tilde{\mathbf{x}}$'s with *different indices* $i,j$, i.e. mixing different convolutional windows. This means that $\kappa_4$, and also all higher order cumulants, *cannot* be written in terms of the linear kernel, which does not mix different conv-window indices. This is in contrast to the kernel (second cumulant) of this linear CNN which is identical to that of a corresponding linear fully connected network (FCN): $K\,(\mathbf{x},\mathbf{x}') = \frac{\sigma_a^2\sigma_w^2}{NS}\mathbf{x}^\mathsf{T}\mathbf{x}'$ It is also in contrast to the higher cumulants of the corresponding linear FCN, where all cumulants can be expressed in terms of products of the linear kernel.

### F.2 Sixth cumulant and above

The even moments in terms of cumulants for a vector valued RV with zero odd moments and cumulants are (see [31]):

$$\kappa^{\mu_1\mu_2} = \kappa^{\mu_1,\mu_2} \tag{F.14}$$
$$\kappa^{\mu_1\mu_2\mu_3\mu_4} = \kappa^{\mu_1,\mu_2,\mu_3,\mu_4} + \kappa^{\mu_1,\mu_2}\kappa^{\mu_3,\mu_4}\,[3]$$
$$\kappa^{\mu_1\mu_2\mu_3\mu_4\mu_5\mu_6} = \kappa^{\mu_1,\mu_2,\mu_3,\mu_4,\mu_5,\mu_6} + \kappa^{\mu_1,\mu_2,\mu_3,\mu_4}\kappa^{\mu_5,\mu_6}\,[15] + \kappa^{\mu_1,\mu_2}\kappa^{\mu_3,\mu_4}\kappa^{\mu_5,\mu_6}\,[15]$$

where the moments are on the l.h.s. (indices with no commas) and the cumulants are on the r.h.s. (indices are separated with commas). Thus, the sixth cumulant is

$$\kappa^{\mu_1,\mu_2,\mu_3,\mu_4,\mu_5,\mu_6} = \kappa^{\mu_1\mu_2\mu_3\mu_4\mu_5\mu_6} - \kappa^{\mu_1,\mu_2,\mu_3,\mu_4}\kappa^{\mu_5,\mu_6}\,[15] - \kappa^{\mu_1,\mu_2}\kappa^{\mu_3,\mu_4}\kappa^{\mu_5,\mu_6}\,[15] \tag{F.15}$$

In the linear case, the analogue of $\kappa^{\mu_1,\mu_2}\kappa^{\mu_3,\mu_4}\kappa^{\mu_5,\mu_6}$ is (15 such pairings, where only the numbers "move", not the $i,j,k$)

$$\frac{1}{\lambda^3}K\,(\mathbf{x}_1,\mathbf{x}_2)\,K\,(\mathbf{x}_3,\mathbf{x}_4)\,K\,(\mathbf{x}_5,\mathbf{x}_6) = (1_i2_i)\,(3_j4_j)\,(5_k6_k) \tag{F.16}$$

and the analogue of $\kappa^{\mu_1,\mu_2}\kappa^{\mu_3,\mu_4}\kappa^{\mu_5,\mu_6}$ is

$$\frac{C}{\lambda^3}\kappa_4\,(\mathbf{x}_1,\mathbf{x}_2,\mathbf{x}_3,\mathbf{x}_4)\,K\,(\mathbf{x}_5,\mathbf{x}_6) \tag{F.17}$$
$$= \{(1_i2_j)\,[(3_i4_j) + (4_i3_j)] + (1_i3_j)\,[(2_i4_j) + (4_i2_j)] + (1_i4_j)\,[(2_i3_j) + (3_i2_j)]\}\,(5_k6_k)$$
$$= \{(1_i2_j)\,(3_i4_j) + (1_i2_j)\,(4_i3_j) + (1_i3_j)\,(2_i4_j) + (1_i3_j)\,(4_i2_j) + (1_i4_j)\,(2_i3_j) + (1_i4_j)\,(3_i2_j)\}\,(5_k6_k)$$

Below, we found the 6th moment for a linear CNN to be

$$\left\langle \phi_{i,c}^1 \phi_{i,c}^2 \phi_{j,c}^3 \phi_{j,c}^4 \phi_{k,c}^5 \phi_{k,c}^6 \right\rangle \tag{F.18}$$

$$= (1_i 2_i)(3_j 4_j)(5_k 6_k) + (1_i 3_j)(2_i 4_j)(5_k 6_k) + (1_i 4_j)(2_i 3_j)(5_k 6_k)$$
$$+ (1_i 2_i)(3_j 5_k)(4_j 6_k) + (1_i 3_j)(2_i 5_k)(4_j 6_k) + (1_i 5_k)(2_i 3_j)(4_j 6_k)$$
$$+ (1_i 2_i)(4_j 5_k)(3_j 6_k) + (1_i 5_k)(2_i 4_j)(3_j 6_k) + (1_i 4_j)(2_i 5_k)(3_j 6_k)$$
$$+ (1_i 5_k)(3_j 4_j)(2_i 6_k) + (1_i 3_j)(4_j 5_k)(2_i 6_k) + (1_i 4_j)(3_j 5_k)(2_i 6_k)$$
$$+ (2_i 5_k)(3_j 4_j)(1_i 6_k) + (3_j 5_k)(2_i 4_j)(1_i 6_k) + (4_j 5_k)(2_i 3_j)(1_i 6_k)$$

Notice that for every blue term we have exactly 6 red terms, so all of the colored terms will exactly cancel out and only the uncolored terms will survive. There are 8 such uncolored terms for each one of the 15 pairings, thus we will ultimately have 120 such pairs, thus the sixth cumulant is

$$\kappa_6 \left( \mathbf{x}_1, \ldots, \mathbf{x}_6 \right) = \frac{\lambda^3}{C'^2} \sum_{i,j,k=1}^N \left( \bullet_i \bullet_j \right) \left( \bullet_i \bullet_k \right) \left( \bullet_j \bullet_k \right) [120] \tag{F.19}$$

where the $[120]$ stands for the number of ways to pair the numbers $\{1, ..., 6\}$ into the form $\left( \bullet_i \bullet_j \right) \left( \bullet_i \bullet_k \right) \left( \bullet_j \bullet_k \right)$.

We can thus identify a pattern which holds for any even cumulant of arbitrary order $2m$:

$$\kappa_{2m} \left( \mathbf{x}_1, \ldots, \mathbf{x}_{2m} \right) = \frac{\lambda^m}{C^{m-1}} \sum_{i_1, \ldots, i_m=1}^N \left( \bullet_{i_1}, \bullet_{i_2} \right) \cdots \left( \bullet_{i_{m-2}}, \bullet_{i_{m-1}} \right) \left( \bullet_{i_{m-1}}, \bullet_{i_m} \right) \cdots [(2m-1)!]$$
$$\tag{F.20}$$

where the indices $i_1, \ldots, i_m$ obey the following:

1. Each index appears exactly twice in each summand.

2. Each index cannot be paired with itself, i.e. $\left( \bullet_{i_1}, \bullet_{i_1} \right)$ is not allowed.

3. The same pairing can appear more than once, e.g. $(1_i 2_j)(3_i 4_j)(5_k 6_\ell)(7_k 8_\ell)$ is OK, in that $i, j$ are paired together twice, and so are $k, \ell$.

# G   Feature learning phase transition

## G.1   Field theory derivation of the statistics of the hidden weights covariance

Although our main focus was on the statistics of the DNN outputs, our function-space formalism can also be used to characterize the statistics of the weights of the intermediate hidden layers. Here we focus on the linear CNN toy model given in the main text, where the learnable parameters of the student are given by $\theta = \{w_{c,s}, a_{i,c}\}$. Consider first a prior distribution in output space, where throughout this section we denote: $\vec{f} \equiv (f_1, \ldots, f_n)$, i.e. the vector of outputs on the training set alone (without the test point). Since we are interested in the statistics of the hidden weights, we will introduce an appropriate source term in weight space $J_{c,s}$

$$P_0 \left[ \vec{f}, \{J_{c,s}\} \right] \propto \int dw \int da \, \exp \left( -\frac{1}{2\sigma_w^2} \sum_{c,s} \left( w_{c,s} - \sigma_w^2 J_{c,s} \right)^2 \right) P_0(a) \prod_{\mu=1}^n \delta \left( f_\mu - z_{\theta,\mu} \right)$$
$$\tag{G.1}$$

where $z_{\theta,\mu}$ is the of output of the CNN parameterized by $\theta$ on the $\mu$'th training point. Given some loss function $\mathcal{L}$, the posterior is given by

$$P \left[ \vec{f}, \{J_{c,s}\} \right] = P_0 \left[ \vec{f}, \{J_{c,s}\} \right] e^{-\mathcal{L}/\sigma^2} \tag{G.2}$$

The posterior mean of the hidden weights is thus

$$\partial_{J_{c,s}} \log \left( \int_{\mathbb{R}^n} d\vec{f} P \left[ \vec{f}, \{J_{c,s}\} \right] \right) \bigg|_{J=0} = \langle w_{c,s} \rangle_{P[\vec{f}, \{J_{c,s}\}]} - \sigma_w^2 J_{c,s} \bigg|_{J=0} = \langle w_{c,s} \rangle_{P[\vec{f}, \{J_{c,s}\}]}$$
$$\tag{G.3}$$

and the posterior covariance can be extracted from taking the second derivative, namely

$$\partial_{J_{c_1,s_1}}\partial_{J_{c_2,s_2}}\log\left(\int_{\mathbb{R}^n}d\vec{f}P\left[\vec{f},\{J_{c,s}\}\right]\right)\Bigg|_{J=0} \tag{G.4}$$

$$= \langle w_{c_1,s_1}w_{c_2,s_2}\rangle_{P[\vec{f},\{J_{c,s}\}]} + \sigma_w^4 J_{c_1,s_1}J_{c_2,s_2}\Bigg|_{J=0} - \sigma_w^2\delta_{s_1s_2}\delta_{c_1c_2}$$

$$= \langle w_{c_1,s_1}w_{c_2,s_2}\rangle - \sigma_w^2\delta_{s_1s_2}\delta_{c_1c_2}$$

Our next task is to rewrite these expectation values over weights under the posterior as expectation values of DNN training outputs $(f(\mathbf{x}_\mu))$ under the posterior. To this end we write down the kernel of this simple CNN such that it depends on the source terms:

$$K_J(\mathbf{x},\mathbf{x}') = \sum_{i,i',c,c',s,s'}\langle a_{i,c}w_{c,s}\tilde{x}_{i,s}a_{i',c'}w_{c',s'}\tilde{x}'_{i',s'}\rangle_{P[\vec{f},J_{c,s}]} \tag{G.5}$$

$$= \underbrace{\sum_{i,i',c,c'}\langle a_{i,c}a_{i',c'}\rangle_a}_{\delta_{ii'}\delta_{cc'}/CN}\sum_{s,s'}\langle w_{c,s}\tilde{x}_{i,s}w_{c',s'}\tilde{x}'_{i',s'}\rangle_{w,J}$$

$$= \frac{1}{CN}\sum_{i,c}\sum_{s,s'}\langle w_{c,s}\tilde{x}_{i,s}w_{c,s'}\tilde{x}'_{i,s'}\rangle_{w,J}$$

$$= \frac{1}{N}\sum_i\sum_{s,s'}\frac{1}{C}\sum_c\underbrace{\langle w_{c,s}w_{c,s'}\rangle_{w,J}}_{(1/S)\delta_{ss'}+(1/S^2)J_{cs}J_{cs'}}\tilde{x}_{i,s}\tilde{x}'_{i,s'}$$

$$= \frac{1}{NS}\frac{1}{C}\sum_i\sum_s\sum_c\tilde{x}_{i,s}\tilde{x}'_{i,s} + \frac{1}{NS^2}\sum_i\sum_{s,s'}\underbrace{\left(\frac{1}{C}\sum_c J_{cs}J_{cs'}\right)}_{\equiv B_{ss'}}\tilde{x}_{i,s}\tilde{x}'_{i,s'}$$

$$= \frac{1}{NS}\mathbf{x}^\mathsf{T}\mathbf{x}' + \frac{1}{NS^2}\sum_i\tilde{\mathbf{x}}_i^\mathsf{T}B\tilde{\mathbf{x}}'_i$$

where $B \in \mathbb{R}^{S\times S}$. This can be written as $(d = NS)$

$$K_J(\mathbf{x},\mathbf{x}') = \frac{1}{NS}\mathbf{x}^\mathsf{T}\left(I_d + \frac{1}{S}\begin{pmatrix}B & & \\ & \ddots & \\ & & B\end{pmatrix}\right)\mathbf{x}' \tag{G.6}$$

We can now write the second mixed derivatives of $K_J$ to leading order in $J$ as

$$-\partial_{J_{c_1,s_1}}\partial_{J_{c_2,s_2}}K_J^{-1}(\mathbf{x},\mathbf{x}') = -NS\mathbf{x}^\mathsf{T}\left[\partial_{J_{c_1,s_1}}\partial_{J_{c_2,s_2}}\left(I_d + \frac{1}{S}\begin{pmatrix}B & & \\ & \ddots & \\ & & B\end{pmatrix}\right)^{-1}\right]\mathbf{x}'$$
$$\tag{G.7}$$

$$= -NS\mathbf{x}^\mathsf{T}\left[\partial_{J_{c_1,s_1}}\partial_{J_{c_2,s_2}}\left(I_d - \frac{1}{S}\begin{pmatrix}B & & \\ & \ddots & \\ & & B\end{pmatrix}\right)\right]\mathbf{x}'$$

$$= \frac{N}{C}\sum_i\sum_{s,s'}\tilde{x}_{i,s}\tilde{x}'_{i,s'}\partial_{J_{c_1,s_1}}\partial_{J_{c_2,s_2}}\sum_c J_{cs}J_{cs'}$$

$$= \frac{N}{C}\delta_{c_1c_2}\sum_i\sum_{s,s'}\tilde{x}_{i,s}\tilde{x}'_{i,s'}\left(\delta_{ss_2}\delta_{s_1s'} + \delta_{s's_2}\delta_{s_1s}\right)$$

$$= 2\frac{N}{C}\delta_{c_1c_2}\sum_i\tilde{x}_{i,s_1}\tilde{x}'_{i,s_2}$$

Next we take the large $C$ limit and thus have a posterior of the form $P[\vec{f}, J] = P_0[\vec{f}, J]e^{-\mathcal{L}/\sigma^2}$ where $P_0[\vec{f}]$ contains only $K_J^{-1}$ and none of the higher cumulants. Having the derivatives of $K_J^{-1}$ w.r.t. $J$ we can proceed in analyzing the derivatives of the log-partition function for the posterior w.r.t $J$. In particular the covariance matrix of the weights averaged over the different channels is

$$\frac{1}{C} \sum_{c_1,c_2} \partial_{J_{c_1,s_1}} \partial_{J_{c_2,s_2}} \log\left(\int_{\mathbb{R}^n} d\vec{f} P\left[\vec{f}, J\right]\right) \tag{G.8}$$

$$= -\frac{1}{C} \sum_{\mu,\nu=1}^n \left\{\sum_{c_1,c_2} [\partial_{J_{c_1,s_1}} \partial_{J_{c_2,s_2}} K_J^{-1}(\mathbf{x}_\mu, \mathbf{x}_\nu)] \frac{\int_{\mathbb{R}^n} d\vec{f} P[\vec{f}] f(\mathbf{x}_\mu) f(\mathbf{x}_\nu)}{\int_{\mathbb{R}^n} d\vec{f} P[\vec{f}]}\right\}$$

$$= 2\frac{N}{C^2} \sum_i \sum_{\mu,\nu=1}^n \tilde{x}_{i,s_1}^\mu \tilde{x}_{i,s_2}^\nu \frac{\int_{\mathbb{R}^n} d\vec{f} P[\vec{f}] f(\mathbf{x}_\mu) f(\mathbf{x}_\nu)}{\int_{\mathbb{R}^n} d\vec{f} P[\vec{f}]} \sum_{c_1,c_2} \delta_{c_1 c_2}$$

$$= 2\frac{N}{C} \sum_i \sum_{\mu,\nu=1}^n \tilde{x}_{i,s_1}^\mu \tilde{x}_{i,s_2}^\nu \frac{\int_{\mathbb{R}^n} d\vec{f} P[\vec{f}] f(\mathbf{x}_\mu) f(\mathbf{x}_\nu)}{\int_{\mathbb{R}^n} d\vec{f} P[\vec{f}]}$$

The above result is one of the two main points of this appendix: we established a mapping between expectation values over outputs and expectation values over hidden weights. Such a mapping can in principle be extended to any DNN. On the technical level, it requires the ability to calculate the cumulants as a function of the source terms, $J$. As we argue below, it may very well be that unlike in the main text, only a few cumulants are needed here.

To estimate the above expectation values we use the EK limit, where the sums over the training set are replaced by integrals over the measure $\mu(\mathbf{x})$, the $f$'s are replaced as $f(\mathbf{x}_\mu) \to \frac{\lambda}{\lambda+\sigma^2/n} g(\mathbf{x})$ and we assume the input distribution is normalized as $\int d\mu(\mathbf{x}) x_i x_j = \delta_{ij}$. Following this we find

$$2\frac{N}{C}\left(\frac{\lambda}{\lambda+\sigma^2/n}\right)^2 \int d\mu(\mathbf{x}) d\mu(\mathbf{x}') g(\mathbf{x}) g(\mathbf{x}') \sum_i \tilde{x}_{i,s_1} \tilde{x}'_{i,s_2} \tag{G.9}$$

$$= 2\frac{N}{C}\left(\frac{\lambda}{\lambda+\sigma^2/n}\right)^2 \sum_i \underbrace{\int d\mu(\mathbf{x}) g(\mathbf{x}) \tilde{x}_{i,s_1}}_{a_i^* w_{s_1}^*} \underbrace{\int d\mu(\mathbf{x}') g(\mathbf{x}') \tilde{x}'_{i,s_2}}_{a_i^* w_{s_2}^*}$$

$$= 2\frac{N}{C}\left(\frac{\lambda}{\lambda+\sigma^2/n}\right)^2 \underbrace{\sum_i (a_i^*)^2}_{1} w_{s_1}^* w_{s_2}^*$$

$$= 2\frac{N}{C}\left(\frac{\lambda}{\lambda+\sigma^2/n}\right)^2 w_{s_1}^* w_{s_2}^*$$

Comparing this to our earlier result for the covariance Eq. G.4 we get

$$2\frac{N}{C}\left(\frac{\lambda}{\lambda+\sigma^2/n}\right)^2 w_{s_1}^* w_{s_2}^* = \frac{1}{C}\sum_{c_1,c_2}\left(\langle w_{c_1,s_1} w_{c_2,s_2}\rangle - \sigma_w^2 \delta_{s_1 s_2} \delta_{c_1 c_2}\right) \tag{G.10}$$

$$= \frac{1}{C}\sum_{c_1,c_2} \langle w_{c_1,s_1} w_{c_2,s_2}\rangle - \sigma_w^2 \delta_{s_1 s_2}$$

Multiplying by $S = 1/\sigma_w^2$ and recalling that $\lambda = 1/NS$ we get

$$\langle [\Sigma_W]_{s_1 s_2}\rangle = \delta_{s_1 s_2} + \frac{2}{C}\frac{\lambda}{(\lambda+\sigma^2/n)^2} w_{s_1}^* w_{s_2}^* + O\left(1/C^2\right) \tag{G.11}$$

Repeating similar steps while also taking into account diagonal fluctuations yields another factor of $\left(\frac{1}{\lambda}+\frac{n}{\sigma^2}\right)^{-1}$ on the diagonal, thus arriving at the result as it appears in the main text:

$$\langle [\Sigma_W]_{s_1 s_2}\rangle = \left(1+\left(\frac{1}{\lambda}+\frac{n}{\sigma^2}\right)^{-1}\right)\delta_{s_1 s_2} + \frac{2}{C}\frac{\lambda}{(\lambda+\sigma^2/n)^2} w_{s_1}^* w_{s_2}^* + O\left(1/C^2\right) \tag{G.12}$$

The above results capture the leading order correction in $1/C$ to the weights covariance matrix. However the careful reader may be wary of the fact that the results in the main text require $1/C$ corrections to all orders and so it is potentially inadequate to use such a low order expansion deep in the feature learning regime, as we do in the main text. Here we note that not all DNN quantities need to have the same dependence on $C$. In particular it was shown in Ref. [26], that the weight's low order statistics is only weakly affected by finite-width corrections whereas the output covariance matrix is strongly affected by these. We conjecture that this is the case here and that only the cumulative effect of many weights, as reflected in the output of the DNN, requires strong $1/C$ corrections.

This conjecture can be verified analytically by repeating the above procedure on the full prior (i.e. the one that contains all cumulants), obtaining the operator in terms of $f$'s corresponding the weight's covariance matrix, and calculating its average with respect to the saddle-point theory. We leave this for future work.

### G.2   A surrogate quantity for the outlier

Since we used moderate $S$ values in our simulations (to maintain a reasonable compute time), we aggregated the eigenvalues of many instances of $\Sigma_W$ across training time and across noise realizations. Although the empirical histogram of the spectrum of $\Sigma_W$ agrees very well with the theoretical MP distribution (solid smooth curves in Fig. 2A), there is a substantial difference between the two at the right edge of the support $\lambda_+$, where the empirical histogram has a tail due to finite size effects. Thus it is hard to characterize the phase transition using the largest eigenvalue $\lambda_{\max}$ averaged across realizations. Instead, we use the quantity $\mathcal{Q} \equiv \mathbf{w}^{*\mathsf{T}}\Sigma_W\mathbf{w}^*$ as a surrogate which coincides with $\lambda_{\max}$ for $C \ll C_{\mathrm{crit}}$ but behaves sensibly on both sides of $C_{\mathrm{crit}}$, thus allowing to characterize the phase transition.

## H   Further details on the numerical experiments

### H.1   Additional details for the CNN toy model

In our experiments, we used the following hyper-parameter values. Learning rates of $\eta = 10^{-6}, 3 \cdot 10^{-7}$ which yield results with no appreciable difference in almost all cases, when we scale the amount of statistics collected (training epochs after reaching equilibrium) so that both $\eta$ values have the same amount of re-scaled training time: we used 10 training seeds for $\eta = 10^{-6}$ and 30 for $\eta = 3 \cdot 10^{-7}$. We used a gradient noise level of $\sigma^2 = 1.0$, but also checked for $\sigma^2 \in \{0.1, 0.01\}$ and got qualitatively similar results to those reported in the main text.

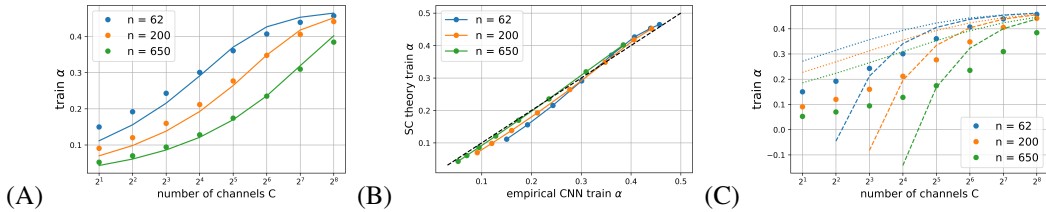

Figure 3:   **(A)** The CNNs' cosine distance $\alpha$, defined by $\langle f \rangle = (1 - \alpha)g$ between the ensemble-averaged prediction $\langle f \rangle$ and ground truth $g$ plotted vs. number of channels $C$ for the training set (for the test set see Fig. 1 in the main text). As $n$ increases, the solution of the self-consistent equation 16 (solid line) yields an increasingly accurate prediction of these empirical values (dots). **(B)** Same data as in (A), presented as empirical $\alpha$ vs. predicted $\alpha$. As $n$ grows, the two converge to the identity line (dashed black line). Solid lines connecting the dots here are merely for visualization purposes. **(C)** The theoretical predictions of the self-consistent theory but using only the fourth cumulant (dotted lines), and the predictions of perturbation theory to order $1/C$ (dashed lines, truncated to avoid large negative values in the figure).

In the main text and here we do not show error bars for $\alpha$ as these are too small to be appreciated visually. They are smaller than the mean values by approximately two orders of magnitude. The error bars were found by computing the empirical standard deviation of $\alpha$ across training dynamics and training seeds.

## H.2 Convergence of the training protocol to GP for the toy CNN model

In Fig. 4 we plot the MSE between the outputs of the trained CNNs and the predictions of the corresponding GP. We see that as $C$ becomes large the slope of the MSE tends to $-2.0$ indicating the $O(1/C)$ scaling of the leading corrections to the GP. This illustrates where we enter the perturbative regime of GP, and we see that this happens for larger $C$ as we increase the conv-kernel size $S$, since this also increases the input dimension $d = NS$. Thus it takes larger $C$ to enter the highly over-parameterized regime.

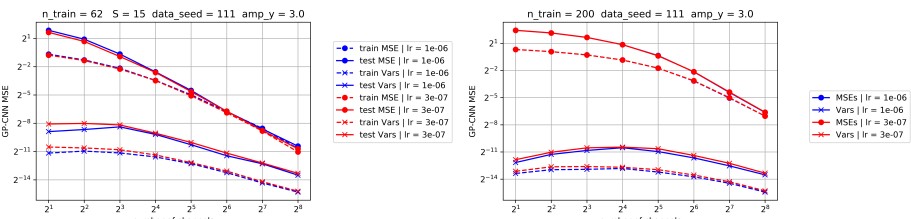

Figure 4: CNN-GP MSEs for different $S$, indicating where the perturbative regime starts (slope approaching $-2.0$). For $S = 15$ this happens around $C = 2^5$ whereas for $S = 30$ this happens around $C = 2^7$.

## H.3 Additional details for the Myrtle5 CNN experiment

In this subsection we give additional details on the results mentioned in §4.3. We trained the Myrtle-5 CNN [41] with our training protocol on subsets ($n \in \{16, 32, 64\}$) of CIFAR-10. We used a training noise level of $\sigma^2 = 0.005$ and learning rates $\eta \in [2 \cdot 10^{-5}, 2 \cdot 10^{-7}]$ with un-averaged (i.e. `sum` flag in PyTorch) MSE loss. Thus these learning rates correspond to learning rates higher by a factor of $n$ times 10 (the number of categories in CIFAR-10) when training with standard MSE (i.e. `mean` flag in PyTorch). Training was done either with a constant learning rate or in some cases with an initial phase lasting for $10^5$ epochs with 20-times the final learning rate used to sample from equilibrium. The overall number of epochs was on the scale of $10^6$ and the outputs were recorded every $100 - 400$ epochs on the train and test sets.

To provide approximate samples from the posterior distribution that are approximately uncorrelated, several steps were taken. First, we discarded a burn-in phase where the train loss relaxes to its minimal values (to within $2 - 3\%$ from its initial value) by judging where the train-loss saturates. Second, we calculate the auto-correlation of all the outputs after that burn-in phase which roughly decayed exponentially and calculate the auto-correlation time (ACT) of that decay in terms of epochs. We then down-sampled the recorded outputs on intervals corresponding to that ACT, typically obtaining between 100 to 300 samples. To verify that the learning rate is sufficiently small we first examined the training loss, calculated its variance, and made sure that it is much smaller than the mean. We then compared the train and test loss of two small learning rates differing by a factor of two and reported results where the change is less than $10\%$. We verified that such changes had minor effects on the non-normality measures relative to the statistical error.

Table 1 summarizes our results. The columns of the table are as follows:

- **CNN-GP train and test (the $\Delta$-Train and $\Delta$-Test columns)-** the MSE between the predictions of the finite Myrtle-5 CNN and the corresponding GP normalized by the $L2$ norm of the target function (the one-hot encoding of the categorical labels). The corresponding GP was calculated using the Neural Tangents library [35], using an NNGP kernel.

- **acc of ave -** the full test accuracy of the predictions of the CNN outputs after averaging these outputs over the equilibrium dynamics.

- **ave of acc -** first compute the full test accuracy of the predictions of the CNN outputs for each time point, then average over the equilibrium dynamics.

- **Non-normality measures -** $\kappa_4^{\text{CNN}}$ was obtained by taking each of the $10n$ time series produced by the dynamics, down-sampling them according to the ACTs, and for each calculating its variance and 4th cumulant and dividing the latter by the variance to the power

of 2. The reported $\kappa_4^{\text{CNN}}$ is the square root of the sum of squares of this resulting list of $10n$ normalized 4th cumulants. Turning to $\kappa_4^{\text{PCA}}$ it is the same as the previous quantity only instead of generating a time series to each data-point and label — replacing the data-points with their projections on the eigenvectors of the posterior covariance of the associated Gaussian Process. This is meant to pick up potential correlations between data-points. Last, $\kappa_4^{\text{Gauss}}$ is obtained by repeating the first process, for the same amount of ACTs and same number of series ($10n$), however with Gaussian uncorrelated random variables of unit standard deviation.

- **ACTs -** The number of ACTs that fit in the number of epochs after the burn-in phase.

Table 1: Myrtle5 CNN on CIFAR-10: Normality of fluctuations and departure from GP predictions. Error bars are within 20% of the reported values for $\Delta$-Train and $\Delta$-Test and 10% for $\kappa_4^{\text{CNN}}$ and $\kappa_4^{\text{PCA}}$ whereas $\kappa_4^{\text{Gauss}}$ has negligible error bars.

| $n$ | $C$ | $\Delta$-Train | $\Delta$-Test | acc of ave | ave of acc | $\kappa_4^{\text{CNN}}$ | $\kappa_4^{\text{PCA}}$ | $\kappa_4^{\text{Gauss}}$ | ACTs |
|-----|------|--------|--------|--------|--------|-------|-------|-------|-----|
| 16 | 1024 | 0.004 | 0.006 | - | - | 0.356 | 0.371 | 0.356 | 202 |
| 32 | 32 | 0.400 | 0.564 | 19.10% | 18.28% | 0.371 | 0.375 | 0.356 | 209 |
| 32 | 256 | 0.128 | 0.101 | 18.10% | 17.82% | 0.532 | 0.442 | 0.503 | 103 |
| 32 | 512 | 0.033 | 0.038 | 17.67% | 17.74% | 0.309 | 0.280 | 0.302 | 288 |
| 64 | 512 | 0.079 | 0.093 | 21.96% | 21.89% | 0.395 | 0.380 | 0.448 | 135 |

Looking at Table 1, we can make several observations:

- As expected, for $n = 16$ and $C = 1024$ we see a very small difference between the predictions of the finite CNN and those of the corresponding GP (small $\Delta$-Train and $\Delta$-Test). Not surprisingly, the non-normality measures indicate that the fluctuations are very close to being Gaussian: $\kappa_4$ of the CNN and that of a Gaussian differ only at the 5th decimal point (not shown). To get a sense of the scale of this measure, consider deforming a Gaussian distribution by applying the mapping $x \mapsto \sin(ax)/a$ with $a = 0.426$ to a standard Gaussian random variable. After this mapping, for $x$ that equals the standard deviation we would have a deformation of 3%, resulting in a value of $\kappa_4^{\text{CNN}} = 0.53$. Hence, this measure is quite sensitive to deviations from normality.

- For the three $n = 32$ experiments, we see that as $C$ decreases the finite CNN predictions grow further apart from those of the corresponding GP but the non-normality measure ($\kappa_4^{\text{CNN}}/\kappa_4^{\text{GP}}$) stays roughly the same and close to 1.0, indicating that the fluctuations remain close to being Gaussian even for small $C$. This is consistent with our description of finite-width CNNs as shifted GPs, since this description is a result of a saddle-point approximation and the latter holds when the fluctuations are approximately Gaussian.

- The test performance of our training protocol for these tiny training sets is comparable and in fact better than those reported in [41], who used standard mini-batch SGD training ($11.83\% \pm 1.34\%$ for $n = 20$, $12.16\% \pm 2.20\%$ for $n = 40$, and $18.96\% \pm 2.04\%$ for $n = 80$). This demonstrates that we are in an interesting regime of DNN performance, at least for these tiny data-sets.

# I   Quadratic fully connected network

One of the simplest settings where GPs are expected to strongly under-perform finite DNNs is the case of quadratic fully connected DNNs [28]. Here we consider some positive teacher $g$ and a student DNN $f$ of the form

$$g(\mathbf{x}) = (\mathbf{w}_* \cdot \mathbf{x})^2 - \sigma_w^2 \|\mathbf{x}\|^2 \qquad f(\mathbf{x}) = \sum_{m=1}^{M} (\mathbf{w}_m \cdot \mathbf{x})^2 - \sigma_w^2 \|\mathbf{x}\|^2 \qquad \text{(I.1)}$$

where $\mathbf{w}_*, \mathbf{x} \in \mathbb{R}^d$. The $\|\mathbf{x}\|^2$ shift is not part of the original model but has only a superficial shift effect useful for book-keeping later on.

At large $M$ and for $w_{m,i}$ drawn from $\mathcal{N}(0, \sigma_w^2/M)$, the student generates a GP prior. It is shown below that the GP kernel is simply $K(\mathbf{x}, \mathbf{x}') = \frac{2\sigma_w^4}{M}(\mathbf{x} \cdot \mathbf{x}')^2$. As such it is proportional to the kernel of the above DNN with an additional linear read-out layer. The above model can be written as $\sum_{ij} x_i[P_{ij} - \sigma_w^2 \delta_{ij}]x_j$ where $P_{ij}$ is a positive semi-definite matrix. The eigenvalues of the matrix appearing within the brackets are therefore larger than $-\sigma_w^2$ whereas no similar restriction occurs for DNNs with a linear read-out layer. This extra restriction is completely missed by the GP approximation and, as discussed in Ref. [28], leads to strong performance improvements compared to what one expects from the GP or equivalently the DNN with the linear readout layer. Here we demonstrate that our self-consistent approach at the saddle-point level captures this effect.

We consider training this DNN on $n$ train points $\{\mathbf{x}_\mu\}_{\mu=1}^n$ using noisy GD training with weight decay $\gamma = M\sigma^2/\sigma_w^2$. We wish to solve for the predictions of this model with our shifted target approach. To this end, we first derive the cumulants associated with the effective Bayesian prior $(P_0(\vec{f}))$ here. Equivalently stated, obtain the cumulants of the equilibrium distribution of $\vec{f}$ following training with no data, only a weight decay term. This latter distribution is given by

$$P_0\left(\vec{f}\right) = \int d\mathbf{w} \, e^{-\frac{M}{2\sigma_w^2}\sum_{m=1}^M \|\mathbf{w}_m\|^2} \prod_{\mu=1}^{n+1} \delta\left(f_\mu - \sum_{m=1}^M (\mathbf{w}_m \cdot \mathbf{x}_\mu)^2 + \sigma_w^2 \|\mathbf{x}_\mu\|^2\right) \qquad \text{(I.2)}$$

To obtain the cumulants, we calculate the cumulant generating function of this distribution given by

$$\mathcal{C}(t_1, ..., t_{n+1}) \qquad \text{(I.3)}$$
$$= \log\left(\int \prod_{m,i=1}^{M,d} \frac{dw_{m,i}}{\sqrt{2\pi M^{-1}\sigma_w^2}} e^{-\sum_{m,i=1,1}^{M,d} M\frac{w_{m,i}^2}{2\sigma_w^2} + \sum_{\mu=1}^{n+1} it_\mu\left[\sum_{m,i=1,1}^{M,d}(\mathbf{w}_m \cdot \mathbf{x}_\mu)^2 - \sigma_w^2 \|\mathbf{x}_\mu\|^2\right]}\right)$$
$$= M\log\left(\int \prod_{i=1}^d \frac{dw_i}{\sqrt{2\pi M^{-1}\sigma_w^2}} e^{-M\frac{\|\mathbf{w}\|^2}{2\sigma_w^2} + \sum_{\mu=1}^{n+1} it_\mu\left[(\mathbf{w} \cdot \mathbf{x}_\mu)^2\right]}\right) - \sum_{\mu=1}^{n+1} it_\mu \sigma_w^2 \|\mathbf{x}_\mu\|^2$$
$$= M\log\left(\int \prod_{i=1}^d \frac{dw_i}{\sqrt{2\pi M^{-1}\sigma_w^2}} e^{-\frac{\mathbf{w}^\mathsf{T}\left[I - 2M^{-1}\sigma_w^2 \sum_\mu it_\mu \mathbf{x}_\mu \mathbf{x}_\mu^\mathsf{T}\right]\mathbf{w}}{2M^{-1}\sigma_w^2}}\right) - \sum_{\mu=1}^{n+1} it_\mu \sigma_w^2 \|\mathbf{x}_\mu\|^2$$
$$= -\frac{M}{2}\log\left(\det\left[I - 2M^{-1}\sigma_w^2 \sum_\mu it_\mu \mathbf{x}_\mu \mathbf{x}_\mu^\mathsf{T}\right]\right) - \sum_{\mu=1}^{n+1} it_\mu \sigma_w^2 \|\mathbf{x}_\mu\|^2$$
$$= -\frac{M}{2}\text{Tr}\left(\log\left[I - 2M^{-1}\sigma_w^2 \sum_\mu it_\mu \mathbf{x}_\mu \mathbf{x}_\mu^\mathsf{T}\right]\right) - \sum_{\mu=1}^{n+1} it_\mu \sigma_w^2 \|\mathbf{x}_\mu\|^2$$

Taylor expanding this last expression is straightforward. For instance up to third order is gives

$$\mathcal{C}(t_1, ..., t_{n+1}) = \frac{M}{2}\sum_{\mu_1,\mu_2}(2M^{-1}\sigma_w^2)^2 \frac{it_{\mu_1}it_{\mu_2}}{2}(\mathbf{x}_{\mu_1} \cdot \mathbf{x}_{\mu_2})(\mathbf{x}_{\mu_2} \cdot \mathbf{x}_{\mu_1}) \qquad \text{(I.4)}$$
$$+ \frac{M}{2}\sum_{\mu_1,\mu_2,\mu_3}(2M^{-1}\sigma_w^2)^3 \frac{it_{\mu_1}it_{\mu_2}it_{\mu_3}}{3}(\mathbf{x}_{\mu_1} \cdot \mathbf{x}_{\mu_2})(\mathbf{x}_{\mu_2} \cdot \mathbf{x}_{\mu_3})(\mathbf{x}_{\mu_3} \cdot \mathbf{x}_{\mu_1}) + ...$$

from which the cumulants can be directly inferred, in particular the associated GP kernel given by

$$K\left(\mathbf{x}_\mu, \mathbf{x}_\nu\right) = 2M^{-1}\sigma_w^4 \left(\mathbf{x}_\mu \cdot \mathbf{x}_\nu\right)^2 \qquad \text{(I.5)}$$

Following this, the target shift equation, at the saddle-point level, becomes

$$\Delta g_\nu = \partial_{t_\nu}\left(\mathcal{C}(t_1..t_n, t_{n+1}=0) - \sum_{\mu_1,\mu_2}\frac{K(\mathbf{x}_{\mu_1},\mathbf{x}_{\mu_2})}{2!}it_{\mu_1}it_{\mu_2}\right)\Big|_{t_1..t_n=\frac{\delta g_1}{\sigma^2}..\frac{\delta g_n}{\sigma^2}} \tag{I.6}$$

$$= -\sum_\mu K(\mathbf{x}_\nu,\mathbf{x}_\mu)\frac{\hat{\delta}g_\mu}{\sigma^2} + \sigma_w^2\,\mathrm{Tr}\left(\mathbf{x}_\nu\mathbf{x}_\nu^\mathsf{T}\left[I - 2M^{-1}\sigma_w^2\sum_\mu\frac{\hat{\delta}g_\mu}{\sigma^2}\mathbf{x}_\mu\mathbf{x}_\mu\right]^{-1}\right) - \sigma_w^2\|\mathbf{x}_\nu\|^2$$

$$= -\sum_{\mu=1}^n K(\mathbf{x}_\nu,\mathbf{x}_\mu)\frac{\hat{\delta}g_\mu}{\sigma^2} + \sigma_w^2\mathbf{x}_\nu^\mathsf{T}\left[I - 2M^{-1}\sigma_w^2\sum_{\mu=1}^n\frac{\hat{\delta}g_\mu}{\sigma^2}\mathbf{x}_\mu\mathbf{x}_\mu^\mathsf{T}\right]^{-1}\mathbf{x}_\nu - \sigma_w^2\|\mathbf{x}_\nu\|^2$$

$$\hat{\delta}g_\nu = (g_\nu - \Delta g_\nu) - \sum_{\mu,\mu'=1}^n K(\mathbf{x}_\nu,\mathbf{x}_\mu)\tilde{K}_{\mu,\mu'}^{-1}(g_{\mu'} - \Delta g_{\mu'})$$

The above non-linear equation for the quantities $\hat{\delta}g_1,\ldots,\hat{\delta}g_n$ could be solved numerically, with the most numerically demanding part being the inverse of $\tilde{K}_{\mu,\nu} = K(\mathbf{x}_\mu,\mathbf{x}_\nu) + \sigma^2\delta_{\mu,\nu}$ on the training set.

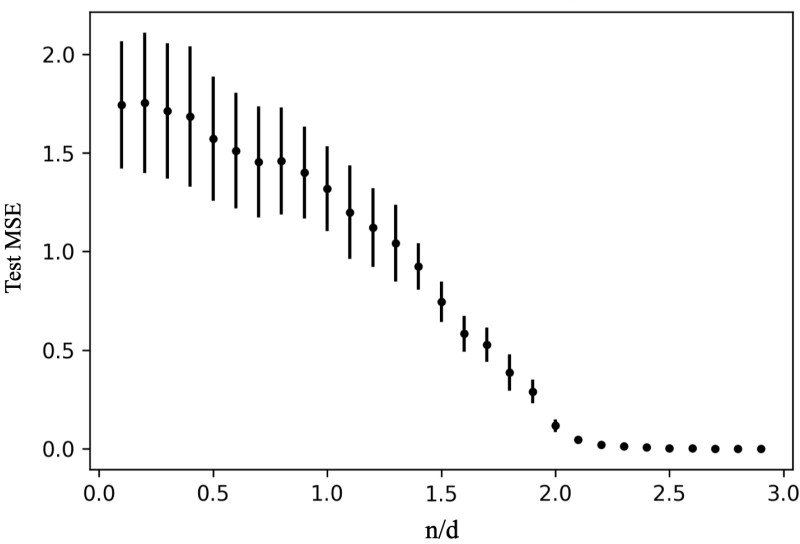

Test MSE as a function dataset size (n) for d=20

Figure 5: Test MSE as a function of $n/d$ for the phase retrieval model as predicted by our self-consistent equation at the saddle-point level (without any EK-type approximation). Train and test data are drawn uniformly from the $d = 20$ hypersphere with radius 1. The graph shows the median test MSE of 60 different data sets. Our approach captures the desired $n = 2d$ threshold value [28] whereas lazy-learning/GP will predict a cross over at $n = O(d^2)$.

Figure 5 shows the numerical results for the test MSE as obtained by solving the above equations for $\hat{\delta}g$ on the training set, taking $\nu = *$ in these equation together with the self-consistent $\hat{\delta}g_\mu$ to find the mean-predictor, and taking the average MSE of the latter over the test set. Both test and train data sets were random points sampled uniformly from a $d$ dimensional hypersphere of radius one. The test dataset contained 100 points and the figure shows the test MSE as a function of $n/d$ where $d = 20$, $\sigma_w^2 = 1$, $\sigma^2 = 2.76\cdot 10^{-6}$, $M = 4d$, and $w_i^*$ drawn from $\mathcal{N}(0,1)$. The non-linear equations were solved using the Newton-Krylov algorithm together with gradual annealing from $\sigma^2 = 1$ down to the above values. The figure shows the median over 60 data sets. Remarkably, our self-consistent approach yields the expected threshold values of $n/d = 2$ [28] separating good and poor performance. Discerning whether this is a threshold or a smooth cross-over in the large $d$ limit is left for future work.

Turning to analytics, one can again employ the EK approximation as done for the CNN. However taking $\sigma^2$ to zero invalidates the EK approximation and requires a more advance treatment as in Ref.

[11]. We thus leave an EK type analysis of the self-consistent equation at $\sigma^2 = 0$ for future work and instead focus on the simpler case of finite $\sigma^2$ where analytical predictions can again be derived in similar fashion to our treatment of the CNN.

To simplify things further, we also commit to the distribution $[\mathbf{x}_\mu]_i \sim \mathcal{N}(0, 1/d)$. In this setting $K(\mathbf{x}, \mathbf{x}')$ has two distinct eigenvalues w.r.t. to this measure, the larger one ($\lambda_0 = 2M^{-1}\sigma_w^4 \left(\frac{2}{d^2} + \frac{1}{d}\right)$) associated with $f(\mathbf{x}) = \|\mathbf{x}\|^2$ and a smaller one ($\lambda_2 = 2M^{-1}\sigma_w^4 \frac{2}{d^2}$) associated with $x_i x_j$ (with $i \neq j$) and $\sum_i a_i x_i^2$ (with $\sum_{i=1}^d a_i = 0$) eigenfunctions.

Next we argue that provided the discrepancy is of the following form

$$\hat{\delta} g_\mu = \alpha g(\mathbf{x}_\mu) + \beta \sigma_w^2 \|\mathbf{x}\|_\mu^2 \tag{I.7}$$

then within the EK limit the target shift is also of the form of the r.h.s. with $\alpha_\Delta$ and $\beta_\Delta$ and the target shift equations reduce to two coupled non-linear equations for $\alpha$ and $\beta$. Following the EK approximation, we replace all $\sum_\mu$ in the target shift equation with $n \int d\mu_x$ and obtain

$$\Delta g(\mathbf{x}) = -\frac{n}{\sigma^2} \int d\mu_{x'} K(\mathbf{x}, \mathbf{x}') \hat{\delta} g(\mathbf{x}') + \sigma_w^2 \mathbf{x}^\mathsf{T} \left[ I - 2M^{-1}\sigma_w^2 \frac{n}{\sigma^2} \int d\mu_{x'} \hat{\delta} g(\mathbf{x}') \mathbf{x}' \mathbf{x}'^\mathsf{T} \right]^{-1} \mathbf{x} - \sigma_w^2 \|\mathbf{x}\|^2 \tag{I.8}$$

Next we note that the $i \neq j$ element of the matrix $\int d\mu_x g(\mathbf{x}) \mathbf{x} \mathbf{x}^\mathsf{T}$ is given by

$$\int d\mu_x g(\mathbf{x}) x_i x_j = \int d\mu_x \left( (\mathbf{w}_* \cdot \mathbf{x})^2 - \sigma_w^2 \|\mathbf{x}\|^2 \right) x_i x_j = 2d^{-2} w_i^* w_j^* \tag{I.9}$$

whereas for $i = j$ we obtain

$$\int d\mu_x g(\mathbf{x}) x_i x_i = 3d^{-2}((w_i^*)^2 - \sigma_w^2) + \sum_{j \neq i} d^{-2}((w_j^*)^2 - \sigma_w^2) \tag{I.10}$$

$$= 3d^{-2}((w_i^*)^2 - \sigma_w^2) + \sum_j d^{-2}((w_j^*)^2 - \sigma_w^2) - d^{-2}((w_i^*)^2 - \sigma_w^2)$$

$$= 2d^{-2}((w_i^*)^2 - \sigma_w^2)$$

taking this together with the simpler term ($\int d\mu(\mathbf{x}) \frac{\beta}{\alpha} \sum_i x_i x_i \mathbf{x} \mathbf{x}^\mathsf{T} = \frac{\beta(d+2)}{\alpha d^2} I$)

$$\Delta g(y) = -\frac{n}{\sigma^2} \int d\mu_{x'} K(\mathbf{x}, \mathbf{x}') \left[ \alpha g(\mathbf{x}') + \beta \sigma_w^2 \|\mathbf{x}'\|^2 \right] + \dots$$

$$\sigma_w^2 \mathbf{x}^\mathsf{T} \left[ I - 2\lambda_2 \frac{n}{\sigma_w^2 \sigma^2} \sigma_w^2 d^{-2} \left[ \mathbf{w}_* \mathbf{w}_*^\mathsf{T} - \sigma_w^2 \left( 1 - \frac{\beta(d+2)}{2\alpha} \right) I \right] \right]^{-1} \mathbf{x} - \sigma_w^2 \|\mathbf{x}\|^2 \tag{I.11}$$

Consider the matrix $(\mathbf{w}_* \mathbf{w}_*^\mathsf{T} + bI)$, appearing in the above denominator with $b = \left( \frac{\beta \sigma_w^2 (d+2)}{2\alpha} - \sigma_w^2 \right)$, and note that

$$\mathbf{x}^\mathsf{T} \cdot (\mathbf{w}_* \mathbf{w}_*^\mathsf{T} + bI)^n \mathbf{x} = (\mathbf{w}_* \cdot \mathbf{x})^2 \frac{(\|\mathbf{w}_*\|^2 + b)^n - b^n}{\|\mathbf{w}_*\|^2} + b^n \|\mathbf{x}\|^2 \tag{I.12}$$

Plugging this equation into a Taylor expansion of the denominator of Eq. I.11 one finds that all the resulting terms are of the desired form of a linear superposition of $g(\mathbf{x})$ and $\|\mathbf{x}\|^2$. Considering the first term on the r.h.s. of Eq. I.11, $\beta \|x\|^2$ is already an eigenfunction of the kernel whereas $g(\mathbf{x})$ can be re-written as

$$g(\mathbf{x}) \equiv \sum_{i \neq j} w_i^* w_j^* x_i x_j + \sum_i (w_i^*)^2 x_i^2 - \sigma_w^2 \|\mathbf{x}\|^2 \tag{I.13}$$

$$= \sum_{i \neq j} w_i^* w_j^* x_i x_j + \sum_i \left( (w_i^*)^2 - \frac{\|\mathbf{w}_*\|^2}{d} \right) x_i^2 + \left( \frac{\|\mathbf{w}_*\|^2}{d} - \sigma_w^2 \right) \|\mathbf{x}\|^2$$

so that the first two terms on the r.h.s. are $\lambda_2$ eigenfunctions and the last one is a $\lambda_0$ eigenfunctions. Summing these different contributions along with the aforementioned Taylor expansion, one finds that $\Delta g(\mathbf{x})$ is indeed a linear superposition of $g(\mathbf{x})$ and $\|\mathbf{x}\|^2$.

Next we wish to write down the saddle-point equations for $\alpha$ and $\beta$. For simplicity we focus on the case where $g(\mathbf{x})$ is chosen orthogonal to $\|\mathbf{x}\|^2$ under $d\mu(\mathbf{x})$, namely $\|\mathbf{w}_*\|^2 = d\sigma_w^2$. Under this choice the self-consistent equations become

$$\alpha = \frac{\frac{\sigma^2}{n}}{\lambda_2 + \frac{\sigma^2}{n}} \left[ 1 - \frac{\alpha c \sigma_w^2}{1 - \alpha b c} \left( \frac{1}{1 - \frac{\alpha \|\mathbf{w}_*\|^2 c}{1 - \alpha b c}} \right) + \frac{n}{\sigma^2} \alpha \lambda_2 \right] \tag{I.14}$$

$$\beta = -\frac{\frac{\sigma^2}{n}}{\lambda_0 + \frac{\sigma^2}{n}} \left[ \frac{\alpha c}{1 - \alpha b c} \left( b + \sigma_w^2 \frac{1}{1 - \frac{\alpha \|\mathbf{w}_*\|^2 c}{1 - \alpha b c}} \right) - \frac{n}{\sigma^2} \beta \lambda_0 \right]$$

where the constant $b$ was defined above and $c = \frac{n\lambda_2}{\sigma_w^2 \sigma^2}$.

Next we perform several straightforward algebraic manipulations with the aim of extracting their asymptotic behavior at large $n$. Noting that $c\sigma_w^2 = \frac{n}{\sigma^2}\lambda_2$, $c\|\mathbf{w}_*\|^2 = 2d\frac{n}{\sigma^2}\lambda_2$, and $\alpha c b = -\frac{n}{\sigma^2}(\alpha\lambda_2 - 2\beta\lambda_0)$ we have

$$\alpha = \frac{1}{\lambda_2 + \frac{\sigma^2}{n}} \left[ \frac{\sigma^2}{n} - \frac{\alpha\lambda_2}{1 + \frac{n}{\sigma^2}(\alpha\lambda_2 - 2\beta\lambda_0)} \left( \frac{1}{1 - \frac{\alpha d \frac{n}{\sigma^2}\lambda_2}{1 + \frac{n}{\sigma^2}(\alpha\lambda_2 - 2\beta\lambda_0)}} \right) + \alpha\lambda_2 \right] \tag{I.15}$$

$$\beta = \frac{-1}{\lambda_0 + \frac{\sigma^2}{n}} \left[ \frac{2\alpha\frac{\lambda_0}{d+2}}{1 + \frac{n}{\sigma^2}(\alpha\lambda_2 - 2\beta\lambda_0)} \left( \frac{\beta(d+2)}{2\alpha} - 1 + \frac{1}{1 - \frac{\alpha d \frac{n}{\sigma^2}\lambda_2}{1 + \frac{n}{\sigma^2}(\alpha\lambda_2 - 2\beta\lambda_0)}} \right) - \beta\lambda_0 \right]$$

Further simplifications yield

$$\alpha = \frac{1}{\lambda_2 + \frac{\sigma^2}{n}} \left[ \frac{\sigma^2}{n} - \frac{\alpha\lambda_2}{1 + \frac{n}{\sigma^2}(\alpha\lambda_2 - 2\beta\lambda_0 - \alpha d\lambda_2)} + \alpha\lambda_2 \right] \tag{I.16}$$

$$\beta = \frac{-1}{\lambda_0 + \frac{\sigma^2}{n}} \left[ \frac{2\alpha\frac{\lambda_0}{d+2}}{1 + \frac{n}{\sigma^2}(\alpha\lambda_2 - 2\beta\lambda_0)} \left( \frac{\beta(d+2)}{2\alpha} + \frac{\alpha\frac{n}{\sigma^2}d\lambda_2}{1 + \frac{n}{\sigma^2}(\alpha\lambda_2 - 2\beta\lambda_0 - \alpha d\lambda_2)} \right) - \beta\lambda_0 \right]$$

noting that $d\lambda_2 = 2(\lambda_0 - \lambda_2)$ we find

$$\alpha = \frac{1}{\lambda_2 + \frac{\sigma^2}{n}} \left[ \frac{\sigma^2}{n} - \frac{\alpha\lambda_2}{1 - \frac{n}{\sigma^2}(\alpha\lambda_2 + 2(\beta + \alpha)\lambda_0)} + \alpha\lambda_2 \right] \tag{I.17}$$

$$\beta = \frac{-1}{\lambda_0 + \frac{\sigma^2}{n}} \left[ \frac{2\alpha\frac{\lambda_0}{d+2}}{1 + \frac{n}{\sigma^2}(\alpha\lambda_2 - 2\beta\lambda_0)} \left( \frac{\beta(d+2)}{2\alpha} + \frac{\alpha\frac{n}{\sigma^2}d\lambda_2}{1 - \frac{n}{\sigma^2}(\alpha\lambda_2 + 2(\beta + \alpha)\lambda_0)} \right) - \beta\lambda_0 \right]$$

The first equation above is linear in $\beta$ and yields in the large $d$ limit

$$\beta = -\alpha - \frac{\alpha}{d(1 - \alpha)} + \frac{\sigma^2}{2\lambda_0 n} \tag{I.18}$$

It can also be used to show that

$$\frac{\alpha\lambda_2}{1 - \frac{n}{\sigma^2}(\alpha\lambda_2 + 2(\beta + \alpha)\lambda_0)} = (1 - \alpha)\frac{\sigma^2}{n} \tag{I.19}$$

which when placed in the second equation yields

$$\beta = \frac{\lambda_0}{\lambda_0 + \frac{\sigma^2}{n}} \cdot \frac{\frac{n}{\sigma^2}\beta(\alpha\lambda_2 - 2\beta\lambda_0) + 2(1 - \alpha)\alpha}{1 + \frac{n}{\sigma^2}(\alpha\lambda_2 - 2\beta\lambda_0)} \tag{I.20}$$

At large $n$, we expect $\alpha$ and $\beta$ to go to zero. Accordingly to find the asymptotic decay to zero, one can approximate $\alpha(1 - \alpha) \approx \alpha$, and similarly $\alpha/(1 - \alpha) \approx \alpha$. This along with the large $d$ limit simplifies the equations to a quadratic equation in $\beta$

$$\beta^2 \frac{2\lambda_0 n}{\sigma^2} \left( 1 - \frac{\lambda_0}{\lambda_0 + \sigma^2/n} \right) + \beta \left( -1 - 2\frac{\lambda_0}{\lambda_0 + \sigma^2/n} \right) + \frac{\lambda_0}{\lambda_0 + \sigma^2/n} \frac{\sigma^2}{\lambda_0 n} \tag{I.21}$$

which for $\sigma^2/n \ll \lambda_0$ simplifies further into

$$2\beta^2 - 3\beta + \frac{\sigma^2}{2\lambda_0 n} = 0 \tag{I.22}$$

yielding

$$\beta = \frac{4}{18} \frac{\sigma^2}{\lambda_0 n} \tag{I.23}$$

$$\alpha = \frac{5}{18} \frac{\sigma^2}{\lambda_0 n} \tag{I.24}$$

We thus find that both $\alpha$ and $\beta$ are of the order of $\frac{\sigma^2/n}{2\lambda_0} = n^{-1} \frac{Md\sigma^2}{4\sigma_w^4}$. Hence $n$ scaling as $Md$ ensures good performance. This could have been anticipated as for small yet finite $\sigma^2$ each $n$ can be seen as a soft constrained on the parameters of the DNN and since the DNN contains $Md$ parameters $n = O(Md)$ should provide enough data to fix the student's parameters close to the teacher's.