# OpenReview forum: "A self consistent theory of Gaussian Processes captures feature learning effects in finite CNNs"
_NeurIPS.cc/2021/Conference — NeurIPS 2021 Poster_

### Official Review · Reviewer_9Tuv · 2021-07-11

**Rating:** 7
**Confidence:** 2

**Summary:**

This paper introduces a new perspective on feature learning in wide neural networks by demonstrating that DNNs trained with noisy gradients and large training data converge to solutions with mean predictions dictated by GP regression with shifted targets when using MSE loss. The authors then consider two toy CNN models, one linear and one non-linear, where they are able to make more analytic progress with their theory, and demonstrate the benefits of of feature learning of wide enough but finite NNs compared to their corresponding infinite-width GPs in terms of sample complextiy on student-teacher tasks. Finally, they also show a phase transition (marked by a critical value of width/channels) in the distribution of eigenvalues empirical covariance of weights in the first layer of a CNN instantiation, between a feature learning regime (where there is an outlier eigenvalue that is associated to the teacher's features) and a non-feature learning regime. Experimental evidence is provided to support these claims.

**Limitations And Societal Impact:**

- I'm a bit unsure what the setup is in Figure 1. E.g. what is C*, what is the input dataset? Also isn't it concerning that as you increase the number of channels \alpha gets larger (so the discrepancy with the ground truth increases)?
- This isn't really a limitation, but it's quite hard for someone (like myself) who is not well versed in the techniques used in statistical physics to understand all the details of the author's arguments.

**Main Review:**

*originality* The view of trained NNs as GP regression with modified targets is novel as far as I can tell, as are the insights developed on toy CNN models. The paper is clear in how it uses and extends previous work in my opinion.

*quality* I enjoyed reading this paper, and believe it is of high quality. The approach is non-rigorous but I believe technically sound, and the insights that the authors obtain (e.g. the main equations 8 and 16) are interesting. The choice of toy models is also made well, in that they are simple enough to obtain analytic results but also complicated enough to lead to profound results of benefits of feature learning (in terms of sample complexity).

*clarity* the paper is well written and organised, particularly with regards to relation to previous work.

*significance* The paper examines a key question which is of feature learning in wide NNs, which will be of interest to the general NeurIPS community. The new results about trained NNs being GP regression with shifted targets is certainly surprising to me, and I can see these techniques being built upon in future work.

Minor points:
- The comment on line 20 about 'DNNs map to Bayesian inference on GPs governed by the NTK' is incorrect. https://arxiv.org/abs/2007.05864 shows you need to add in extra randomness to the initialised NN to obtain a Bayesian interpretation to trained NNs in the NTK regime.
- where -> were in line 332
- There is another concurrent paper to [41] in line 77, https://arxiv.org/abs/2106.06615, which provides similar results.

**Time Spent Reviewing:**

3

---

> ### Author Response · Authors · 2021-08-10
> **Response to Reviewer 9Tuv**
>
> 'DNNs map to Bayesian inference on GPs governed by the NTK' (in relation to He et al. 2020) -
> We thank the reviewer for pointing out this work. We have added the paper by He et al. 2020 in line 20, in reference to the NTK thereby hopefully helping the keen reader to clarify this subtlety. We also mentioned the mapping to GPs may require ensembling. We note that since our training protocol involves injected white noise at each time step of training, we focus on the correspondence described in Ref. [30], which relates DNNs to the NNGP kernel, not the NTK. It would nonetheless be interesting to extend our approach to capture finite width/channel correction to this Bayesian NTK setting.
>
> "Add Ref. Noci et al. 2021" -
> The reference was added to the main text.
>
> Setup in Figure 1 -
> $C^*$ is the number of channels in the teacher network as indicated in Eq. 12, which we took to be $C^*=1$ in the results reported. The inputs $\mathbf{x}$ are sampled from a standard Gaussian, as indicated in lines 183-184. The fact that as we increase the number of channels  $\alpha$ gets larger is not concerning, it is precisely the motivation for this toy example: since the targets are generated by the teacher which has only $C^*=1$ channels, it is actually optimal for the student to have a matching number of channels $C=1$ and as $C$ increases we approach the GP limit which is highly sub-optimal in this setting, as discussed in the paragraph beginning in line 185.
>
> Statistical physics jargon/techniques -
> We have tried to keep jargon and technical details in the main text at a minimal level. Following your comment, we will try to identify more points in the text where our phrasing could be modified so as to be accessible to a wider audience.

---

> > ### Comment · Reviewer_9Tuv · 2021-08-23
> > **Thank you**
> >
> > Thanks for comments and the clarification regarding Figure 1, I am maintaining both my rating and confidence score.

---

### Official Review · Reviewer_ZQro · 2021-07-16

**Rating:** 7
**Confidence:** 3

**Summary:**

The authors develop a self-consistent Gaussian Processes (GPs) theory to account for finite size effects in Deep neural networks (DNNs). The equivalence between GPs and DNNs is a well known result that holds in the so called ‘lazy learning’ regime in the infinite width/channel limit, where the DNNs’ parameters don’t change during training. In practical contexts, however, DNNs are finitely sized and operate in a different regime, called ‘feature learning’, where parameters change substantially during training. This paper provides a theoretical framework to account for finite size effects, and so it allows to investigate the feature learning regime.

To accomplish this, the authors consider the partition function $Z$ associated with the DNN posterior distribution $P(f)$, in the case of training via noisy (full batch) gradient flow with MSE loss. $Z$ is given by an integral over the prior distribution $P_0(f)$, which can be expressed in terms of cumulants. Following their notation, $C$ is the hyper-parameter controlling the over-parametrization. In the infinite-$C$ limit, only the first two cumulants survive, and the standard GP-DNN equivalence is recovered, with the GP kernel given by the second cumulant.
For $C$ finite, all cumulants contribute, and the authors have to rely on a saddle point approximation to make progress. By doing so, they arrive at a self-consistent set of equations for the target shift that describes the predictive mean of the DNN. In fact, the predictive mean is given by the same GP-like expression found for the infinite-$C$ limit, though with shifted targets. Next, the authors consider a second order expansion of the action so as to evaluate the posterior covariance.

In order to test their results, the authors consider two examples of DNN architectures that over-perform their associated GPs. Of the two, they mostly focus on the case of a linear two-layer CNN, in a teacher-student set up.  For this simple architecture, the cumulants can be worked out analytically. However, in order to deal with the series appearing in the expression of $P_0(f)$, the authors consider the limit of infinite number of training data, also known as Equivalent Kernel limit. This leads to an analytical expression for $\alpha$, the proportionality factor between the discrepancy in prediction and the target value. Numerical verifications confirm their theoretical predictions (showing mismatches decreasing with the number of training samples).
By modelling the empirical weight covariance matrix as a Wishart matrix with a rank-one perturbation, the authors are able to detect a feature learning transition for this model. This is marked by the appearance of outliers in the associated spiked Marchenko-Pastur eigenvalue spectrum.

Finally, the authors consider the case of a two-layer FCN with average pooling and quadratic activations, with a rank-1 teacher. They provide the expression of the cumulant generating function, as well as the equation for the targets shift, and leave the analysis to Appendix I.

**Limitations And Societal Impact:**

The authors do not put great effort in explaining the limitations of their analysis. In particular, the obstacles for their theory to be applied to different DNN architectures should be discussed with more details.

**Main Review:**

The overall quality of the paper is good: I found it technically solid, clearly written, original (at least to my knowledge), and all claims are well supported. The organisation of the paper is fine but could be improved (see my comments below).

I have the following comments for the authors:

1. Other works from the literature (see e.g. https://arxiv.org/abs/1906.08034) distinguish lazy and feature learning regimes in the infinite-$C$ limit, according to the scaling of the network's parameters with $C$. In this paper, feature learning appears to be solely a finite size effect (the parameters’ scaling with $C$ gives the lazy learning regime in the infinite-$C$ limit). I wonder if the self consistent theory developed here would work for different scalings of the parameters with $C$, and in which case, as I presume the cumulants too will scale differently with $C$. I would appreciate if the authors could comment on this point.

2. The authors use a saddle point (SP) method in a somewhat non standard way, as explained in the first paragraph of Appendix A.4. I think the paper would gain in clarity with a sentence in the main text that spells out this point. I would also suggest to state more clearly that the approximation assumes $n$ large, with $n$ the training set size, in the paragraph where the SP method is introduced. A further improvement would be to comment on the error induced by this approximation when $n$ is finite, as e.g. to explain the mismatches observed in figure 1 (increasing as $n$ decreases).

3. With the SP method, the authors are able to find the expression for the posterior covariance matrix by expanding the action to quadratic orders. This coincides with the posterior covariance of a GP with a kernel given by $K + \Delta K$ (with $\Delta K$ properly defined in eq. (9)). As far as I understand, the term $\Delta K$ makes this kernel different from the one appearing in the predictive mean with shifted targets. If this is the case, the predictions for finitely sized DNNs cannot be really described as equivalent to a GP regression with shifted targets (while the equivalence in the infinite-$C$ limit is exact). If so, this should be stated clearly in the paper, as to avoid misinterpretations.

4. Although section 4.4 provides a valid and interesting investigation of the feature learning transition in the CNN model, it is not completely clear to me how the proposed analysis fits with the rest of the paper, i.e. if/how the self-consistent shifted target approach is relevant for the task. I would recommend the authors to make this clearer in the main text; as of now, it appears to be more of a stand alone section (also, it is not immediately obvious why the feature-to-lazy learning transition observed in this specific model is of interest, nor is it clear if the same method could be applied to different architectures).

5. The discussion of the FCN architecture presented in the main text is limited to the equation for the target shift, while the rest of it is left to Appendix I. Even there, there is no figure showing a comparison between theoretical predictions and empirical results. I would have expected to see a plot of this kind in the main text (i.e. one similar to fig. 1), as to support the general applicability of the proposed method. Basically, the content of the FCN discussion doesn't seem to be enough for a separate section in the main text.

**Time Spent Reviewing:**

16

---

> ### Author Response · Authors · 2021-08-10
> **Response to Reviewer ZQro**
>
> Self-consistent theory for different scaling of the parameters with $C$, specifically the Mean Field regime -
> We thank the reviewer for this interesting comment. It is indeed true that in our setting feature learning is a finite size effect. If one attempts to apply the so-called Mean-Field (MF) scaling (where the network last layer weights scale as $1/C$ rather than $1/\sqrt{C}$ as in our paper), then we can say the following:
> 1. By inspecting Eq. B.1 it is clear that for the MF scaling the $r$'th cumulant would scale as $\kappa_r = O(1/C^{r-1})$ rather than $\kappa_r = O(1/C^{r/2 - 1})$ as in our paper. More generally, for the last layer weights scaling as $1/C^\beta$ the cumulants would scale as $\kappa_r = O(1/C^{r/\beta-1})$.
> 2. However, we should point out a subtle yet important issue: the NTK scaling vs. mean-field scaling are a choice about initialization. In contrast, in our correspondence between stochastic processes (which tend to GPs as $C \to \infty$) and DNNs trained with noisy gradients and weight decay, this scaling arises as a choice of the weight decay parameter $\gamma$ appearing in Eq. 1 (see Ref. [30] for more details). This is seen from inspecting the relation between the variance of the weights and the gradient noise $\sigma$ and weight decay $\gamma$, e.g. for the last layer weights: $\sigma_a^2 = 2\sigma^2 / \gamma_a$. So the MF scaling implies that $\gamma_a = O(C^2)$. This very strong weight decay is expected to give rise to a network with essentially zero output throughout training, and in particular, does not seem to support any kind of feature learning.
> 3. This pathological behavior in the MF scaling can also be seen by directly inspecting the action in Eq. A.14: assuming the discrepancies $it_\mu$ remains finite, all cumulants, including the kernel, vanish as $C \to \infty$ so we are left with a trivial prediction $\left\langle it_{\mu}\right\rangle = g_\mu / \sigma^2$ which, as anticipated in the previous bullet point, corresponds to a DNN whose output is identically zero.
>
> Clarify main text discussion of SP approximation, and mismatch in Fig. 1 -
> We have added a sentence in the paragraph where the SP method is introduced stating the non-standard use of the SP approximation and the fact that it relies on $n$ being large. In the original submission, we mentioned in the caption of Fig. 1 that our predictions become more accurate as $n$ grows, but we have also added an extra sentence in the text body emphasizing this point.
>
> Clarify $K$ vs. $K + \Delta K$ for the mean predictor -
> We thank the reviewer for pointing out this potential source of confusion. Indeed, the $K$'s appearing in the self-consistent equations for the mean predictor correspond to the original kernel, i.e. the second cumulant of the prior $P_0$ without any shifts. Thus, the predictive mean for finite DNNs can be described as equivalent to a GP regression with shifted targets. A sentence clarifying this point was added to the main text.
>
> Section 4.4: relation to the rest of the paper, generality -
> See our response to Reviewer P6G1.
>
> FCN quadratic architecture -
> We acknowledge that it would have been better to perform a comparison similar to that shown in Fig. 1 also for the quadratic FCN, which includes numerical results for actual NNs trained using our procedure. These numerical simulations take a long time to run and gather statistics due to the noisy training involved, thus we decided to focus only on the linear CNN model. However, on top of the equation for the target shift in the main text, in App. I we provide a numerical solution of the full self-consistent equations (without any EK type approximations) in Fig. 5, which agrees with previous results (Ref. [26]). Additionally, using the EK approximation, we provide the form of the self-consistent equations (Eq. I.13) and their asymptotic behavior at large $n$ (Eq. I.16), which give some analytical insight. We believe this section is of importance since it demonstrates that our self-consistent theory is not restricted to linear DNNs.
>
> Expand on the obstacles for the theory to be applied to different DNN settings -
> See detailed response to reviewer P6G1.

---

### Official Review · Reviewer_P6G1 · 2021-07-20

**Rating:** 6
**Confidence:** 4

**Summary:**

In this work, the authors solve for the infinite time mean predictor of a model with weight decay trained via noisy full batch gradient descent. The authors develop a general formalism appropriate for model weights initialized with mean zero and variance related to the weight decay and gradient noise parameters and push this formalism to explicit characterization of the model output for the case of one-hidden-layer linear teacher student CNNs trained with MSE loss and quadratic teacher student fully connected DNNs. In the former case, the authors observe a phase transition between the lazy and feature learning regimes as the model width decreases.

**Ethical Concerns:**

I do not see any ethical concerns.

**Limitations And Societal Impact:**

Some limitations, and encouragement for the authors to discuss the limitations, are discussed in the review above. The authors have adequately discussed societal impact.

**Main Review:**

The formalism and solvable model presented are interesting and have the potential to provide important insight into more realistic deep learning systems. At present, however, the manuscript falls short in a few key areas. I am open to increasing my score if these concerns are addressed (potentially using the additional page).

**Relevance to realistic DL models**

The theoretical and empirical setup presented here is interesting, but quite limited. Even if the formalism requires restrictive assumptions, the paper would be much stronger if the qualitative phenomena exhibited by the solvable models could be explored empirically in more realistic settings. For example, can one see evidence of the spiked-MP phase transition in the spectrum of the weights for realistic models trained with a teacher target? Does the same transition also empirically explain models trained with categorical labels? Do the solvable models make additional predictions that can be investigated empirically in a wider context?

**Relationship to existing literature**

Another area where the paper could be significantly strengthened is casting the present work in light of previous work. The authors have a fairly inclusive related work section, however there are two qualitative discussions which are important to include.
i) The discussion of how student weights learn teacher weights presented in Section 4.4 seems quite similar to the analysis in 1809.10374. Both discuss this process in terms of low rank perturbation of a noise matrix and the associated phase transition, following Benaych-Georges & Nadakuditi. It is true that the authors of 1809.10374 were not concerned with connecting this behavior to the lazy / feature transition, however more discussion of the connection to their analysis would be welcomed.
ii) Various mechanisms underlying the lazy/feature learning transition have been discussed in the literature. For example a transition as a function of model normalization was discussed in 1812.07956 and 1906.08034, a transition as a function of learning rate was discussed in 2003.02218, and a transition as a function of weight decay strength is implicit in 2006.08643. How does the transition here relate to those mechanisms? If different, are there measurements in realistic models we can perform to disentangle the key factors underlying the transition in relevant setups?

**Additional questions**

Fundamental limitations of formalism v convenient simplification -- It would be nice to emphasize a little more clearly which simplifications are essential for the analysis and which are used for simplicity of presentation. For example, the requirement that the weight initialization is related to the weight decay and gradient noise is essential, however the MSE loss is less essential. Is that correct? Is it correct even if one wants to understand the lazy/feature transition? Perhaps this could be emphasized more explicitly for more of the modeling assumptions?

Do these results hold for both C > n and C < n? If C <n, is the measure in 2 still unconstrained? If \sigma were 0, \tilde{K} would not be invertible for C<n so expressions such as 5 would be singular with non-zero sigma, is there any avatar of this discontinuity?

Do these results shed light on the \sigma-> 0, finite \gamma limit?

**Minor questions / comments**

- The authors provide a consistency condition for the saddle point approximation, but comment that one cannot reformulate the action with an explicit pre-factor. Can one introduce auxiliary fields (e.g. as in hep-th/0306133) to accomplish this?

- 2106.10165 -- which came out post submission -- might, none-the-less, be nice to reference in a final version, as they also attempt to use corrections to large width to understand the feature learning / lazy transition.

-----------------------------------

After the response from the authors and discussion with other reviewers. I have decided to revise my score up.

I appreciate the time the authors took in responding to my questions! I am only partially appeased by my main question -- what aspects of this solvable system are applicable (at least approximately) in realistic systems? The authors point to [27] ("Implicit Self-Regularization in Deep Neural Networks...") as a potential example of the spiked MP behavior they are analyzing in realistic setups. Though [27] does exhibit a variety of empirical distributions for spectra, it is not clear to me that they are realizing a version of the lazy feature transition discussed by the authors. That being said, I do think a model that can be analyzed through the non-perturbative lazy/feature transition is attractive. I also appreciated the additional discussion of Lampinen and Ganguli.


**Time Spent Reviewing:**

4.5

---

> ### Author Response · Authors · 2021-08-10
> **Response to Reviewer P6G1 - Part 1**
>
> Relevance to realistic DL models -
>
> As the reviewer is right to point out, toy models should represent some qualitative phenomena that are measurable in more advanced real-world settings. We believe our approach, and the toy models we solve using it, indeed exhibit phenomena that are more general.
>
> First, considering the spiked-MP phase, Ref. [27] ("Implicit Self-Regularization in Deep Neural Networks...") studies the same type of posterior weight matrices as we do, and reports various empirical classes (or phases) of these matrices. In particular, for some cases, such as LeNet5 and MiniAlexNet trained on CIFAR10, a spiked MP distribution is found (see Table 4. of that work). Notably, as these are trained on CIFAR10 and not a teacher DNN with a single channel, one expects several outliers and not just one. Indeed we view $\mathbf{w}_*$ of our toy model as corresponding to a Gabor filter in the more general context.
>
> Regarding categorical labels, we have formulated a set of self-consistent equations for a general smooth loss function, not necessarily the MSE loss (see App. C), and also a specific result for the cross-entropy loss (see Eq (C.5) in the appendix). However, drawing predictions from these is more challenging since, even without finite-width/channel effects, the theory is non-linear due to the cross-entropy loss. Hence our ability to deduce the effect of the target shift analytically is more challenging and we leave it for future work.
>
> Regarding other measurable effects - our saddle-point approach implies that the posterior distribution over the training-set is approximately a GP, however a non-centered one with a different kernel. This can be verified via normality tests in real-world DNNs trained according to our GD+noise protocol. This also raises the intriguing possibility of extracting these effective kernel matrices and target shifts numerically and applying theoretical tools and bounds from GPs to more complex DNNs.
>
> Last, we note that deep learning and machine learning are rich and versatile and real-world models probably show a variety of different empirical behaviors, each most likely requiring different analytical approaches and approximations. In accordance, our work does not pertain to encompass all phenomena but rather be a tool relevant to some portion of the models one encounters.
>
>
> Section 4.4 vs. Lampinen and Ganguli 2019 (Ref [20]) -
>
> The reviewer is correct in noticing the similarity to this paper, however, there are several important differences which we list here and have also added to the text:
> 1. Ref. [20] considers the statistics of the noisy teacher (i.e. the linear teacher network corrupted by additive iid noise), whereas we consider the statistics of the first layer of the trained student network.
> 2. Ref. [20] considers a fully connected architecture whereas we consider a CNN. Moreover, they consider the case where the input and output dimensions are both large with a fixed aspect ratio, while the hidden layer width is finite (thereby yielding a low-rank signal of the linear teacher that is corrupted by noise). The analog of this "aspect ratio" in our study is $S/C$, and, as mentioned above, refers to the student, not the teacher. The signal refers to the feature of the teacher network $\mathbf{w}^*$ that is only apparent in the spectrum for sufficiently small $C$.
> 3. The phase transition described in Ref. [20] occurs as the magnitude of the singular values of the teacher vary, thus it has no connection to the feature / lazy transition. In contrast, we focused on the transition that arises as one varies $C$. As mentioned in the text, this affects both the magnitude of the outlier and the shape of the bulk. We associate this transition to one separating a regime where the teacher leaves a clear signature in the spectrum of the student's weights, and one where it does not.
>
>
> Relation to other previous studies on phase transition as a function of model normalization, learning rate, weight decay strength -
>
> We thank the reviewer for pointing out these relevant works which we will include in our revised version. Regarding the first two, our training protocol involves equilibration and noisy training and hence initialization should not affect our results. The analog of initialization here, in its effect on the posterior, is weight decay which implicitly controls all our cumulants. Taking different scalings here can drastically enhance or reduce the non-linearities in the model and hence most likely affect feature learning as well (which comes hand in hand with non-linearities). For instance, taking a larger weight decay for the last layer can yield a Mean-Field (MF) scaling of the cumulants generally making them smaller (see a detailed answer on this matter to reviewer ZQro).
> In this regard, the scaling we chose is unique and is an essential assumption, much like in the Central Limit Theorem where one considers \textit{standardized} sums of random variables: $Y_N = \frac{1}{\sqrt{N}} \sum_{i=1}^N Z_i$ to get sensible results.
>
> Considering the interesting phenomena of learning rate induced transitions, here we find it harder to establish any concrete connection. One way to go will be to derive finite time discretization corrections to the ergodic distribution predicted by Langevin dynamics (see for instance https://arxiv.org/pdf/2012.04728.pdf). While interesting to explore, we do not see a direct connection between such a perturbation theory and perturbation theory for channel/width.

---

> ### Author Response · Authors · 2021-08-10
> **Response to Reviewer P6G1 - Part 2**
>
> limitations of formalism vs. convenient simplification for simplicity of presentation -
>
> In this paper, we set out to explore the accuracy of a certain saddle-point approximation, based on the intuition that since the non-quadratic terms in the action involve contributions of many data points-- a mean-field/Gaussian behavior can be expected at large $n$. As the reviewer is right to point out, the behavior is more subtle than just taking large $n$. For instance, a more careful heuristic argument, discussed in App. A, shows the importance of having non-negligible $\hat{\delta} g / \sigma^2$ and also the importance of having what we dubbed collective large $n$ contributions. There we also provide a rigorous necessary condition, albeit in the form of a rather opaque expression requiring case-by-case evaluation.
>
> Having analyzed these models and provided a more general heuristic argument, we believe we provide sufficient evidence that this saddle-point approach is worth the attention of the community. However, admittedly, providing a clear and transparent picture of its validity range and the scope of phenomena it can capture is beyond our current understanding. Below we thus partially address the points raised by the referee.
>
> Regarding the collectivity assumption, we have also recently analyzed a toy model where one can distinguish between collective large $n$ behavior and non-collective behavior. The model is that of a simple two-layer fully connected DNN. It has the advantage that can calculate the cumulant generating function and its first derivative, and additionally, any subsequent derivative is easy to compute in a straightforward manner and evaluate our saddle-point criterion. Specifically the cumulant generating function here is $\mathcal{C}(\vec{t}) = -\frac{N}{2}\log(1 - \sum_{\mu, \nu} it_{\mu} K(x_\mu,x_\nu) it_\nu / N)$ where $N$ is the width of the hidden layer and $K(x_\mu, x_\nu) = (x_\mu \cdot x_\nu)/d$, where $d$ is the input dimension. Here one notices that at large $n$ the behavior is collective for $n \gg d$ and non-collective for $n \ll d$. Indeed in the latter case taking $d \rightarrow \infty$ effectively makes $K(x_\mu, x_\nu)$ diagonal. The fact the collective behavior, beneficial for our saddle-point approach, only occurs in the limit where the GP already performs well--- sets this example apart from the previous two we considered. In accordance, we find that unlike in the previous cases, the saddle-point is appropriate here either at large $1/N$ (not $n/N$ as in common perturbation approaches) or interestingly for a large target. Depending on the reviewer we may include this discussion as an appendix in the revised submission.
>
> Regarding weight initialization, in practice, it is important to keep it on par with the weight decay, so as to ensure quicker numerical convergence to the steady-state Gibbs distribution. However, by definition, the initialization should not matter provided ergodicity holds. The main assumption, which was already put to the test directly (Refs. [24, 30]) and indirectly [https://arxiv.org/abs/1803.00885] in various works, is that in the over-parameterized regime, GD + noise dynamics is, to any practical extent, ergodic. Indeed in a system with many more degrees of freedom (DNN parameters) than constraints (data-points) ergodicity seems, at least in hindsight, like a natural first assumption [see also a related argument here https://arxiv.org/pdf/1406.2572.pdf].
>
> Regarding MSE versus, say, cross-entropy loss, we first note that in App. C (Eq (C.6.)) we show how the target shift formalism looks very similar there and works with the discrepancy $\hat{\delta} g$ in the regression replaced quite neatly with a discrepancy in terms of probabilities. Making analytical prediction in this more advanced setting is somewhat harder since we lose the ability to use the Equivalent Kernel approximation due to the non-quadratic form of the cross-entropy loss. Still one can imagine solving the self-consistent equations numerically. We believe this will yield sensible predictions but establishing this requires further work.
>
> Regarding the lazy-learning to feature-learning transition, our main aim was to demonstrate that our saddle-point approach works within a "feature learning" regime. The transition we focused on in this work is relevant for transfer learning. Indeed in a transfer learning setting the recipient DNN has no knowledge of the last layer weights $a_i$, and so extracting the outliers in $[\Sigma_W]_{ss'} = \sum_{c} w_{sc} w_{cs'}$ is a harder task. Doing this extraction by inspecting the spectrum (i.e. diagonalization), one finds a phase transition corresponding to the emergence of an outlier. One may therefore expect the recipient DNN to fail in extracting the signal/outlier in the regime where a more complicated algorithm (diagonalization) fails to do so.
>
> With knowledge of $a_i$ the circumstances differ, as one can filter out much of the random noise in $w_{cs}$. Indeed one can reconstruct the output of the DNN which contains some signature of $\mathbf{w}_*$ (provided the train-loss improved from its initialization value). Checking the possibility of a phase transition within the trained DNN ensemble and/or within our saddle-point approach, is an interesting question which is however left for future work.
>
>
> Under-parameterized vs. over-parameterized regimes, finite $\gamma$ $\sigma \to 0$ limit, avatar of discontinuity -
>
> The reviewer points to potential issues to taking low $C$ and $\sigma^2$, as the DNN may become under-parameterized. We first point out that our aim was to study the finitely over-parameterized regime, which is also the regime in which many real-world DNNs operate. Indeed for the values of $n$ and $\sigma^2$ we chose none of our models are in an under-parameterized regime. In particular:
> 1. All our models had non-noisy teachers of the same type as the student, hence as $n$ grows large, the constraints it imposes become redundant.
> 2. For our CNN model, even if one considers the $\sigma^2 \to 0$ limit the inverse GP kernel on the data is generally invertible for $n \le d$ whereas, due to the form of the teacher and student, we only need $n = O(\sqrt{d})$ to achieve perfect performance. Moreover even if one takes $n>d$, in which case the GP kernel on the data must have zero eigenvalues, then our noiseless teacher target will be orthogonal to the eigenvectors corresponding to these zero eigenvalues, allowing one to effectively invert the matrix (akin to the pseudo-inverse).
>
> It is indeed interesting to ask what would be potential avatars of the transition from the under-parameterized to the over-parameterized regimes in our formalism, especially when one takes $\sigma = 0$ and adds noise to the teacher. One interesting possibility which we haven't explored is that many saddle-point solutions will appear with similar amplitudes (i.e. $e^{-\mathcal{S}_{SP}}$). Another is that having a $\hat{}{\delta} g$ which diminishes too rapidly with $n$, will invalidate the saddle-point approach following the criterion we proposed in App. A.4 and the latter heuristic criterion.
>
>
> validity condition for the saddle-point approximation -
>
> We thank the reviewer for pointing out this work. One of the difficulties we foresee in applying such tools directly is the involved form of the non-linearity making a standard Hubbard-Stratonovich transformation less practical. Here we also point out a new panel added to Figure 1 of the main text showing that quite a few high order terms in the cumulant expansion are required in our CNN model to capture the DNN's behavior.
>
> In fact, we are actively working on some simpler models in the hope to cast our saddle-point approach in more standard terms. However, we do not have any publishable results on this at the moment.
>
>
> Cite Roberts and Yaida 2021, and comparing perturbative vs. non-perturbative approaches -
>
> We have added this Ref., and a comment emphasizing that one may expect perturbative approaches to be appropriate only when the GP limit is already a reasonable approximation to the DNN behavior. Even in such circumstances, we are not aware of any work estimating the benefit of perturbative corrections in real-world settings - namely, how much of the performance gap between GPs and DNNs can they carry. The fact that these corrections often scale as $n$ over the over-parameterization parameter [see for instance Fig. S5 in https://arxiv.org/pdf/1902.06720.pdf and also in the GD+noise context https://arxiv.org/abs/2004.01190] suggests that real-world DNNs may operate well outside the perturbative regime.
>
> In our paper, we purposely focused on toy examples where the finite DNN behaves drastically different from its corresponding GP, thus our non-perturbative approach is essential to capture the DNN behavior. This is demonstrated in an additional panel added to Fig. 1 (and App. H), that shows the predictions of the leading order perturbative corrections in our toy CNN example. The figure shows that the perturbative approach is completely inappropriate for this model, as it completely misses the CNN behavior, for sufficiently small values of $C$.

---

> ### Author Response · Authors · 2021-08-30
> **Creating a discussion**
>
> Dear Reviewer,
>
> We hope you found our reply satisfactory. However, if that's not the case we would be happy to clarify or expand on any specific matter whilst the discussion period is still open.
>
> Thank you

---

### Official Review · Reviewer_ZAE8 · 2021-07-26

**Rating:** 6
**Confidence:** 4

**Summary:**

This work addresses the question of how finite-width DNNs differ from their infinite-width GP limit. This is important since, while the GP limit is a powerful theoretical tool for studying DNNs, it's unable to capture _feature learning_, a key behavior of DNNs. The authors show that the mean predictor of a finite DNN trained with noisy gradients and weight decay corresponds to GP regression on a shifted target, and give an approximation for the shift based on the cumulants of the prior. They apply the framework to two toy models. In particular, for a two-layer linear CNN they derive a large-training-set approximation for the cumulants, and show that as the width increases, there is a phase transition out of the feature-learning regime.

**Limitations And Societal Impact:**

Yes

**Main Review:**

My impression is that this paper makes important contributions – particularly the characterization of the finite-width DNN as shifted GP regression and the the demonstration of feature learning. However, my familiarity with the topic is limited and the paper is written in a fairly inaccessibly way, which makes it hard for me to fully evaluate the contribution.


 * I felt that the experimental sections could be made more convincing, since as written, I found the discussion to be imprecise and hard to follow. For example:
	 * The discussion of how large the training set must be for the CNN vs the GP to "perform well" is somewhat vague.
	 * In section 4.3, the idea is to show that the prediction of $\alpha$ improves as $n$ grows. But it was unclear if the existence of such $\alpha$ depends on linearity occurring in the limit $C\rightarrow \infty$ anyways, (eqn. E.3)? And there is the fact that the linearity only results from the EK large $n$ limit, so it's confusing to say that the prediction of $\alpha$ is improving as $n\rightarrow\infty$. And there is the approximation from finding $\alpha_{test}$ using $\alpha_{train}$. In essence, I didn't follow the discussion well enough to feel confident that what is demonstrated indicates an improved prediction.
	 * In section two 4.4, I'm uncertain what is meant by saying the phase transition "becomes sharp as one takes $n, S  \rightarrow  \infty$." Is this demonstrated?

 * It would be nice to see some comparison of the predictions of this approach compared to those of a leading-order finite width correction, e.g. [30]. Are these other approaches incapable of exhibiting feature learning?
 * A number of terms and notation are introduced without explanation, and I'm unsure which are standard to someone more familiar with the topic. Perhaps adding additional references to background material could aid the uninitiated. For example, $C$ is not defined when first introduced, SRN is undefined, and I still don't fully understand what "self-consistent" means.
 * Minor: "self consistent" should be hyphenated everywhere.


**Time Spent Reviewing:**

5

---

> ### Author Response · Authors · 2021-08-10
> **Response to Reviewer ZAE8 - Part 1**
>
> paper is written in a fairly inaccessible way -
>
> We acknowledge that our presentation of the work has been less than optimal in several aspects. Following the detailed inputs from all the reviewers as well as input from several seminars we gave on this work, we believe the presentation has been improved (see list of main changes below as well as various smaller changes following the comments by the reviewers).
>
>
> how large the training set must be for the CNN vs the GP to "perform well" is somewhat vague -
>
> As the text now reads, for a GP trained in a regime where the Equivalent Kernel (EK) approximation holds (roughly, large $n$ and large $\sigma^2 n$), one can decompose the target on the basis of kernel eigenvectors and find that to capture well an eigenvector with eigenvalue $\lambda$, one must require  $\sigma^2/n \ll \lambda$ [Rasmussen and Williams, 2005]. For the case of the GP corresponding to our linear CNN model, there is only a single eigenvalue $\lambda=1/d$ and we used $\sigma^2=1.0$ in our experiments. Thus, we find that $n \ll d=NS$ yields poor GP performance while $n \gg d$ yields excellent performance with $n=d$ being the crossover scale.
>
> Turning to the CNN at small $C$, we are not aware of any off-the-shelf tools to predict its performance. In fact, our point is to offer such a tool. Still, one can reason that since a student can express the teacher's target while having only $C^*(N+S)$ free parameters (where $C^*$ is the number of channels in the teacher network), $n = C^*(N+S)$ will be the crossover scale to good performance.
> For small $C$ and large $N, S$ this opens a large regime where the GP is inferior to the CNN, in that the former requires much more training points than the latter in order to achieve similar performance.
>
>
> section 4.4: phase transition becomes sharp as one takes $n, S \to \infty$ -
>
> The phase transition we refer to here is signaled by the departure of an outlier eigenvalue (a signature of $\mathbf{w}^*$) from the MP-sea (the bulk of eigenvalues stemming from the noise). It is a well-established fact that as the two dimensions ($S, C$) of the matrix $\Sigma_W$ increase, this transition becomes sharp - that is as one varies the parameters of the model (e.g. the ratio of dimensions, the variance of the random entries, the size of the rank-one perturbation) there is a clear separation into a detectable from a non-detectable phase.
>
> The logical flow we use is that we:
> (1) Assume that the distribution of the weights indeed follows an MP distribution with a rank-1 perturbation.
> (2) Support this assumption numerically.
> (3) Deduce the parameters of the MP distribution and the rank 1 perturbation analytically.
> (4) Use the aforementioned results on the transition in the pristine MP case to argue that conditioned on point (1) there must be a phase transition here as well.
> In our case, it would imply that the outlier departs from the bulk close to the theoretical value of $C_{crit}$ (Eq. 18). This is demonstrated numerically in Fig. 2 panel B.
>
> We further note that the existence of a sharp transition between having an outlier and having no outlier in the spectrum of a large random matrix is quite a robust effect that does not go away under small perturbation of changes to the random matrix ensemble.
>
>
> Section 4.3, linearity and improved prediction of $\alpha$ as $n$ grows -
>
> Indeed the work contains two harmonious yet distinct approximations. One is the saddle-point approximation hinging on large $n$ and the collectivity assumption described in App. A.4. The outcome of this first step is a self-consistent equation involving GP inference on the same data set as the original one but with a shifted target ($\Delta g$). In general, predicting analytically GP inference is not straightforward and involves inverting a large and non-trivial matrix ($\tilde{K}^{-1}$). For the second model, we simply performed such inverses numerically, whereas for the CNN model we relied on the standard EK limit (and perturbative corrections to that limit) which holds, roughly speaking, when $\sigma^2 n$ is large (see Ref. [10] for a more detailed discussion). Importantly, the EK approximation washes away any specific dependence on the data points and instead yields the data-set average predictions which are a continuous function of $\mathbf{x}$, the input. Still, one finds that this continuous function captures well the behavior given a specific dataset at least at large $n$.
>
> In the CNN setting, EK and its corrections yield (as one can also argue based on symmetries) a prediction proportional to the target and $\alpha$ is simply the proportionality constant. Thus on a formal level, the existence of $\alpha$ does not require large $n$. However in practice, if one wishes to work in the regime where EK works well (perhaps with several perturbative corrections as in Ref. [10]) and in the regime where it faithfully captures the behavior of a single data-set rather than data-set averaged predictions --- large $n$ is indeed required, as the reviewer is right to mention.
>
> The $q$ factors ($q_{train}$ and $q_{test}$) used in the work, describe the aforementioned perturbative corrections to the simple EK limit--- more concretely, if $f_{EK}(x)$ is the prediction according to the strict EK limit, then $q_{test/train} f_{EK}(x)$ is the predictions on the test/train set taking into account all perturbative corrections to the EK limit.
>
> The fact that these corrections amount to two simple scaling factors stems from the fact that the GP kernel, in this case, is very simple and contains only a single set of degenerate eigenvalues. In the appendix, we derived $q_{train}$ analytically up to third order in the perturbation theory introduced in Ref. [10] and found good agreement with empirical values in the limit of large $C$. Similar derivation can be done for $q_{test}$. However in Fig. 1. of the main text, in order to disentangle these two sources of error (saddle-point corrections versus EK limit corrections), we extracted $q_{test}$ and $q_{train}$ empirically by solving the GP problem at $C \rightarrow \infty$ numerically. Then used these resulting parameters in the self-consistent equation to predict the performance at finite $C$.
>
> We comment that the EK approximation is rather well understood by now and was not the main focus of this work. We believe that the potential error it introduces is small compared to other effects. Indeed for the graphs at large $n$ the error due to different dataset draws was much smaller than other sources of error. Also, as we comment next, our full self-consistent theory yields predictions for the CNN behavior that are clearly much better than alternative theories (see next bullet point).
>
> To improve the presentation of this topic, we will include a short paragraph in the main text summarizing the above discussion.

---

> > ### Author Response · Authors · 2021-08-10
> > **Response to Reviewer ZAE8 - Part 2**
> >
> > Comparison of the quality of predictions to a perturbative approach, and specifically feature learning -
> > We thank the reviewer for this very appropriate comment, we have added these results in Fig. 1 on the test set and in App. H for the training set and added a short discussion of this issue. We have added the predictions of two extra theoretical results:
> > 1. The solution to the self-consistent equation Eq. 16 but taking into account only the fourth cumulant, which is found by solving the equation while ignoring the terms in $[\cdots]^{-1}$ that arise from the geometric series of summing all higher-order cumulants.
> >
> > 2. A purely first-order perturbative result, which is analogous to the results in Refs. [30, 37].
> >
> > Let us make a few comments on these results: In this toy model and parameter regime, the predictions of both 1. and 2. above are clearly inferior to our self-consistent predictions that take all cumulants into account. For sufficiently large $C$, all predictions, including our own theory, coincide as they all converge to the GP limit. Generally, we see that the predictions of 1. tend to overshoot the empirical $\alpha$ whereas those of 2. tend to undershoot, at least for sufficiently small $C$, to the point of predicting negative values of $\alpha$, which is clearly wrong. In contrast, the predictions of 1. above are always $0 \le \alpha \le 1$. Perturbation theory gives sensible results for large $C$ and small $n$, which is expected, but diverges very quickly as one departs from this regime.
> >
> >
> > unexplained terms and notation: $C$, SNR, self-consistent -
> >
> > Depending on the context, $C$ is either the number of channels of some CNN (as in our linear CNN toy example) or more generally the hyper-parameter that controls over-parameterization, as in §3 where it is defined this way. In other papers, it is simply named the "width" of the DNN, though notice that this can be confusing since for FCNs "width" means the number of neurons in a hidden layer whereas for CNNs it means the number of channels.
> >
> > SNR in the context of §4.4 corresponds to the ratio of the coefficients of the two terms in Eq. 17: the first term represents the noise (the bulk of the MP distribution) and the second term represents the signal (the magnitude of the rank-1 component of the teacher's feature).
> >
> > The term "self-consistent" is borrowed from statistical physics where it describes the equations one gets from a mean-field approximation (see for instance https://www2.ph.ed.ac.uk/~mevans/sp/sp10.pdf). It relates to our Eqs. 5,7,8 describing the target shift and the discrepancy of the predictor: the target shift is given in terms of the discrepancy and vice versa, so one can plug one of these sets of equations into the other set to obtain a set of closed implicit equations which can be solved numerically, as we do for several examples. In other words, we are looking for a target shift given by some function of the discrepancies which are predicted using a GP on that same target shift. This can be viewed a self-consistency condition for the target shift. We comment the saddle-point approximations are often related to mean-field approximations, in the sense that the saddle-point equations coincide with the mean-field equation.
> >
> > The revised version will contain extra explanations when we first introduce these three items.

---

> ### Author Response · Authors · 2021-08-30
> **Creating a discussion**
>
> Dear Reviewer,
>
> We hope you found our reply satisfactory. However, if that's not the case we would be happy to clarify or expand on any specific matter whilst the discussion period is still open.
>
> Thank you

---

> > ### Comment · Reviewer_ZAE8 · 2021-09-04
> > **Thanks for your reply.**
> >
> > Thanks very much for your detailed reply, which has helped clarify several things for me. I also appreciate the additional experiments comparing to other theories. My overall assessment is positive and I’d be happy to see this paper accepted. My score remains a 6 and my confidence is increased to 4.

---

### Decision · Program_Chairs · 2021-09-27

**Decision:**

Accept (Poster)

**Comment:**

This paper generated a lot of lively discussion! Across the board the reviewers thought the theoretical development was interesting and insightful. However, several of the referees thought the exposition could be made more approachable for non-experts. On the other hand, several of the reviewers were concerned that the teacher-student setting in this paper was a bit too idealized; they thought the authors did not do a good enough job synthesizing the results to come up with practical lessons for researchers that were backed up by experimental evidence on more realistic networks. Although I recommend acceptance since the theory seems quite interesting, I do think the impact of the work would be significantly improved if the authors take some time to run experiments on realistic systems before the camera ready deadline to see if vestiges of their theory remain.